# Subventricular zone/white matter microglia reconstitute the empty adult microglial niche in a dynamic wave

Lindsay A Hohsfield[1,2], Allison R Najafi[1,2], Yasamine Ghorbanian[3,4], Neelakshi Soni[1,2], Joshua Crapser[1,2], Dario X Figueroa Velez[1], Shan Jiang[5], Sarah E Royer[1,3,6], Sung Jin Kim[1,2], Caden M Henningfield[1,2], Aileen Anderson[1,2,3,6,7], Sunil P Gandhi[1], Ali Mortazavi[5], Matthew A Inlay[1,3,4], Kim N Green[1,2]*

[1]Department of Neurobiology and Behavior, Irvine, United States; [2]Institute for Memory Impairments and Neurological Disorders, Irvine, United States; [3]Sue and Bill Gross Stem Cell Research Center, Irvine, United States; [4]Department of Molecular Biology and Biochemistry, Irvine, United States; [5]Department of Developmental and Cell Biology, Irvine, United States; [6]Department of Anatomy and Neurobiology, Irvine, United States; [7]Department of Physical Medicine & Rehabilitation, University of California, Irvine, Irvine, United States

**Abstract** Microglia, the brain's resident myeloid cells, play central roles in brain defense, homeostasis, and disease. Using a prolonged colony-stimulating factor 1 receptor inhibitor (CSF1Ri) approach, we report an unprecedented level of microglial depletion and establish a model system that achieves an empty microglial niche in the adult brain. We identify a myeloid cell that migrates from the subventricular zone and associated white matter areas. Following CSF1Ri, these amoeboid cells migrate radially and tangentially in a dynamic wave filling the brain in a distinct pattern, to replace the microglial-depleted brain. These repopulating cells are enriched in disease-associated microglia genes and exhibit similar phenotypic and transcriptional profiles to white-matter-associated microglia. Our findings shed light on the overlapping and distinct functional complexity and diversity of myeloid cells of the CNS and provide new insight into repopulating microglia function and dynamics in the mouse brain.

*For correspondence: kngreen@uci.edu

## Introduction

Microglia represent the largest population of immune cells in the brain, constituting 5–10% of brain cells in the adult central nervous system (CNS). As resident tissue macrophages, microglia are responsible for immune defense and resolution, tissue maintenance, neuronal support, and synaptic integrity (*Salter and Stevens, 2017*; *Tay et al., 2017*; *Wolf et al., 2017*). Their central role in the CNS makes microglia attractive drug targets for neurological disorders/injuries. However, developing effective therapies that manipulate microglia requires further understanding of microglial origins, diversity, homeostasis, and dynamics.

Microglia arise from yolk sac-derived erythromyeloid progenitors and colonize the brain as embryonic microglia during early stages of development (i.e. E8.5 – E9.5) (*Ginhoux et al., 2010*; *Kierdorf et al., 2013*). These immature myeloid cells, displaying amoeboid morphology and high proliferative potential, enter the brain via the meninges and ventricles in mice (*Lelli et al., 2013*; *Ueno et al., 2013*; *Swinnen et al., 2013*; *Xavier et al., 2015*), as well as, via the leptomeninges, choroid plexus, and ventricular zone in humans (*Tay et al., 2017*; *Verney et al., 2010*; *Monier et al., 2007*; *Ginhoux et al., 2013*). Microglial colonization occurs first in the white matter

(WM) (e.g. internal capsule, external capsule, and cerebral peduncle) and continues to the sub- and then cortical plate as cells proliferate and migrate in a radial and tangential manner (*Verney et al., 2010*). In adulthood (P28 and onward), microglia become fully mature, exhibiting ramified morphology and expressing canonical microglial signature genes: *P2ry12*, *Tmem119*, *Siglech*, *Cx3cr1*, *Olfml3*, *Fcrls*, and *Sall1* (*Dubbelaar et al., 2018*; *Baufeld et al., 2018*).

Recent single-cell RNA sequencing studies have identified transcriptionally distinct microglial gene signatures associated with disease (e.g. disease-associated microglia [DAM], microglial neurodegenerative [MGnD] phenotype [*Keren-Shaul et al., 2017*; *Krasemann et al., 2017*; *Mathys et al., 2017*; *Masuda et al., 2019*]), injury (e.g. injury-responsive microglia [IRM] *Hammond et al., 2019*), and brain region-specific areas/developmental stages (e.g. proliferative-region-associated microglia [PAM], axon tract-associated microglia [ATM] *Masuda et al., 2019*; *Hammond et al., 2019*; *Li et al., 2019*; *Matcovitch-Natan et al., 2016*), including the recent discovery of white matter-associated microglia (WAM) (*Safaiyan et al., 2021*). WAMs have been identified as a population of microglia in WM tracts from the corpus callosum that share parts of the DAM gene signature, including genes involved in phagocytosis, and increase with aging and disease (*Safaiyan et al., 2021*). In line with this, a distinct subset of microglia (PAMs/ATMs) has been described in the axonal tracts of the corpus callosum during development, sharing not only a similar location to WAMs, but morphological (i.e. amoeboid) and phagocytosis-associated gene profile (*Masuda et al., 2019*; *Hammond et al., 2019*; *Li et al., 2019*). Despite this, homeostatic microglia appear less heterogeneous during adulthood (*Hammond et al., 2019*; *Li et al., 2019*). These findings shed light on microglial diversity and state changes during health and disease; however, it remains unclear whether adult homeostatic microglia exist as one population, with the ability to change from one transcriptional/functional state to another, or whether they exist as heterogeneous subpopulations with distinct propensities.

Microglial homeostasis and dynamics are maintained by many signaling factors, including transforming growth factor-beta, Il-34, and colony-stimulating factor 1 (CSF1) (*Butovsky et al., 2014*; *Elmore et al., 2014*). Recent studies exploring the homeostatic kinetics of the microglia in the adult brain have revealed that these cells are long-lived (*Tay et al., 2017*; *Réu et al., 2017*) and self-renew, even after acute 80–95% depletion and subsequent repopulation (*Elmore et al., 2014*; *Zhan et al., 2019*; *Bruttger et al., 2015*). However, no approach to date has been able to deplete all microglia (*Waisman et al., 2015*; *Han et al., 2017*), and the rapid proliferation of surviving microglial cells would obscure the detection of other myeloid cells that contribute to the CNS environment and/or repopulation in the adult brain. While conventional depletion paradigms have shown that microglia have a remarkable capacity to repopulate from the presence of few surviving cells, we sought to investigate the consequences of eliminating these few remaining cells on microglial population dynamics.

To address this, we have optimized a colony-stimulating factor 1 receptor (CSF1R) inhibitor approach that involves sustained inhibitor administration, building on our prior work that microglia are dependent on this signaling for their survival (*Elmore et al., 2014*). This approach results in a delayed repopulation of myeloid cells that reconstitute the brain in a sequential manner previously unseen in the adult brain. We show that repopulating cells emerge from the subventricular zone (SVZ)/WM areas and traffic throughout the brain parenchyma via WM tracts before spreading out radially and tangentially through the rest of the brain in a dynamic wave of proliferating cells. Following full brain reconstitution, these repopulating cells remain phenotypically, transcriptionally, and functionally distinct from endogenous microglia, demonstrating unique gene expression profiles that are enriched for DAM genes and unique phenotypic properties similar to WAMs. Together, these data highlight the utility of CSF1R inhibitors in identifying and studying myeloid cell homeostasis and dynamics.

## Results

### Sustained high dose of CSF1R inhibitor unmasks a distinct form of myeloid cell CNS repopulation

In previous studies, we have shown that 7 day treatment of the brain penetrant CSF1R/KIT/FLT3 inhibitor PLX3397 (Pexidartinib; 600 ppm in chow) eliminates ~90–98% of microglia in the CNS (*Elmore et al., 2014*; *Najafi et al., 2018*). During depletion, surviving microglia are seen scattered

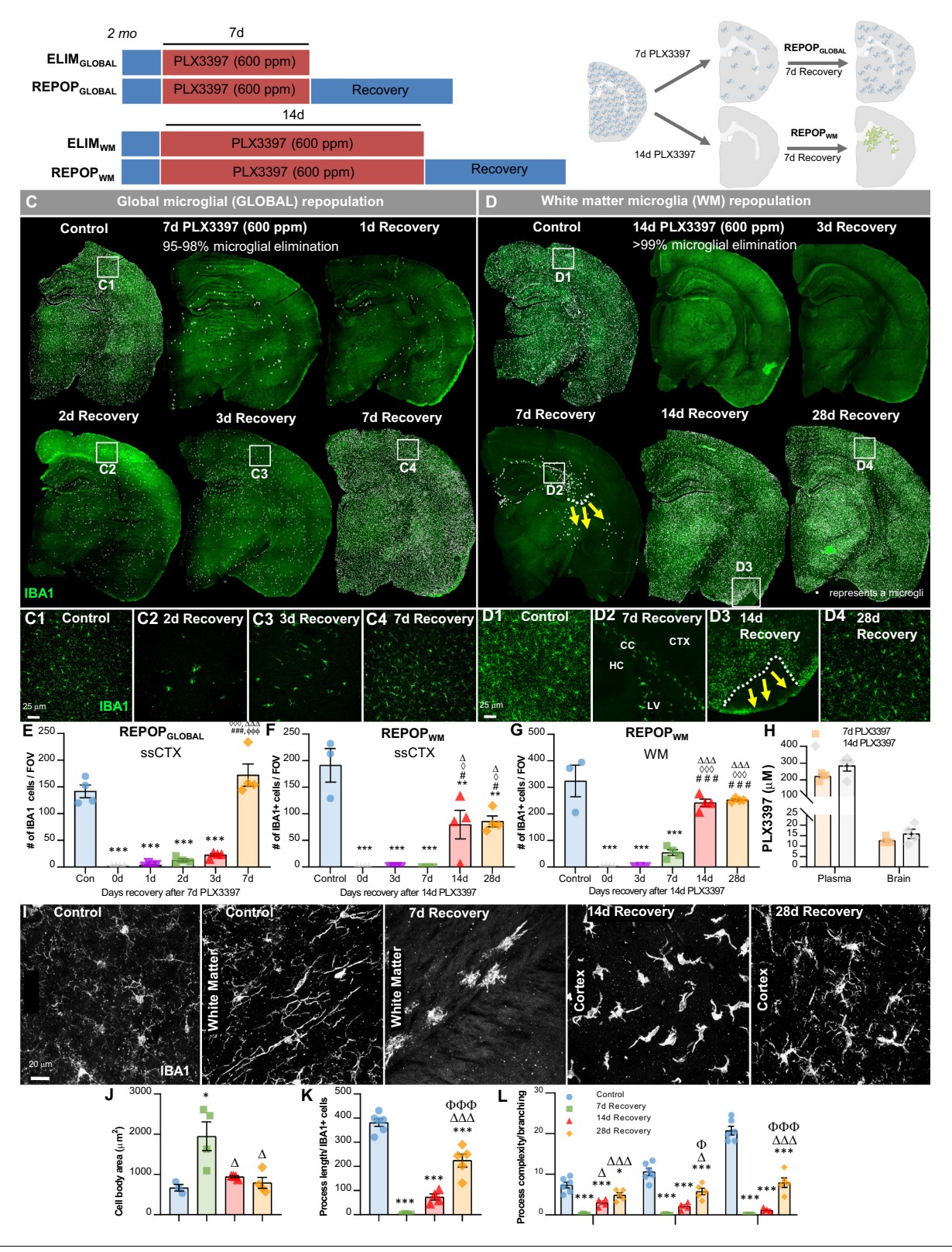

**Figure 1.** Sustained high dose of CSF1R inhibitor unmasks a distinct form of myeloid cell CNS repopulation. (A–B) Experimental paradigm and schematic depicting dose and duration of PLX3397 (600 ppm) treatment and subsequent inhibitor withdrawal allowing for global microglial (GLOBAL) and white matter microglia (WM) repopulation. For GLOBAL repopulation: 2-month-old wild-type (WT) mice were treated with 600 ppm of PLX3397 for 7 days, achieving ~90–98% brain-wide microglial depletion, with remaining microglia visibility dispersed throughout the brain parenchyma, and then

*Figure 1 continued on next page*

*Figure 1 continued*

placed on control diet for 7 days (7d recovery) allowing for microglial repopulation. At 7 day recovery, repopulating microglia reconstitute the brain from areas in which previously remaining microglia were deposited. For WM repopulation: 2-month-old WT mice were treated with 600 ppm of PLX3397 for 14 days, achieving 99.98% brain-wide microglial depletion, and then placed on control diet for 7 days allowing for microglial repopulation. At 7 day recovery, repopulating myeloid cells reconstitute the brain in specific neuroanatomical niches (e.g. ventricle, subventricular zone, white matter tracts, caudoputamen). (C–D) Representative immunofluorescence whole brain images of myeloid cells (IBA1, green) at each time point of treatment and recovery during GLOBAL (C) and WM (D) repopulation, with white dots superimposed over microglia. Due to the differential kinetics of these two forms of repopulation, repopulation (i.e. recovery) was evaluated at different timepoints, including initial stages with few cells, mid-repopulation, and full brain reconstitution. For GLOBAL repopulation: 2-month-old WT mice treated with control, 7 days of PLX3397 (7d PLX3397), 7 days of PLX3397 followed by 1 day on control diet (1d Recovery), 7 day of PLX3397 followed by 2 days on control diet (2d Recovery), 7 days of PLX3397 followed by 3 days on control diet (3d Recovery), and 7 days of PLX3397 followed by 7 days on control diet (7d Recovery). For WM repopulation: 2-month-old WT mice treated with control, 14 days of PLX3397 (14d PLX3397), 14 days of PLX3397 followed by 3 days on control diet (3d Recovery), 14 days of PLX3397 followed by 7 days on control diet (7d Recovery), 14 days of PLX3397 followed by 14 days on control diet (14d Recovery), and 14 days of PLX3397 followed by 28 days on control diet (28d Recovery). (C1–C4, D1–D4) Inserts of higher resolution confocal images of IBA1$^+$ cells during repopulation. White dotted lines and yellow arrows highlight 'wave' edge and direction. (E–G) Quantification of IBA1$^+$ cells per field of view (FOV) at each time point in cortical and white matter regions, respectively during GLOBAL (E) and WM (F–G) repopulation. (H) Pharmacokinetics analysis of PLX3397 levels in plasma and brain of mice treated with 7 day and 14 day PLX3397 (600 ppm). (I) Representative 63x immunofluorescence images of myeloid cells (IBA1, white) display morphological alterations. (J–L) Quantification of IBA1$^+$ cell morphology: cell body area in the white matter tract (J), average process/ filament length (K), and process complexity/branching (L) in the piriform cortex. Level 1–3+ indicates level of branching from the cell body. Data are represented as mean ± SEM (n=3–4). *p < 0.05, ** p < 0.01, *** p < 0.001; significance symbols represent comparisons between groups: (E) control *, 0d #, 1d ◊, 2d Δ, 3d Φ; (F–G) control *, 0d #, 3d ◊, 7d Δ. CC, corpus callosum; CTX, cortex; HC, hippocampus; LV, lateral ventricle.

The online version of this article includes the following source data for figure 1:

**Source data 1.** Sustained high dose of CSF1R inhibitor unmasks a distinct form of myeloid cell CNS repopulation.

throughout the brain (*Figure 1A–B,C*) and subsequent withdrawal of the inhibitor results in rapid and spatially homogenous microglial repopulation within 3 days, with cells exceeding control numbers by 7 days (*Figure 1C,E*). Recent studies show that repopulation is dependent on the local proliferation and clonal expansion of surviving microglia (*Elmore et al., 2014*; *Zhan et al., 2019*; *Bruttger et al., 2015*; *Huang et al., 2018*; *Zhan et al., 2020*; *Mendes et al., 2021*); thus, we refer to this type of repopulation as global microglial (GLOBAL) repopulation.

Here, we set out to examine CNS myeloid cell repopulation dynamics in the absence of remaining microglia in the brain. To accomplish this, we utilized a sustained high dose of PLX3397 (600 ppm) for 14 days. This treatment results in pharmacologically unprecedented microglial depletion, in which we observe no IBA1$^+$ cells across whole brain sections (*Figure 1A–B,D*). Although PLX5622 is a CSF1R inhibitor (CSF1Ri) that is more active against CSF1R compared to other related kinases, studies in our lab have shown that high dose PLX3397 vs. high dose PLX5622 achieves higher CNS exposure and microglial depletion efficiency. Pharmacokinetic analysis of microglia-depleted brains at both 7d and 14d treatment of PLX3397 shows that PLX3397 levels remain the same in the CNS despite longer drug exposure (*Figure 1H*).

To explore the differential repopulation dynamics between these microglial elimination paradigms, we treated mice with PLX3397 for 14 days and then withdrew the inhibitor, allowing the CNS to recover for 3, 7, 14, and 28 days (*Figure 1D,F–G*), and compared it to GLOBAL repopulation (*Figure 1C,E*). At 3 day recovery following 14 day PLX3397 treatment, no IBA1$^+$ cells are detectable in most brain sections. By 7 day recovery, IBA1$^+$ cells appear, but are exclusively located near the lateral ventricle and in WM tracts lining the ventricles (*Figure 1D2*). By 14 day recovery, IBA1$^+$ cells have spread throughout most of the CNS; however, some areas of the cortex (e.g. the piriform cortex) remain unoccupied (*Figure 1D3*). These areas display a distinct 'wave' of cells in adjacent unoccupied cortical areas (*Figure 1D3*). At 28 day repopulation, all brain regions are populated with IBA1$^+$ cells, but absolute cell numbers remain 50% lower compared to microglia in control animals, as seen in somatosensory cortices (*Figure 1D4, F*). We subsequently refer to this form of repopulation as white matter microglia (WM) repopulation, due to its distinct characteristics from GLOBAL repopulation.

We have previously shown that within 14–21 days of GLOBAL repopulation, repopulating microglia not only attain similar densities to resident microglia, but also display similar morphologies, cell surface marker expression, gene expression profiles, and inflammatory responses to LPS (*Elmore et al., 2014*; *Elmore et al., 2015*; *Elmore et al., 2018*). In contrast, WM repopulating cells

display larger cell bodies (*Figure 1I–J*) after 7 day recovery (which normalize by 14 day recovery), as well as reduced process/filament length (*Figure 1I,K*) and reduced process/dendrite branching and complexity (*Figure 1I,L*) compared to homeostatic microglia even after 28 days recovery.

## WM repopulation elicits a dynamic wave of repopulating proliferative myeloid cells

In GLOBAL repopulation, microglia repopulate the brain parenchyma in a homogeneous fashion, with repopulating cells displaying no preference for specific locations (*Figure 1C*). In contrast, WM repopulating cells first appear in precise ventricular and WM locations (*Figure 1D2*). To expand upon this initial observation, we sought to define the anatomical niches of this distinct form of repopulation and built a spatial heat map at three brain positions (Bregma 2.58 mm, 1.10 mm, −2.06 mm) along the rostral-caudal axis in brains of mice at 7 day recovery (*Figure 2A*). Throughout this axis, IBA1$^+$ cells initially repopulate the brain within the caudoputamen, particularly in areas near the lateral ventricle and associated WM tracts. At rostral regions of the brain, cells are found within the rostral migratory stream (RMS), a projecting axonal tract from the SVZ to the olfactory bulb. In more caudal brain regions, repopulating cells are seen near the SVZ, caudoputamen, and corpus callosum. Subsequent analysis of the entire brain confirms the presence of these early repopulating cells in areas near the SVZ/ventricular zones, WM tracts, and caudoputamen.

Repopulating cells migrate in a radial and tangential fashion, initially filling the WM and caudoputamen before spreading out through the cortex in a dynamic wave between 7 and 14 day recovery (*Figure 2B–C*). Analysis of the migratory wave during this recovery time from Bregma 0.445 to Bregma −0.08 shows repopulating cell deposition appears to occur in stages, with symmetrical distribution and expansion (*Figure 2—figure supplement 1A*). At Stage I, cells are visible near the lateral ventricle at the intersection of the corpus callosum, caudoputmen, and SVZ and appear to migrate in a radial migration pattern from the ventricular zone into and filling the caudoputamen in an inferior direction. At this stage, cells also appear in WM tracts, specifically in the corpus callosum. At Stage II, cells begin to migrate in a tangential migration pattern utilizing WM tracts to migrate into WM areas or areas near WM tracts. At Stage III, repopulating cells continue to migrate in both a radial and tangential migratory pattern filling ~80–95% of the striatum, including the caudoputamen, lateral septal complex, and pallidum. It is also apparent that repopulating cells are restricted or unable to migrate past the WM tract or corpus callosum between the cerebral nuclei and cerebral cortex (*Figure 2—figure supplement 1B*). At Stage IV, the cells have broken through this WM tract barrier and migrate out in a radial migration pattern moving from the subcortical to the cortical plate (*Figure 2B–D*; *Figure 2—figure supplement 1B*). During Stages III-IV, patches of cortical microglial expansion are occasionally seen; however, the majority of proliferating/Ki67$^+$ cells that contribute to WM repopulation are found in the wave rather than in cortical clusters of expanding microglia.

In contrast to the proliferative profile of previously described GLOBAL-repopulating microglia, in which remaining microglia proliferate throughout the brain to give rise to newly repopulating cells, proliferating WM repopulating myeloid cells are initially found near ventricles and WM tracts (*Figure 2E and E1*, *Figure 2F–G*). As the cells spread and migrate through the brain, proliferation remains localized within the cell 'wavefront' (*Figure 2F, F and F4*). Once out of the front (i.e. in the wake of the wave) IBA1$^+$ cells appear to stop proliferating (i.e. Ki67$^-$; *Figure 2F3*). This wave of proliferating cells is most apparent at Stage IV, led by a wavefront with an average width of ~100–150 µm of amoeboid and proliferating myeloid cells.

## Extensive CSF1R inhibition unveils the presence of CSF1Ri-resistant myeloid cell in the subventricular zone/white matter areas

Having demonstrated that 14 day PLX3397 (600 ppm) treatment and subsequent withdrawal results in reconstitution of the adult brain with a phenotypically distinct myeloid cell with unique tempo-spatial migratory patterns, we next sought to determine the source of these cells. To that end, we first confirmed the extent of microglial depletion with multiple myeloid markers, including microglial-specific P2RY12 and TMEM119 (*Figure 3—figure supplement 1A–B*), as well as in *Cx3cr1$^{CreERT2}$* mice, to explore whether surviving cells were present but just downregulating myeloid markers. In these mice, YFP is permanently expressed in microglia following tamoxifen-inducible lineage tracing, illustrating that depletion is not due to a downregulation in microglial markers, but a loss of cells

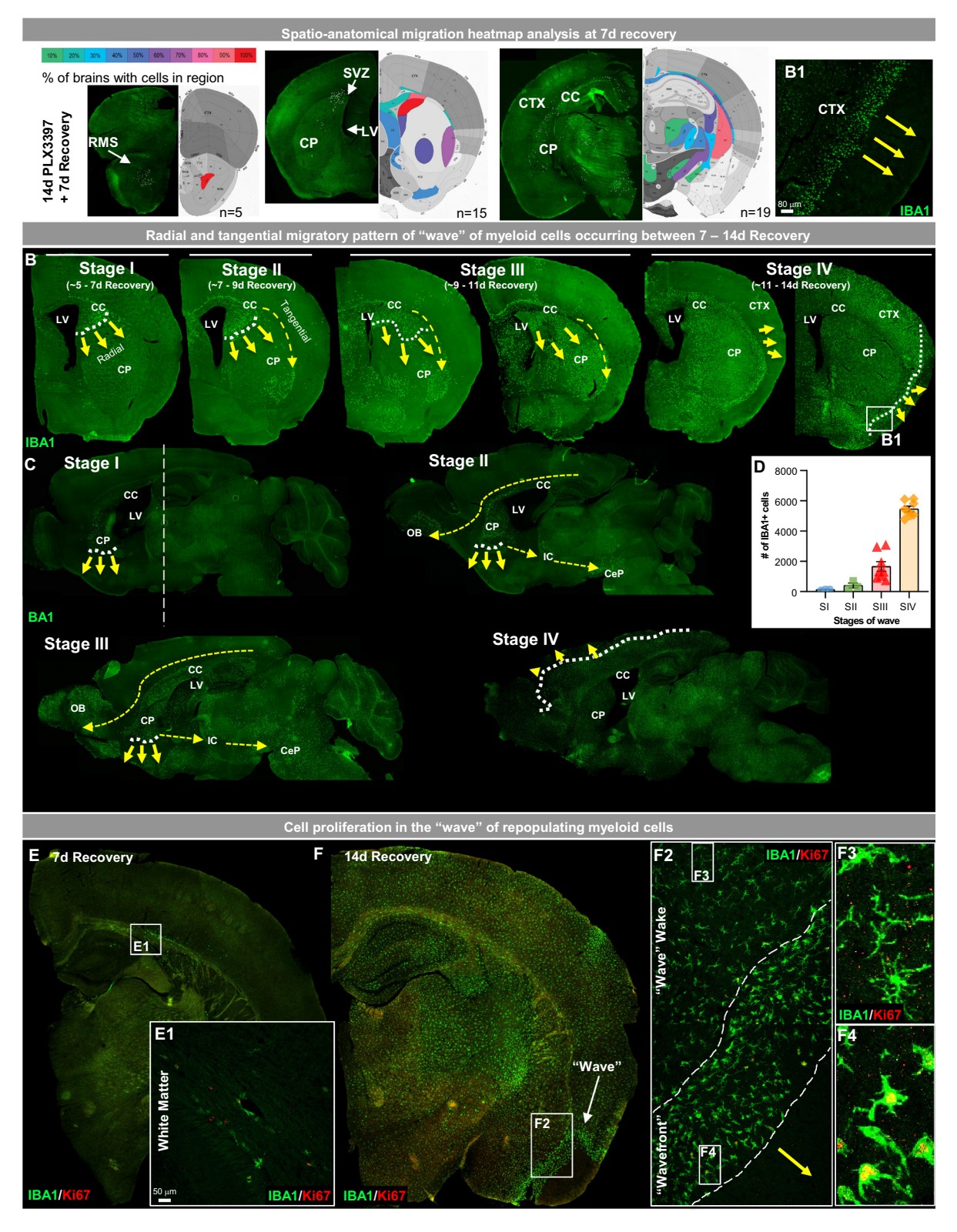

**Figure 2.** White matter microglia (WM) repopulation elicits a dynamic wave of repopulating proliferative myeloid cells. (A) Spatial heat map of WM repopulation at 7 day recovery in three brain positions (Bregma 2.58 mm, 1.10 mm, −2.06 mm) depicting % of brains with cells present in specific brain regions (n=5–19). (B–C) Immunofluorescence whole brain coronal (B) and sagittal (C) section images of IBA1[+] cells (green) show a sequential time course of the 'wave' (see higher resolution image of wavefront in B1 insert) of myeloid cells filling the brain between 7 and 14 day recovery during WM

*Figure 2 continued on next page*

*Figure 2 continued*

repopulation. White dotted lines and yellow arrows indicate the edge and direction of the 'wave', highlighting the radial migratory patterns of WM repopulating cells. Yellow dashed arrows indicate the direction of the tangential migratory pattern of WM repopulating cells, mostly utilizing WM tracts. The straight white dashed line in C shows the Bregma position at which coronal sections were taken for B. (E–F) Representative immunofluorescence whole brain images of myeloid cells (IBA1, green) and proliferating cells (Ki67, red) at 7 (A) and 14 (B) day recovery. Insets show higher resolution of IBA1 and Ki67 colocalization in initially repopulating cells (A1) and in cells outside and inside the wavefront (B2-4). Data are represented as mean ± SEM (n=3–8).

The online version of this article includes the following source data and figure supplement(s) for figure 2:

**Source data 1.** White matter microglia (WM) repopulation elicits a dynamic wave of repopulating proliferative myeloid cells.

**Figure supplement 1.** White matter microglia (WM) repopulation elicits a dynamic wave of repopulating proliferative myeloid cells.

(*Figure 3—figure supplement 1C*). While in previous analyses, we observed no IBA1[+] (including Cd11b[+], P2RY12[+], and TMEM119[+]) cells throughout the brains following 14 day PLX3397 treatment, we next conducted an examination throughout the entire brain along the rostral-caudal axis (i.e. every 6th section). With this extensive analysis, we observe a very small number of surviving IBA1[+] cells in treated brains (~15 in 14 day treated PLX3397 brains vs. ~132,000 in control brains = 99.98% depletion; *Figure 3—figure supplement 1D*). Despite this, we describe the highest reported loss of microglial cells in the adult brain. Notably, these few cells (0.02% of cells) are seen exclusively in ventricular (i.e. SVZ) and adjacent WM areas (*Figure 3A–D*). These cells display a lack of canonical microglial markers, including P2RY12 and TMEM119, as well as distinct morphological profiles (*Figure 3E–G*). Examination of brains depleted for 3.5 month PLX3397 (600 ppm) revealed no further surviving cells in the WM/WVZ, suggesting that these cells eventually succumb to CSF1Ri (*Figure 3—figure supplement 1F–H*). These findings indicate that surviving SVZ/WM microglia may possess a different sensitivity to CSF1R inhibitors, possibly relying on other growth factors for survival, or that CSF1Ri kinetics in WM may be different to grey matter areas due to lipid abundance or inhibitor solubility in lipids. Reports show that the population of microglia residing in the adult SVZ and adjacent RMS display a distinct morphological profile with an amoeboid cell body and fewer/shorter branched processes and exhibit an activated phenotype, similar to WM repopulating cells (*Ribeiro Xavier et al., 2015*; *Goings et al., 2006*; *Böttcher et al., 2019*). Prior descriptions of myeloid cells found in the adult SVZ have found lower expression levels of the microglial-specific marker P2RY12 (*Ribeiro Xavier et al., 2015*). Here, we find repopulating cells are initially negative for both microglial-specific P2RY12 and TMEM119 surface markers, however, express these markers by 28 day recovery (*Figure 3H–I*, *Figure 3—figure supplement 1E*).

To gain insight into the transcriptional profile of these WM repopulating cells, we measured mRNA transcript levels from control, 14 day PLX3397-treated, 7 day recovery, and 28 day recovery whole brain hemispheres, using a Nanostring Immune Profiling panel (~700 immunology-related genes) (*Figure 3J*). Comparing 14 day PLX3397-treated brains (i.e. microglial-depleted) to 7 day recovery brains allowed us to explore the gene expression profile of the initial repopulating cells. This comparison revealed that the most upregulated genes in the 7 day recovery brain are involved in myeloid cell activation/priming, pathogen sensing, and monocyte-macrophage signaling (e.g. *Mrc1, C3ar1, Ccl12, Clec7a, Ccr2/Ccl2, Cybb*, and *Ccl9*), rather than homeostatic microglia signature genes (*Figure 3K*). In comparing 28 day recovery brains to controls, increased expression was detected in several genes associated with myeloid cell signaling (*Ccl8, Cmah, Ly9, Lyz2, Tlr8, C4b*), in particular, major histocompatibility complex I (*H2-D1*) and II (*H2-Aa, H2-Ab1, Cd74*) components and microglial priming (*Clec7a, Cybb*) (*Figure 3L*).

To explore whether WM repopulating microglia maintain resistance to CSF1Ri following their migration from the SVZ, we treated 7 and 14 day recovery mice for 7 days with PLX3397 (600 ppm) (*Figure 3—figure supplement 1I*). At both recovery times, the majority of cells were eliminated (*Figure 3—figure supplement 1K*) showing that WM repopulated cells are susceptible to CSF1Ri treatment and require CSF1R signaling for their survival. Together, these data provide evidence for the existence of a very small population of myeloid cells located in the SVZ and adjacent WM tracts that can uniquely survive 14 day PLX3397 high dose CSF1Ri treatment.

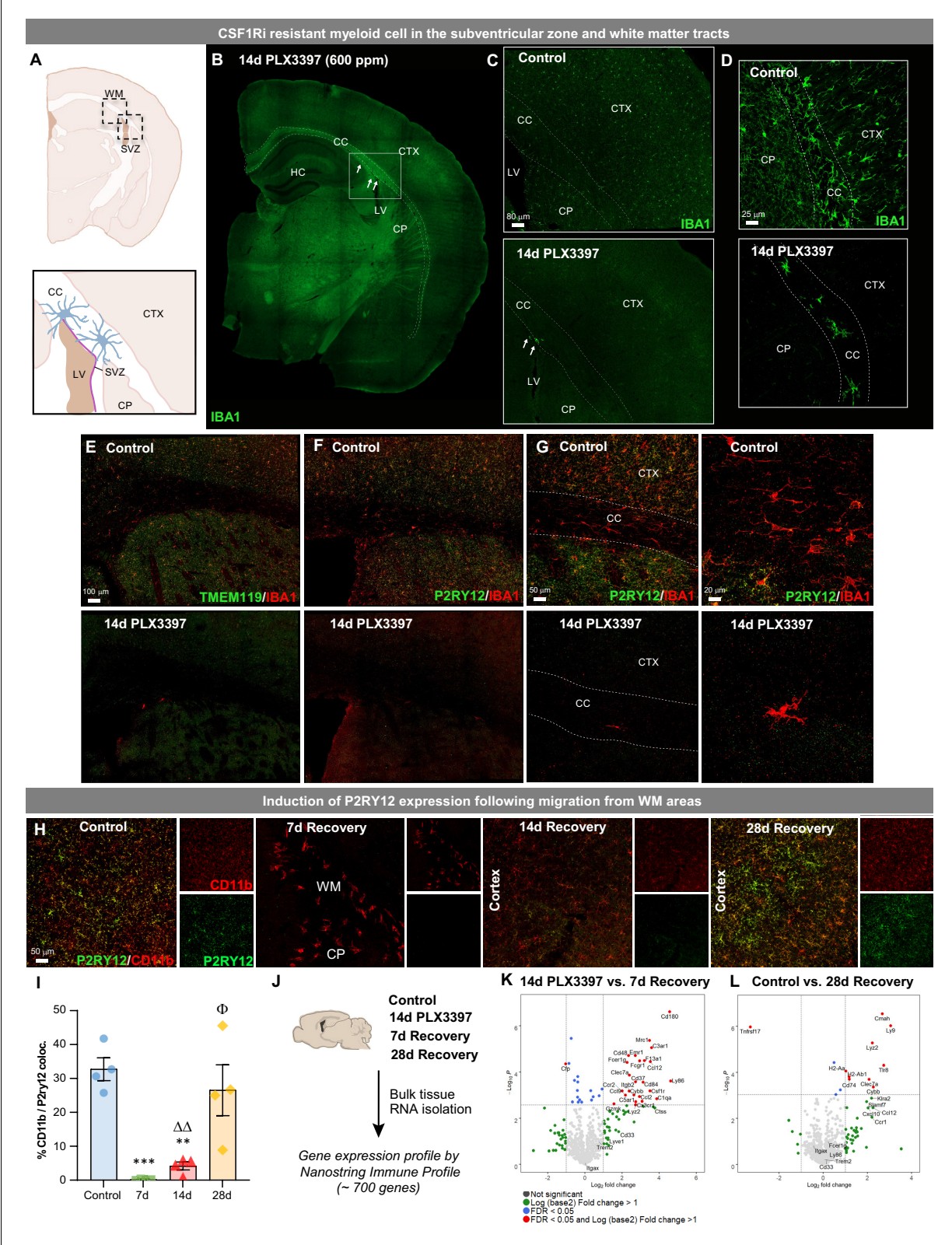

**Figure 3.** Extensive CSF1R inhibition unveils the presence of CSF1Ri-resistant myeloid cells in the subventricular zone and white matter tracts. Two-month-old WT mice treated with vehicle or PLX3397 (600 ppm in chow) for 14 days to evaluate extent of microglial depletion. (**A**) Brain section schematic of SVZ and WM areas where CSF1Ri-resistant myeloid cells are present following 14 day PLX3397 (600ppm in chow) treatment. (**B**) Representative whole brain slice image of CSF1Ri-resistant IBA1$^+$ cells (white arrows) near the SVZ/WM areas (white box) surrounding the lateral
*Figure 3 continued on next page*

Figure 3 continued

ventricle. (C–D) Representative tile scan (C) and high-resolution confocal (D) images of control and 14 day PLX3397 mice showing the deposition of surviving SVZ/WM IBA1+ (green) cells in WM areas (white dotted lines). (E–G) Representative tile scan (E–F) and confocal immunofluorescence image of IBA1+ cells co-stained with microglial-specific markers TMEM119 (E) and P2RY12 (F–G) in control and 14 day PLX3397 mice, showing that CSF1Ri-resistant cells are TMEM119- and P2RY12-. Higher resolution images illustrate atypical morphological profile of CSF1Ri-resistant cells. (H) Representative 20x images of myeloid cells (Cd11b, red) and P2RY12 (green). (I) Quantification of % colocalization of CD11b+ and P2RY12+ cells as seen in (H). (J) Control, 14 day PLX3397, 7 day recovery, and 28 day recovery mouse hemispheres were collected and analyzed for bulk-tissue gene expression changes using Nanostring Immune Profile. (K–L) Volcano plots displaying the fold change of genes (log2 scale) and their significance (y axis, -log10 scale) between 14 day PLX3397 depleted vs. 7 day recovery mice (K) and control vs 28 day recovery (L). Data are represented as mean ± SEM (n=3–5). *p < 0.05, ** p < 0.01, *** p < 0.001; significance symbols represent comparisons between groups: control *, 0d #, 3d ◊, 7d Δ, 14d Φ. CC, corpus callosum; CP, caudoputamen; CTX, cortex; HC, hippocampus; LV, lateral ventricle; SVZ, subventricular zone; WM, white matter.
The online version of this article includes the following source data and figure supplement(s) for figure 3:

Source data 1. Extensive CSF1R inhibition unveils the presence of CSF1Ri-resistant myeloid cells in the subventricular zone and white matter tracts.
Figure supplement 1. Extensive CSF1R inhibition unveils the presence of CSF1Ri-resistant myeloid cell in the subventricular zone/white matter areas.

## WM repopulation occurs due to an unprecedented level of microglial depletion

To conclusively determine whether this unique form of repopulation occurs as a result of the unprecedented level of microglial depletion vs. a 14 day requisite CSF1Ri drug treatment, we utilized H2K-BCL2 mice, a transgenic mouse that overexpresses BCL2 in all hematopoietic cells (*Domen et al., 1998*). Similar to reports in Vav-Bcl2 mice (*Askew et al., 2017*), these mice display elevated microglial densities (*Figure 4A*). In H2K-BCL2 mice, 14 day treatment with PLX3397 (600 ppm) leads to a 61–95% elimination of microglia (*Figure 4B–D*). In line with this, a previous study has shown that overexpression of BCL2 in myeloid cells affords some resistance to tissue macrophage loss in Osteopetrotic (op/op) mice, a mouse lacking functional CSF1 (*Lagasse and Weissman, 1997*).

BCL2 is a major regulator of apoptosis, and enhanced *Bcl2* expression promotes the survival of cells of myeloid cells (*Ogilvy et al., 1999*), thus we postulate that BCL2 protection from cell death occurs through reduction in apoptosis. We and others have shown that CSF1Ri-induced microglial cell death is caspase-dependent (*Elmore et al., 2014*; *Hagan et al., 2020*); however, other mechanisms outside of apoptosis and necroptosis (*Bohlen et al., 2017*) could also play a contributing role (e.g. protease/autophagy). Importantly, subsequent withdrawal of CSF1Ri elicits GLOBAL repopulation, rather than WM repopulation (*Figure 4B–D*), indicating that WM repopulation occurs due to the unprecedented level of microglia depletion rather than drug treatment.

## WM repopulating myeloid cells derive from an existing *Cx3cr1+* cell source originating from the SVZ/WM area

Since repopulating cells first appear in the SVZ and the SVZ is a notable neurogenic/proliferative niche in the brain, we next stained sections containing the SVZ in control, 14 day PLX3397, 5 day and 7 day recovery groups for known precursor cell markers, as well as other cell lineage markers. Between 3 and 5 day recovery, repopulating cells transiently express NESTIN (92% of IBA1 cells were NESTIN+ at 5 day recovery; *Figure 5A*, *Figure 5—figure supplement 1F*), MASH1 (81% of IBA1 cells were MASH1+ at 5 day recovery; *Figure 5B*, *Figure 5—figure supplement 1G*), and TIE2 (50% of IBA1 cells were TIE2+ at 5 day recovery; *Figure 5—figure supplement 1A,H*), but are negative for GFAP (0% of IBA1 cells were GFAP+ at any timepoint during recovery; *Figure 5—figure supplement 1B*), DCX (0% of IBA1 cells were DCX+ during recovery; *Figure 5—figure supplement 1C*), OLIG2 (0% of IBA1 cells were OLIG2+ during recovery; *Figure 5—figure supplement 1D*), and SOX2 (0% of IBA1 cells were SOX2+ during recovery; *Figure 5—figure supplement 1E*) at all timepoints. Consequently, we performed lineage tracing using tamoxifen-inducible *Cre*-recombinases under control of the *Nestin* (*Nestin^CreERT2*) and *Ascl* (*Ascl1^CreERT2*, note: *Ascl1* encodes for MASH1) promoters, along with the myeloid cell-specific line (*Cx3cr1^CreERT2*). *Cre* lines were crossed with YFP reporters for visualization of induced expression (*Figure 5—figure supplement 2A*). Tamoxifen was given immediately following PLX3397 treatment to track the lineage of repopulation cells, except for *Cx3cr1^CreERT2* mice - which were given a 21 day washout period (i.e. tamoxifen was administered 21 days prior to PLX3397 treatment) to restrict labeling to resident vs short-lived peripheral myeloid cells (*Goldmann et al., 2013*). These studies revealed that repopulating cells do not originate from

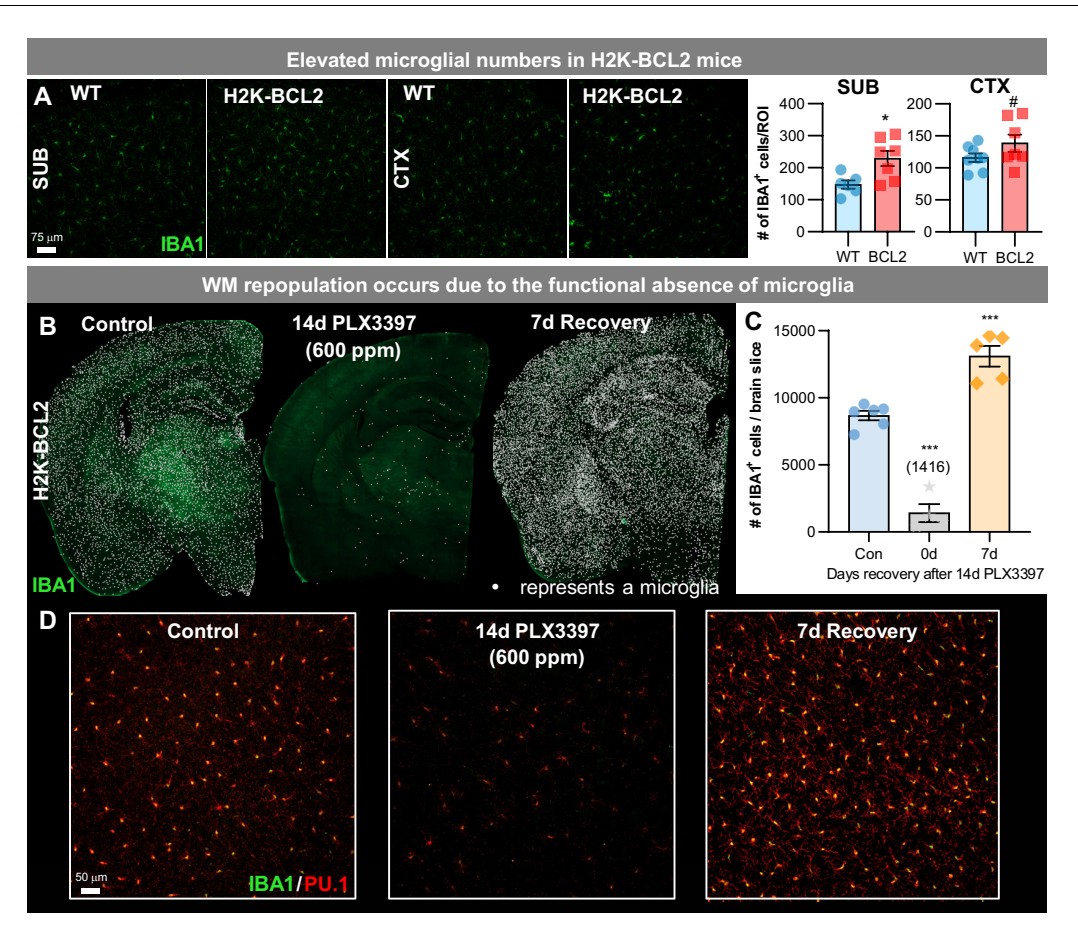

**Figure 4.** WM repopulation occurs due to an unprecedented level of microglial depletion. (**A**) Representative 20x immunofluorescence images of IBA1+ (green) cells in WT and H2K-BCL2 mice. Quantification of IBA1+ cells per region of interest (ROI) in subiculum (SUB) and cortex (CTX). (**B–D**) H2K-BCL2 mice were treated with PLX3397 for 14 days (600 ppm in chow), the drug was withdrawn, and then mice were provided with 7 days to recover, allowing for repopulation. (**B**) Representative whole brain images of IBA1+ (green) cells in control, 14 day PLX3397, and 7 day recovery mice, with white dots superimposed over microglia, showing incomplete microglial depletion leads to GLOBAL repopulation. (**C**) Quantification of IBA1+ cells per whole brain slice, as seen in (**B**). (**D**) Higher resolution images of IBA1+ (green) and PU.1+ (red), a myeloid cell marker, cells. Data are represented as mean ± SEM (n=4–6). # p < 0.1, *p < 0.05, *** p < 0.001.

The online version of this article includes the following source data for figure 4:

**Source data 1.** WM repopulation occurs due to an unprecedented level of microglial depletion.

---

*Ascl1*+ (0% of IBA1 cells were YFP+ during recovery; *Figure 5C*) or *Nestin*+ (0% of IBA1 cells were YFP+ during recovery; *Figure 5D*) cell sources, despite their transient expression of these markers. Consistent with previous reports (*Fonseca et al., 2017*; *Zhao et al., 2019*), we found that the *Cre*-recombinase from the *Cx3cr1^{CreERT2}* line is leaky in the absence of tamoxifen (8% of microglia express YFP in control brains and 12% of microglia express YFP during at 7 day recovery; *Figure 5—figure supplement 2C–D*). Despite this, we show that 94% of microglia in control brains expressed YFP (*Figure 5—figure supplement 2C,B*), 100% of surviving microglia expressed YFP following 14 days of PLX3397 (*Figure 5—figure supplement 2C,B*), and 95% of repopulating microglial cells expressed YFP at 7 day recovery (*Figure 5—figure supplement 2B*), thereby demonstrating that the majority of repopulating cells derive from a *Cx3cr1*+ cell source (*Figure 5E*). These repopulating cells exhibit a similar wave-like migration pattern, appearing first near ventricular areas and lastly in cortical regions (*Figure 5—figure supplement 2G*).

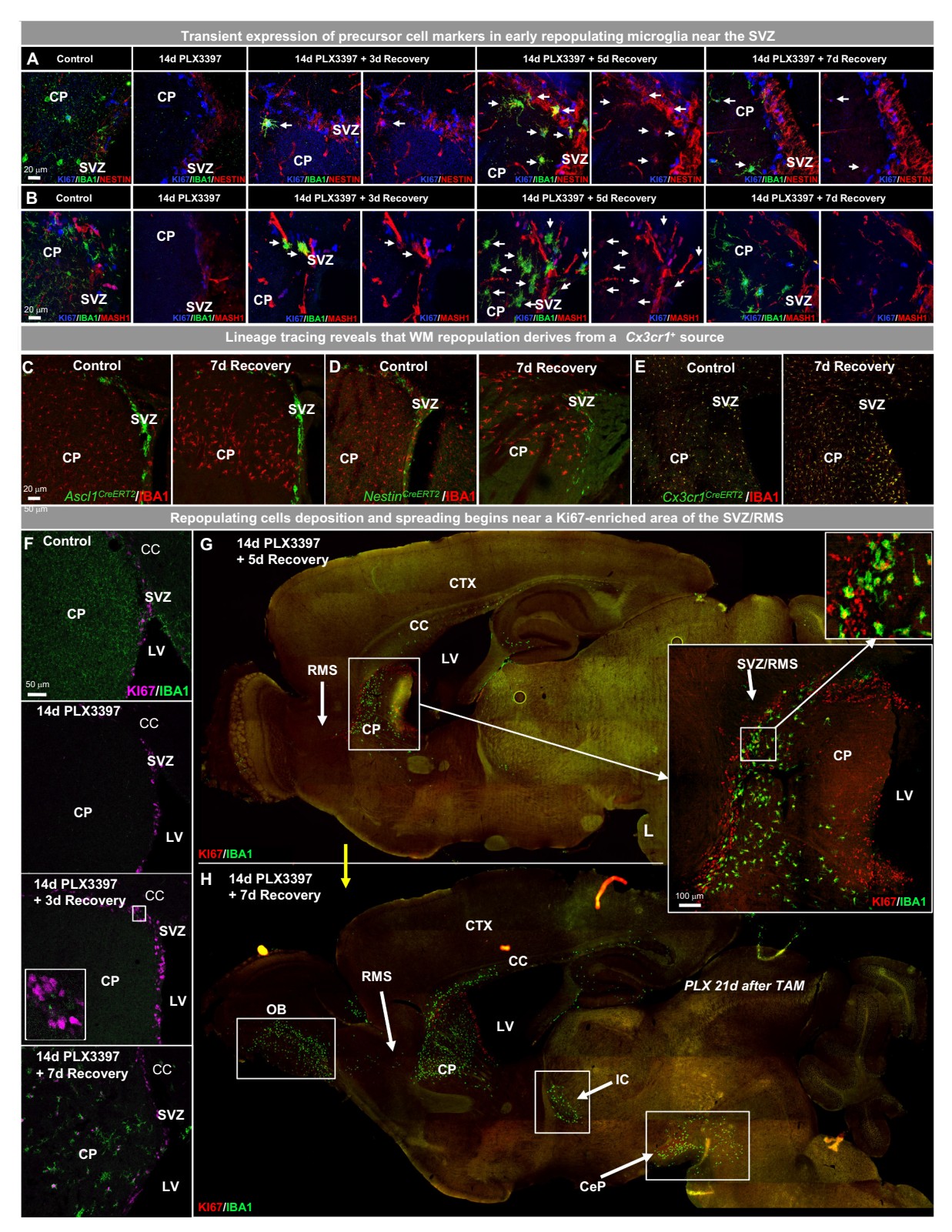

**Figure 5.** WM repopulating myeloid cells derive from an existing *Cx3cr1+* cell source originating from the SVZ/WM area. (A–H) Two-month-old WT mice were treated with PLX3397 (600 ppm) for 14 days, then allowed to recover without PLX3397 for 3, 5, and 7 days. (A–B) Representative 63x immunofluorescence images of proliferating (Ki67+, blue) myeloid cells (IBA1+, green) staining for positive for common cell lineage/precursor cell markers: NESTIN (red, **A**) and MASH1 (red, **B**) in the SVZ of control, 14 day PLX3397, 3 day recovery, 5 day recovery, and 7 day recovery mice. (C–E)
*Figure 5 continued on next page*

Figure 5 continued

CreER-directed lineage-specific labeling. In these mouse lines, tamoxifen-inducible Cre-recombinase is expressed under control of the promoter of interest. When activated by tamoxifen, the CreER fusion protein translocates to the nucleus allowing transient recombination to occur and, when crossed to a YFP reporter, visualization of induced expression via eYFP. (C–D) Representative 20x images of IBA1[+] (red) and associated promoter-driven lineage-derived (YFP, green) cells in control and 7 day recovery Ascl1[CreERT2]/YFP (B) and Nestin[CreERT2]/YFP (C) mice. (E) Representative 20x images of IBA1[+] (red) and Cx3cr1[+] lineage derived (YFP, green) cells in control and 7 day recovery mice. (F–H) Representative coronal (F) and sagittal (G–H) brain images of IBA1[+] (green) and Ki67[+] (red, B–C) cells near SVZ/RMS regions between 5 and 14 days recovery. Inserts provide higher resolution images of cells near the SVZ/RMS proliferative site of repopulation, illustrating the spread of repopulating cells via WM/axonal tracts (i.e. RMS) between the CP and OB and between other WM regions (IC and CeP). CP, caudoputamen; CC, corpus callosum; CTX, cortex; LV, lateral ventricle; RMS, rostral migratory stream; SVZ, subventricular zone; OB, olfactory bulb; IC, internal capsule; CeP, cerebral peduncle.

The online version of this article includes the following source data and figure supplement(s) for figure 5:

**Source data 1.** WM repopulating myeloid cells derive from an existing Cx3cr1+ cell source originating from the SVZ/WM area.
**Figure supplement 1.** Early WM repopulating myeloid cells transiently express some precursor cell markers.
**Figure supplement 2.** WM repopulating myeloid cells derive from an existing *Cx3cr1+* cell source originating from the SVZ/WM area.

## The role of the SVZ/WM area in myeloid cell proliferation and migration signaling during early WM repopulation

In addition to the site of surviving CSF1Ri-resistant microglia, we next explored what role the SVZ/WM area plays in cell proliferation and the spatio-temporal expansion of repopulating cells during WM repopulation. Immunohistochemical analyses for IBA1 and Ki67 show that repopulating cells initially appear between 3 and 7 days recovery, with IBA1[+]/Ki67[+] cells apparent within the SVZ and the adjacent caudoputamen by 7 days recovery (*Figure 5F*). Further evaluation of sagittal sections from mice at 5 day recovery confirms that repopulating cells first populate the parenchyma in the SVZ, specifically from a Ki67[+]-dense region located along the alpha arm of the SVZ (αSVZ) and posterior RMS (pRMS) (*Figure 5G*). These cells subsequently accumulate in the WM areas adjacent to the SVZ (i.e. the corpus callosum), caudoputamen, RMS, olfactory bulb, internal capsule, and cerebral peduncle before eventually filling the parenchymal grey matter (*Figure 5H*). The internal capsule connects the cerebral peduncle, caudoputamen, and RMS, while the RMS connects the SVZ to the olfactory bulb, providing anatomical pathways by which repopulating cells travel to specific brain locations. Of note, the aforementioned niches are the precise locations in which microglia initially colonize the developing brain (*Ueno et al., 2013*; *Verney et al., 2010*; *Ginhoux et al., 2013*). During the 14th – 17th week of gestation in the developing human brain, microglia are found near or within: the optic tract, the WM junction between the thalamus and internal capsule, and the junction between the internal capsule and the cerebral peduncle (*Menassa and Gomez-Nicola, 2018*). Furthermore, a recent study has shown that a second population of amoeboid CX3CR1-expressing microglia emerge from the ventricular zone at embryonic day 20 (E20) and infiltrate the corpus callosum during post-natal day 3–7 (P3-P7) (*Nemes-Baran et al., 2020*), corresponding to a similar postnatal timepoint and location as PAMs/ATMs identified via scRNA-seq (*Hammond et al., 2019*; *Li et al., 2019*).

To examine the transcriptional changes occurring in the SVZ during this early stage of WM repopulation, we micro-dissected the SVZ from control, 14 day PLX3397, and 5 day recovery mice, and performed bulk tissue gene expression analyses via RNA-seq (*Figure 6A*). Gene expression data can be explored at http://rnaseq.mind.uci.edu/green/alt_repop_svz/gene_search.php. In comparing control vs. 5 day recovery, 227 DEGs were identified (FDR < 0.05) between the two groups (*Figure 6B*), with the majority being downregulated microglia-enriched/related genes, reflecting the reduced pool of myeloid cells in the CNS during the early stages of repopulation. Upregulated non-myeloid enriched DEGs in depleted vs. 5 day recovery mice (*Figure 6B*) consisted of genes implicated in cell cycle regulation (*Pak3, Swi5, Psmd11, Stat3*), DNA transcription/recombination/repair/expression (*Alyref2, Swi5, Zfp612, Zfp51, Thumpd1, Prmt5, Taok3, Psmd11, Tox, Stat3*), cell adhesion/migration/proliferation (*Pak3, Anxa1, Cadm1*) and development (*Gfap, Rab14, Zfp612*). Gene ontology (GO) analysis of DEGs between control and 5d recovery SVZ tissue identified the following top four enriched pathways: *myeloid cell differentiation, leukocyte immunity, leukocyte activation, leukocyte chemotaxis and phagocytosis* (*Figure 6C*). Focusing on myeloid genes, *P2ry12, Siglech, Trem2, Cd33* and *Cx3cr1* were least enriched during initial repopulation, whereas *Ccl12, Cd52, Lyz2, Itgb2,* and *Cd84* were highly enriched (*Figure 6D*). To explore the biological relevance of these findings

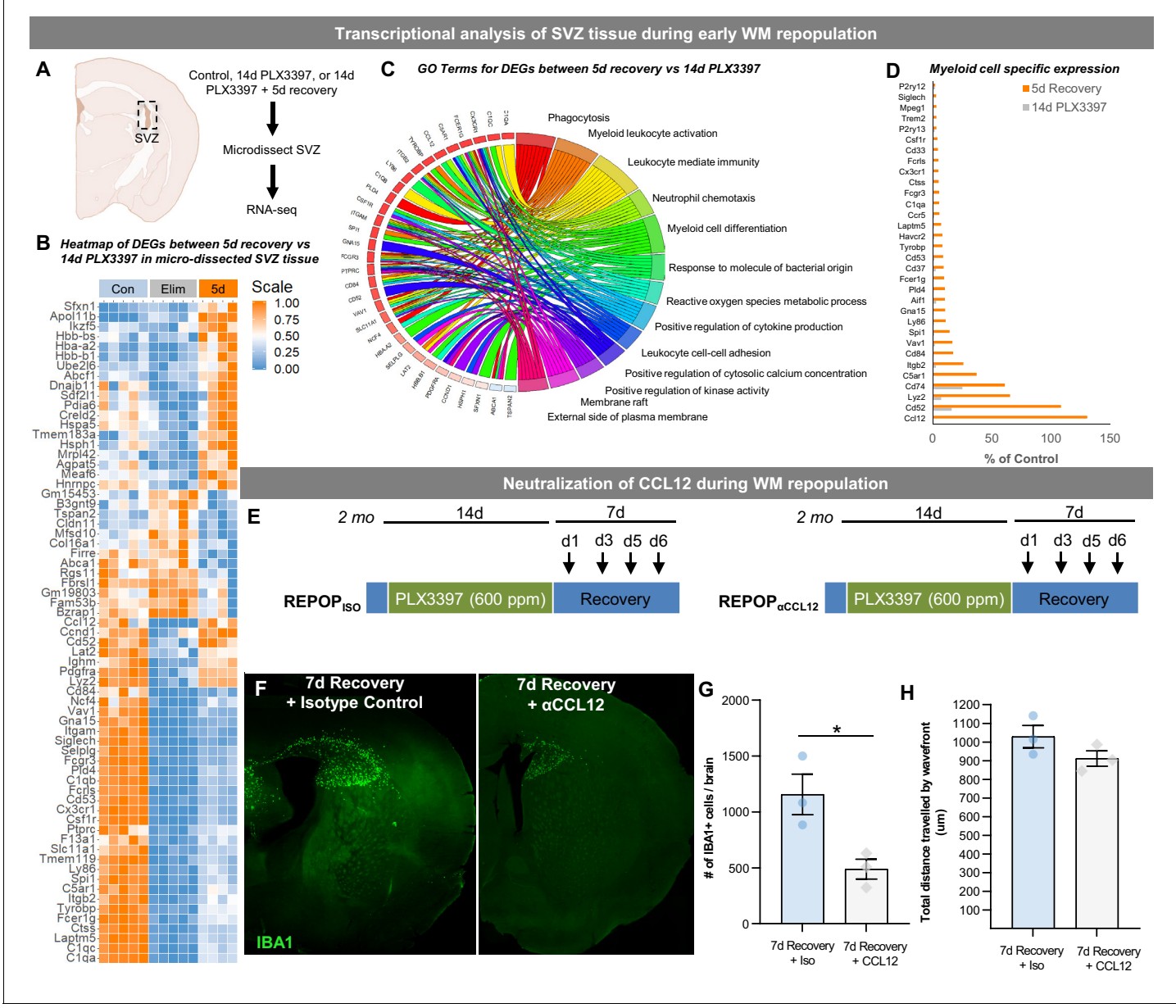

**Figure 6.** The role of the SVZ/WM area in myeloid cell proliferation and migration signaling during early WM repopulation. (A–D) Transcriptional analysis of SVZ tissue during early WM repopulation. (A) Bulk tissue RNA-seq analysis was performed on micro-dissected SVZ tissue from control, 14 day PLX3397, and 5 day recovery brains (n=5). (B) Heatmap of DEGs between 14 day PLX3397 (Elim) and 5 day recovery SVZ tissue. Gene expression data can be explored at http://rnaseq.mind.uci.edu/green/alt_repop_svz/gene_search.php. (C) Gene ontology chord plot of DEGs between control and 5 day SVZ tissue. (D) Plot highlighting expression (% of control) changes in myeloid-associated genes in depleted (14d PLX3397) and repopulated (5d recovery) SVZ tissue. (E–H) Neutralization of CCL12 during WM repopulation. (E) Experiment schematic of CCL12 neutralization study: 2-month-old WT mice were treated for 14 days with PLX3397 (600 ppm) and then placed on control diet for 7 days allowing for WM repopulation. Four i.p. injections were administered of anti-CCL12 antibody or goat IgG (Isotype control) at 1 day recovery, 3 day recovery, 5 day recovery, and 6 day recovery. (F) Representative whole brain images of IBA1[+] cell (green) deposition following treatment. (G–H) Quantification of number of total IBA1[+] cells and total distance traveled by IBA1[+] wavefront in (F). Total distance was calculated by measuring the length from the ventricular edge of SVZ to the leading edge of the IBA1[+] cell wavefront.

The online version of this article includes the following source data for figure 6:

**Source data 1.** The role of the SVZ/WM area in myeloid cell proliferation and migration signaling during early WM repopulation.

and the effect on early repopulation dynamics due to a loss in one of these important genes/signals, we administered an antibody against CCL12, the most highly upregulated gene during early WM repopulation (*Figure 6E*). Here, we show that neutralization of CCL12 results in a significant reduction in repopulating cell numbers at 7 day recovery, but not total distance of cell spreading (*Figure 6F–H*), indicating that this chemokine may play an important role in early repopulating cell proliferation or survival. Together, these data highlight the role of the SVZ and signaling during early WM repopulation.

## WM repopulating cells do not derive from the periphery

Our data show that extensive microglial depletion results in repopulation of the adult brain from myeloid cells that originate from SVZ/WM areas and utilize WM 'highways' to spread throughout the brain before filling the cortex in a distinctive wave-like pattern. As WM repopulating cells maintain distinct phenotypes from microglia, even after extended periods of time in the brain, we concluded that either (1) surviving SVZ/WM myeloid cells represent a distinct myeloid cell type with the capacity to spread and fill the empty microglial brain niche, or (2) extensive microglial depletion stimulates an influx of peripheral cells, which enter near the ventricles and then spread throughout the brain maintaining distinct profiles to their microglial counterparts. Indeed, previous studies have shown that under certain conditions (e.g. in an empty microglial niche) induced by microglial depletion, peripheral myeloid cells can infiltrate and serve as the source for microglial repopulation in the brain parenchyma (*Varvel et al., 2012*; *Lund et al., 2018*; *Cronk et al., 2018*; *Paschalis et al., 2018*). Thus, we reasoned that repopulation could be occurring from peripheral sources and undertook several complementary experimental approaches to explore this.

The choroid plexus is a major route of cellular entry into the CNS (*Ge et al., 2017*), thus we explored the contribution of this site to WM repopulation. Here, we observe that choroid plexus myeloid cells do not repopulate until 14 days recovery, despite the appearance of myeloid cells in the adjacent brain parenchyma (*Figure 7A–B*). Utilizing $Cx3cr1^{GFP/+}/Ccr2^{RFP/+}$ mice, in which CCR2$^+$ cells (mainly monocytes) express RFP, we show that WM repopulating cells are CCR2/RFP$^-$, or not a result of the infiltration of CCR2$^+$ monocytes (*Figure 7—figure supplement 1A–C*). A recent study has posited that CSF1R inhibition suppresses CCR2$^+$ monocyte progenitor cells and CX3CR1$^+$ BM-derived macrophages (among other BM populations) and that these populations do not recover after cessation of CSF1R inhibition (*Lei et al., 2020*). In this study, we evaluated the effects of 14 day PLX3397 600 ppm treatment on peripheral myeloid cells, including CCR2$^+$ and CX3CR1$^+$ myeloid cells, and BM myeloid precursors (*Figure 7—figure supplement 1D–F*). Here, we observe an expansion, not suppression, of HSCs, common myeloid progenitors (CMPs), and granulocyte/monocyte progenitors (GMPs) following CSF1R inhibition and/or recovery (*Figure 7—figure supplement 1F*). However, these changes do not result in significant changes to blood or spleen myeloid cell populations (*Figure 7—figure supplement 1D–E*). CCR2/CCL2 signaling is implicated in many neuropathologies with peripheral cell CNS infiltration (*Chu et al., 2014*), however, we show that CX3CR1 and CCR2 KO in $Cx3cr1^{GFP/GFP}/Ccr2^{RFP/RFP}$ mice elicits no alterations in WM repopulation (*Figure 7C–D*) indicating that this form of repopulation is not dependent on these signaling axes. We further explored the CCR2/CCL2 signaling axis using $Ccl2^{-/-}$ mice. Interestingly, we found that a lack of $Ccl2$ conveyed resistance to CSF1Ri-induced microglial death, with 14 day PLX3397 treatment only eliminating ~90–95% of microglia (*Figure 7—figure supplement 1G–H*). As a result, GLOBAL repopulation was observed in these mice rather than WM repopulation. In combination with findings from H2K-BCL2 mice, these data confirm that >99% microglial elimination is a requirement for WM repopulation, which may not be achieved by 14 day PLX3397 treatment under certain conditions.

Next, we utilized bone marrow (BM) chimeric mice to further determine whether repopulating cells originate from a peripheral source (i.e. non-CCR2+ monocytes or other BM-derived cells). Two-month-old wild-type mice underwent head-shielded (HS) irradiation, followed by retro-orbital administration of GFP$^+$ donor BM cells and 12 weeks of recovery for immune system reconstitution (*Figure 7E–F*). Previous studies have shown that CNS infiltration can occur from the BM upon exposure to head irradiation and consequent BBB permeability (*Eglitis and Mezey, 1997*; *Priller et al., 2001*; *Mildner et al., 2007*). With HS irradiation, however, no GFP$^+$ cells were visible in the parenchyma of control chimeric mice (*Figure 7G,K*), confirming that under normal conditions circulating peripheral cells do not enter the brain. GFP$^+$ cells were visible in the choroid plexus of control HS

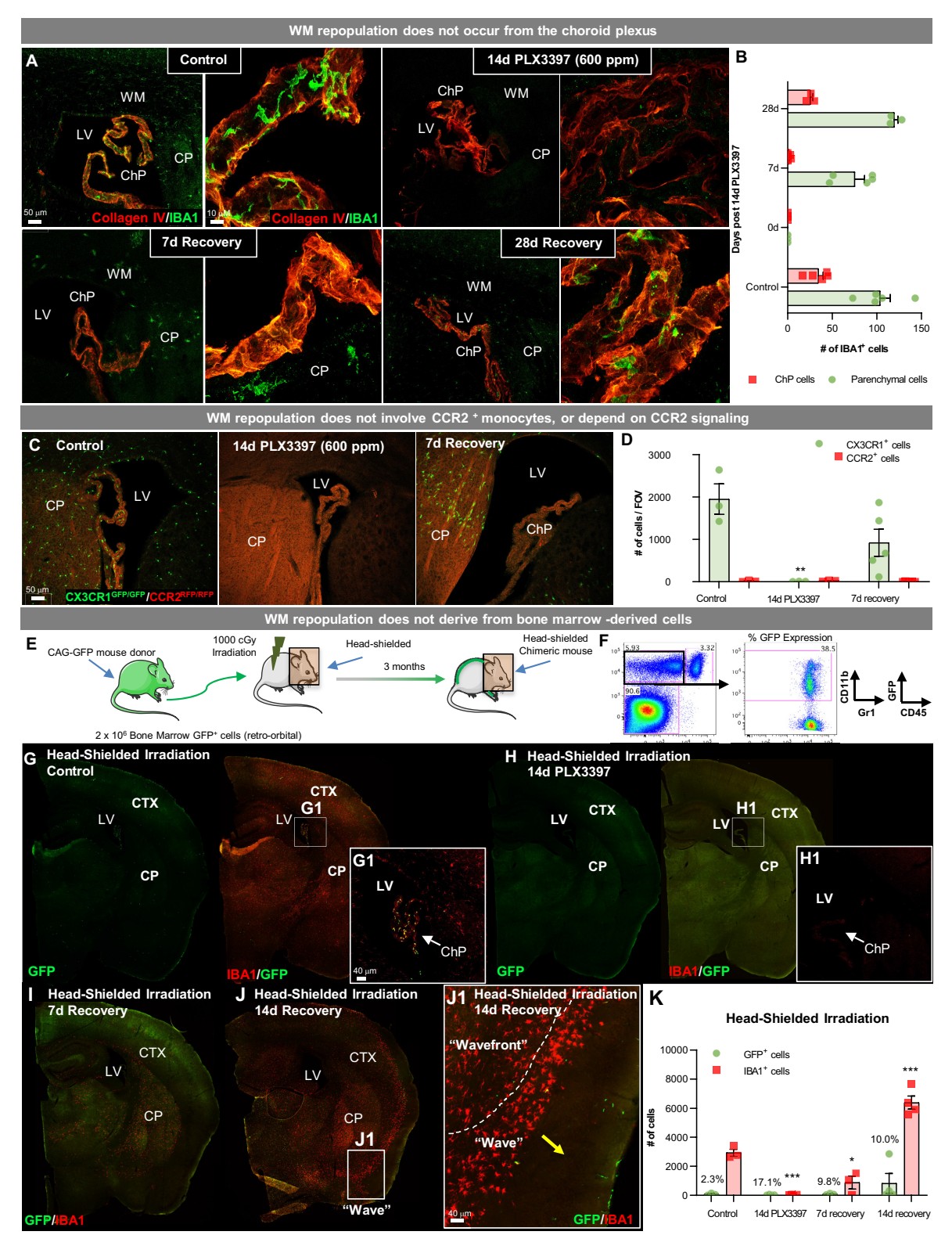

**Figure 7.** WM repopulating cells do not derive from the periphery. (**A**) Representative immunofluorescence 10x (**A**) and 63x (**B**) images of IBA1⁺ cell deposition within the choroid plexus (labeled with Collagen IV) in control, 14 day PLX3397, 7 day recovery, and 28 day recovery mice. (**B**) Quantification of IBA1⁺ cell deposition in the choroid plexus and parenchymal space. (**C, D**) Sustained microglial depletion and WM repopulation in *Cx3cr1*^GFP/GFP^/*Ccr2*^RFP/RFP^ (i.e., *Cx3cr1* and *Ccr2* KO). Representative immunofluorescence images (**C**) and quantification of the number (**D**) of CX3CR1-GFP⁺ (green)

*Figure 7 continued on next page*

Figure 7 continued

and CCR2-RFP$^+$ (red) cells in control, 14 day PLX3397, and 7 day recovery mice (n=3–5). (E) Experimental paradigm: Schematic depicting generation of BM GFP$^+$ chimeras, achieved by head-shielded (HS) irradiation and transplantation of donor GFP$^+$ BM cells. After 3 months, mice were treated for 14 days with PLX3397 and then allowed to recover for 7 or 14 days on control diet. (F) FACS gating strategy to determine % chimerism in BM GFP$^+$ chimeras achieved by head-shielded irradiation. (G–J) Representative whole brain images of GFP$^+$ (green) and IBA1$^+$ (red) cell deposition in HS irradiated control (G), 14 day PLX3397 (H), and mice following 7 (I) and 14 day recovery (J). (J1) Higher resolution images of repopulating cell wavefront seen in HS chimeras during WM repopulation. (K) Quantification of number of GFP$^+$ and IBA1$^+$ cells treated HS irradiated mice. % above bar graph indicates %GFP$^+$IBA1$^+$/IBA1$^+$ cells. Data are represented as mean ± SEM (n=3–7). *p < 0.05, ** p < 0.01, *** p < 0.001. CTX, cortex; CP, caudate putamen; LV, lateral ventricle; ChP, choroid plexus; WM, white matter.

The online version of this article includes the following source data and figure supplement(s) for figure 7:

**Source data 1.** WM repopulating cells do not derive from the periphery.

**Figure supplement 1.** Evaluation of peripheral cell dynamics and CCL2 signaling in WM repopulation.

chimera (*Figure 7G1*), consistent with their partial turnover by circulating BM-derived cells (*Kierdorf et al., 2019*). Fourteen-day PLX3397 eliminated all myeloid cells in HS-irradiated brains (*Figure 7H,K*). By 7 day recovery, WM repopulation was apparent in HS-irradiated chimeric mice, however, IBA1$^+$ cells were GFP$^-$ (*Figure 7I–K*), thus ruling out peripheral BM-derived cells as the source of this form of microglial repopulation. At 14 day recovery, the wave of repopulating myeloid cells was visible in the cortex, as seen in WT mice (*Figure 7J and J1*). It should be noted that since parenchymal repopulating IBA1$^+$ cells were GFP$^-$ in HS chimeric mice, these data provide strong evidence that BM-derived cells, including choroid plexus macrophages - which are GFP$^+$ in HS chimeric control mice - do not contribute to WM repopulation. Together, these data indicate that surviving SVZ/WM myeloid cells serve as the major source of this unique form of CNS myeloid cell repopulation.

## Repopulating myeloid cells are transcriptionally distinct and mount a differential response to inflammatory stimulus compared to homeostatic microglia

We next sought to gain insight into the transcriptional profile of WM repopulating cells once in residence in the brain. In addition, we also investigated whether phenotypic and transcriptional alterations translated into functional consequences and thus explored their response to immune challenge, via LPS administration (*Figure 8A*). Here, we performed RNA sequencing (RNAseq) on FACS-sorted CD11b$^+$CD45$^{int}$ control and 28 day recovery cells at 6 and 24 hr following LPS-induced immune challenge (*Figure 8A*, *Figure 8—figure supplement 1A*). For controls, cells were also collected from mice that did not receive LPS, referred to as 0 hr post LPS. Flow cytometry analysis of CD11b$^+$CD45$^{int}$ cells collected at 28 day recovery shows that these repopulating cells exhibit higher CD45 and CD11b intensities compared to control microglia at baseline. Control microglia show increased CD45 and CD11b in response to LPS, while repopulated cells do not change (*Figure 8—figure supplement 1B–C*).

RNA was extracted from FACS-sorted CD11b$^+$CD45$^{int}$ cells and RNA-seq analysis was performed to establish a high-resolution transcriptome profile of these cells in the absence and presence of LPS. Gene expression data can be explored at http://rnaseq.mind.uci.edu/green/alt_repop_lps/gene_search.php. Unlike GLOBAL repopulation, in which control and repopulating cells display few transcriptional differences (25 DEGs in GLOBAL vs. control *Figure 8—figure supplement 1F–G*), we identified 69 DEGs in WM repopulated myeloid cells compared to control microglia in the absence of LPS (logFC > 1, FDR > 0.05; *Figure 8B*). GO analysis of DEGs between control and 28 day repopulated cells identified the following top five enriched pathways: *regulation of cellular component size, negative regulator of chemotaxis, ameboidal-type cell migration, negative regulation of response to external stimulus*, and *wound healing* (*Figure 8C*). To visualize differences in these cells vs. other myeloid cell subsets, we compared their gene expression profile to previously established myeloid cell signatures, including homeostatic microglia (*Butovsky and Weiner, 2018*; *Bennett et al., 2016*), HSC/BM-derived myeloid cells (*Weiskopf et al., 2016*), border-associated myeloid cells (BAMs) (*Mrdjen et al., 2018*)/CNS-associated macrophages (CAMs) (*Jordão et al., 2019*), DAMs/MGnD/ARMs (*Keren-Shaul et al., 2017*; *Krasemann et al., 2017*; *Mathys et al., 2017*; *Sala Frigerio et al., 2019*), PAMs/ATMs (*Masuda et al., 2019*; *Hammond et al., 2019*;

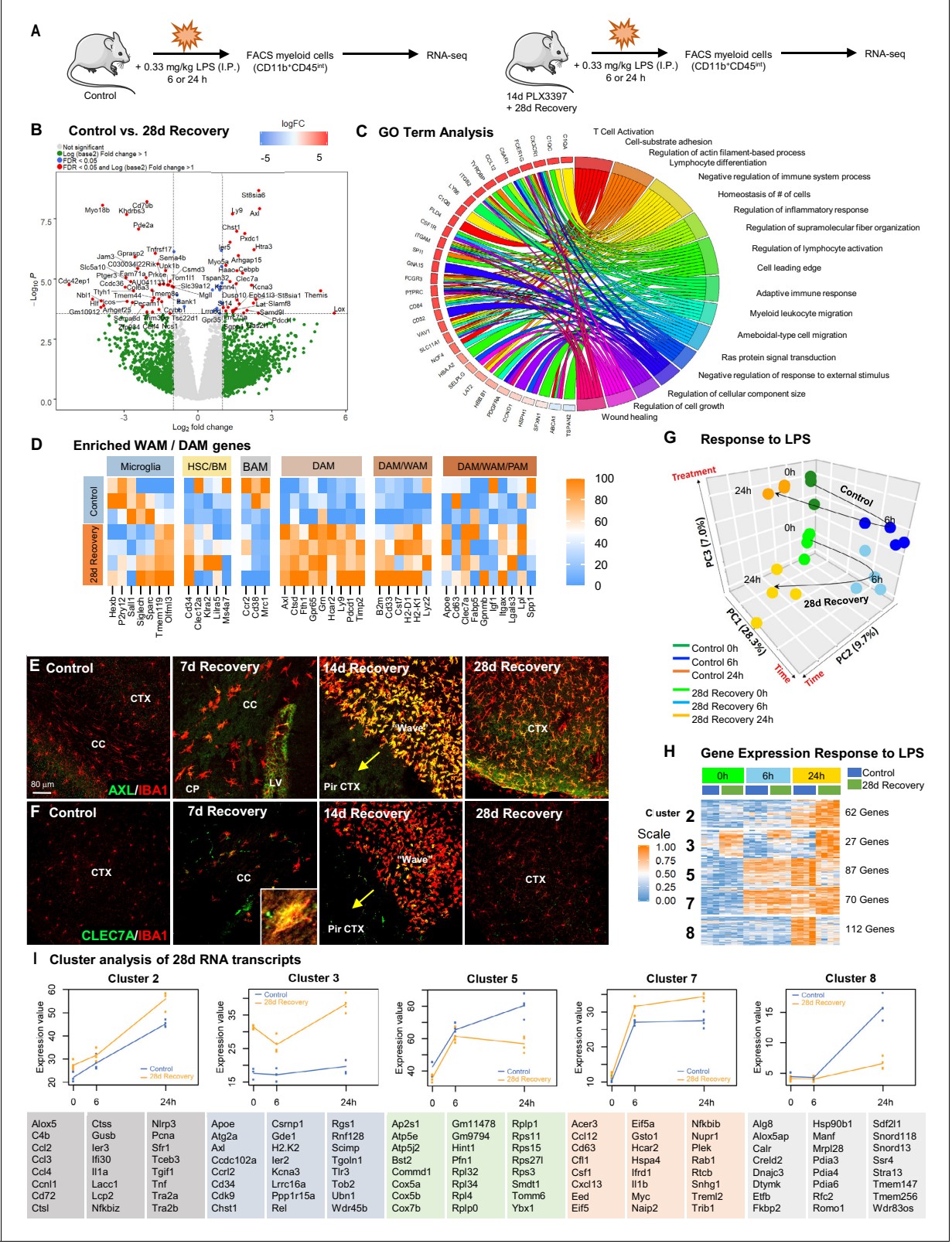

**Figure 8.** Repopulating myeloid cells are transcriptionally distinct and mount a differential response to inflammatory stimulus compared to homeostatic microglia. (A) Two-month-old WT control or 28 day recovery mice were given intraperitoneal injections of either PBS or LPS (0.33 mg/kg) and then collected at 6 or 24 hr post injection. Controls, which were mice that did not receive LPS, are referred to as 0 hr post LPS. Myeloid cells were extracted from whole brain hemispheres, isolated using FACS gating for CD11b+CD45int and processed for RNA-seq. (B) Volcano plots displaying the fold

*Figure 8 continued on next page*

*Figure 8 continued*

change of genes (log2 scale) and their significance (y axis, -log10 scale) between control vs. 28 day recovery mice. (C) Gene ontology chord plot of DEGs between control and 28 day recovery myeloid cells. (D) Heatmap showing expression of genes enriched in DAM, HSC/BM-derived cells, canonical microglia, BAM, PAM, and WAM signatures in control and 28 day repopulating myeloid cells. (E–F) Representative immunofluorescence 20x images of IBA1$^+$ (red) and AXL$^+$ (green, I) or CLEC7A$^+$ (green, J) cells shown in areas with high repopulating cell deposition in control, 7, 14, and 28 day recovery mice. (G) Principal component analysis plot of extracted control and 28 day recovery cells, across time (0 hr, 6 hr, 24 hr) and treatment (+/- LPS), depicting the separation of groups into six clusters. (H) Heatmap of selected time-series cluster analysis of control and 28 day recovery cells. Provided number indicates number of genes per cluster. (I) Time-series cluster analysis of control vs. 28 day recovery myeloid cell response (during WM repopulation) to LPS challenge following 14 day PLX3397 (600 ppm in chow; from H). Clusters showing distinct responses to LPS between control and 28 day WM repopulated cells, across time, were plotted as eigengene values, along with the top represented genes within each cluster. Data are represented as mean ± SEM (n=3–5). *p < 0.05, ** p < 0.01, *** p < 0.001. CP, caudoputamen; CC, corpus callosum; CTX, cortex; LV, lateral ventricle; PirCTX, piriform cortex. Gene expression data can be explored at http://rnaseq.mind.uci.edu/green/alt_repop_lps/gene_search.php.

The online version of this article includes the following source data and figure supplement(s) for figure 8:

**Source data 1.** Repopulating myeloid cells are transcriptionally distinct and mount a differential response to inflammatory stimulus compared to homeostatic microglia.

**Figure supplement 1.** WM repopulating myeloid cells are transcriptionally distinct from homeostatic microglia.

*Li et al., 2019*), and WAMs (*Safaiyan et al., 2021*). Notably, we observe robust enrichment of DAM and WAM-associated genes in repopulating cells, including *Clec7a, Axl, Apoe, Cst7, Ctsd,* and *Ly9* (*Figure 8D*). AXL and CLEC7A, recently identified as WAM markers (*Safaiyan et al., 2021*), immunostaining is apparent in repopulating myeloid cells, particularly in the early stages of repopulation, and in cells located at the wavefront, whereas undetected in microglia from control brains (*Figure 8E–F*). In addition, we also observe a reduction in expression of *Sall1*, a transcription factor unique to microglia (*Buttgereit et al., 2016*), in repopulating compared to homeostatic microglia.

After confirming distinct transcriptome differences between control and 28 day recovery isolated microglia at 0 hr LPS (i.e. in the absence of LPS), we next evaluated their gene expression profiles at 6 hr and 24 hr following LPS challenge. Principal component analysis demonstrated that biological replicates were highly correlated, with samples clustering into six distinct groups (*Figure 8G*). To identify global patterns in gene expression changes over time between our experimental groups, we employed K-means clustering (*Conesa et al., 2006*), revealing nine distinct clusters of genes (*Figure 8H*, *Figure 8—figure supplement 1H*). Cluster 3 contains genes that are significantly different between control and repopulating cells across all time points, including genes implicated in clearance (*Apoe, Atg2a, Axl, Wdr45b*), cell growth/differentiation (*Cd34, Cdk9*), stress (*Ier2, Ppp1r15a*), inflammation (*Ccrl2, H2-K2, Scimp, Tlr3*), and senescence (*Ubn1, Wdr45b*). Clusters 2 (e.g. *C4b, Ccl2, Ccl3, Ccl4, Ctsl, Ctss, Il1a, Nlrp3, Pcna,* and *Tnf*), 5, 7 (e.g. *Ccl12, Cd63, Csf1, Cxcl13, Il1b, Myc, Plek,* and *Nfkbib*), and 8 shows differential responses to LPS at the 24 hr timepoint between control and repopulating cells (*Figure 8I*). GO term analysis of Cluster 3 revealed top enriched pathways: *phagopore assembly site membrane, extrinsic component of membrane,* and *endosome*. Cluster 8 GO term analysis revealed the following enriched pathways: *negative regulation of glucocorticoid secretion, negative regulation of interleukin-1 mediated signaling pathway,* and *connective tissue replacement involved in inflammatory response wound healing*. Together, these findings provide evidence that WM repopulating cells represent a myeloid cell population that are transcriptionally and functionally distinct from adult homeostatic microglia with similarities to the recently identified subset of WAMs, cells absent in the adult (4 months) mouse brain, but comprise 20% of microglia in the aged (24 months) mouse brain (*Safaiyan et al., 2021*). Further studies are needed to evaluate whether these WM repopulating cells exist in the naïve brain as WAMs or whether CSF1Ri treatment induces this phenotype. Further exploration is also warranted on the contributions and possible cross talk with astrocytes and other cell types during repopulation.

## Repopulating myeloid cells elicit few functional differences in behavior, injury, and neuronal-associated structures

Given the transcriptional and functional distinction between WM and homeostatic microglia, we next sought to determine the physiological and functional consequences of filling the adult brain with these cells. Previous studies have shown that replacement of endogenous microglia with GLOBAL repopulating microglia results in no detectable changes in cognitive or behavioral function

(*Elmore et al., 2018*; *Rice et al., 2017*). Here, we performed a battery of cognitive and behavior tasks in 28 day recovery mice (*Figure 9A–H*) and observe that these mice exhibit reduced locomotion (*Figure 9A–B*) and reduced ability to discriminate a novel place change compared to controls (*Figure 9H*). However, we observe no other significant behavior or cognitive disruptions. Overall, these animals do not appear to exhibit overt cognitive, behavioral, or health changes.

Based on transcriptional changes in myeloid signaling and priming, we next explored how these repopulating cells would respond to injury. Live two-photon imaging of cortical myeloid cells was performed in control and 35 day recovery Cx3cr1$^{GFP/+}$/Ccr2$^{RFP/+}$ mice (*Figure 9I–L*). Cx3cr1-GFP$^+$ repopulating myeloid cells react to a focal laser injury with similar rates of migration as homeostatic microglia (*Figure 9J*, *Video 1* and *Video 2*). Furthermore, extension and retraction of processes (i.e. motility) were similar between control microglia and repopulating cells, although repopulating cells displayed fewer and thicker processes as previously demonstrated (*Figure 9K–L*).

Our recent studies have identified a novel contribution of microglia in modulating perineuronal nets (PNNs) in the adult brain (*Crapser et al., 2020b*; *Crapser et al., 2020a*; *Liu et al., 2021*), specialized extracellular matrix assemblies that enwrap neurons and proximal dendrites to regulate synaptic plasticity (*Pizzorusso et al., 2002*), protect against neurotoxins (*Cabungcal et al., 2013*), and enhance signal propagation, among other functions (reviewed in *Crapser et al., 2021*). In control brains, PNNs (as detected by *Wisteria floribunda* agglutinin (WFA) staining) are preferentially found on parvalbumin (PV)-expressing cells. Fourteen-day treatment of high-dose PLX3397 results in significant elevation in PNN staining, corroborating previous findings that microglia play a critical homeostatic role in modulating these structures (*Crapser et al., 2020b*). Following CSF1Ri withdrawal and subsequent 28 day recovery, PNN staining returns to normal levels (*Figure 9M–N*), indicating that WM repopulating cells share similar PNN-regulating capacities as homeostatic microglia.

Accumulating evidence indicates that WM and developing microglia contribute to myelinogenesis/oligodendrocyte progenitor maintenance (*Marsters et al., 2020*; *Hagemeyer et al., 2017*). In early postnatal development, a population of amoeboid microglia migrating from the ventricular zone into the corpus callosum phagocytose oligodendrocyte precursor cells (OPCs) prior to myelination (*Nemes-Baran et al., 2020*), which share similar timepoint and location-specific distributional overlaps with PAMs/ATMs (*Hammond et al., 2019*; *Li et al., 2019*). Thus, we next explored the effects of filling the brain with repopulating cells on WM and WM-associated cells. However, unlike during development in which these cells participate in oligo/myelinogenesis, in 28 day recovery mice we observe no changes in Olig2, a marker for cells of oligodendrocyte lineage (*Figure 9—figure supplement 1A,D*), or PDGFRα, a marker for oligodendrocyte progenitor cells (*Figure 9—figure supplement 1B,E*). Given the similarity of WM cells to WAMs, which have been shown to engulf MBP$^+$ particles/myelin debris, we next stained for MBP, a marker for myelin (*Figure 9—figure supplement 1C,F–G*). Although not statistically significant there appears to be a slight decrease in MBP staining, indicating that these cells could be involved in MBP clearance. Together, these data provide evidence that although WM repopulating cells maintain an altered phenotypic profile, they can still fill the empty microglial niche with few functional consequences.

## Lasting phenotypic and transcriptional profiles of WM repopulating cells after three mo recovery

To determine whether the phenotypic and transcriptional differences in WM repopulating microglia are a result of a slower restoration of the microglial phenotype (i.e. temporary) or sustained after longer recovery time (i.e. reflective of a distinct cell subtype), we treated 2-month-old wild-type mice for 14 days with PLX3397 and then allowed for 3 months of recovery (*Figure 10A–B*). Morphological analysis of IBA1$^+$ cells shows that WM repopulating cells maintain reduced cell density (*Figure 10C*), and amoeboid-like cell characteristics, including larger cell body size (*Figure 10D*), reduced number of processes (*Figure 10E*), and reduced process length (*Figure 10F*). Bulk tissue RNAseq analysis (*Figure 10H*) highlights that transcriptional differences remain in the brain three mo following CSF1Ri withdrawal (*Figure 10I*). Similar to earlier WM repopulation timepoints, DEG analysis highlights a downregulation in homeostatic-associated microglial gene *P2ry12* and upregulation in DAM/WAM-associated genes *Clec7a* (*Figure 10I–J*). GO term analysis shows that upregulated DEGs are involved in *neuron projection development, axon guidance, synaptic transmission,* and *synaptic potential,* providing evidence that WM repopulating cells could play an important role in axonal

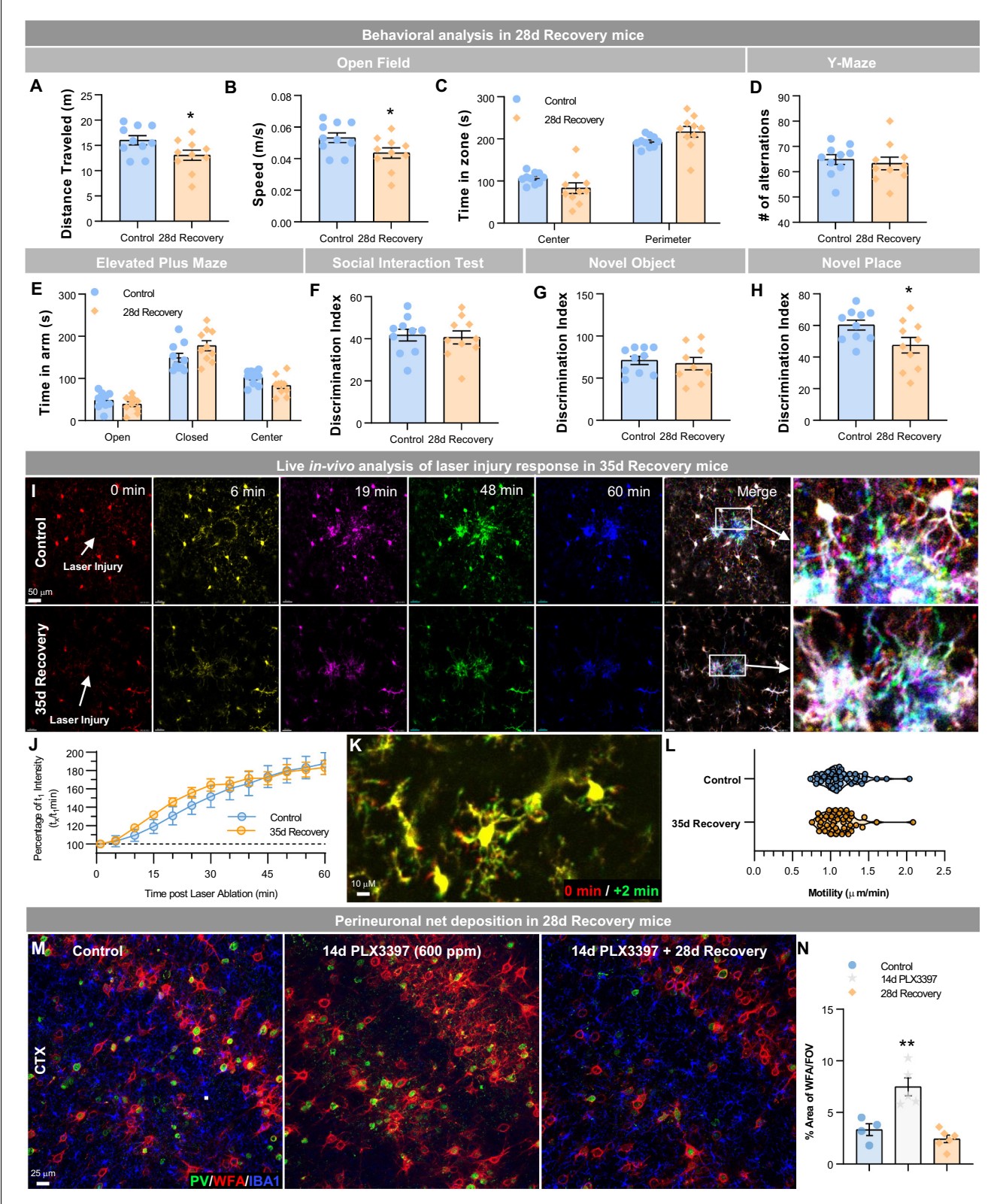

**Figure 9.** Repopulating myeloid cells elicit few functional differences in behavior, injury, and neuronal-associated structures. (A–H) Two-month-old WT mice treated with vehicle or PLX3397 (600 ppm in chow) for 14 days, followed by 28 days of recovery, filling the brain with WM repopulated myeloid cells. Mice underwent behavioral assessment by Open Field, Y-Maze, and Elevated Plus Maze, Social Interaction Test, and Novel Object/Place recognition. In Open Field, distance traveled (A) and average speed (B) were reduced in 28 day recovery groups, but time spent in each zone (C) was

*Figure 9 continued on next page*

*Figure 9 continued*

unchanged. No changes in performance were seen in the Y-maze, as measured by the number of alternations (D). No changes in anxiety behaviors were seen in the elevated plus maze, as measured by time spent in the open or closed arms (E). No changes in social preference were seen in the social interaction test (F). No changes in novel object recognition memory were seen (G), but 28 day recovery mice had a significant impairment in novel place recognition memory (H). Data are represented as mean ± SEM (n=10). *p < 0.05. (I–L) Two-month-old Cx3cr1-GFP[+] mice treated with vehicle or PLX3397 (600 ppm in chow) for 14 days, followed by 35 days of recovery, filling the brain with WM repopulated myeloid cells. (I) Analysis of motility and focal laser response in WM repopulating myeloid cells. Representative images of Cx3cr1-GFP[+] myeloid cell response to laser ablation, over time, obtained via two-photon imaging in control and 35 day recovery mice. (J) Quantification of the average normalized GFP[+] intensity measured within a 50 μm radius of the site of damage over time. (K) Representative image of Cx3cr1-GFP[+] myeloid cell at process motility from 0 min (red) to 2 min (green) time-period. (L) Quantification of process motility (i.e. extension of process μm per min) measuring the difference in visibly moving processes over 2 min in control and 28 day recovery myeloid cells. Data are represented as mean ± SEM (n=4). (M–N) Two-month-old WT mice treated with vehicle or PLX3397 (600 ppm in chow) for 14 days, followed by 28 days of recovery, filling the brain with WM repopulated myeloid cells. (M) Representative immunofluorescence 20x images of Parvalbumin[+] (PV, green), WFA (a marker for perineuronal nets, red), and IBA1[+] (blue) cells shown in the cortex of control, 14 day PLX3397, and 28 day recovery mice. (N) Quantification of % area of WFA per field of view (FOV) in the cortex. Data are represented as mean ± SEM (n=4–6). ** p < 0.01.

The online version of this article includes the following source data and figure supplement(s) for figure 9:

**Source data 1.** Repopulating myeloid cells elicit few functional differences in behavior, injury, and neuronal-associated structures.
**Figure supplement 1.** Effects of WM repopulation on white matter and cells of oligodendrocyte lineage.

---

homeostasis. Together, these data show that WM repopulating microglia represent a distinct population of myeloid cells rather than a slower restoration of the microglial phenotype.

## Discussion

As central players in CNS homeostasis, defense, and disease, intense focus has recently been placed on microglia and our understanding of their cell origins, function and dynamics. For decades the identity and ontogeny of microglial precursors remained controversial; the scientific community debated whether microglia derive from embryonic progenitors or blood-derived monocytes (*Rio-Hortega, 1939*; *Ginhoux and Prinz, 2015*). It is now well-established that microglia arise from yolk sac-derived erythromyeloid progenitors (*Ginhoux et al., 2010*; *Kierdorf et al., 2013*).

We previously reported that adult microglia are dependent on CSF1R signaling for their survival and identified several CSF1R inhibitors that eliminate microglia for extended periods of time without peripheral cell infiltration (*Elmore et al., 2014*). Following CSF1Ri-dependent microglial depletion, we and others have shown that subsequent withdrawal of CSF1R inhibitors from the microglial-depleted brain results in rapid microglial repopulation derived from surviving microglia (*Elmore et al., 2014*; *Zhan et al., 2019*; *Najafi et al., 2018*; *Elmore et al., 2018*; *Rice et al., 2017*; *O'Neil et al., 2018*). The resultant tissue is reconstituted within 14–21 days in a homogenous, tile-like fashion with the replacing microglia fully resembling the original tissue (*Elmore et al., 2014*; *Zhan et al., 2019*; *Elmore et al., 2015*; *Elmore et al., 2018*). Due to the rapid proliferation of these surviving microglia, exploration into the contribution of specific myeloid cell subtypes to the adult CNS has proven difficult, and so we set out to

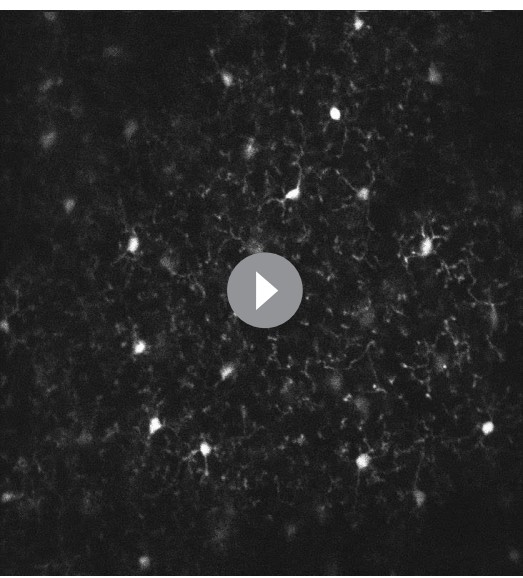

**Video 1.** Cx3cr1-GFP+ myeloid cell response in control mice after laser ablation. Representative video of Cx3cr1-GFP[+] myeloid cell response to laser ablation (1–5 s long) captured over a 62 min time period in control mice obtained via two-photon imaging. Each frame captures 30 s of elapsed time.
https://elifesciences.org/articles/66738#video1

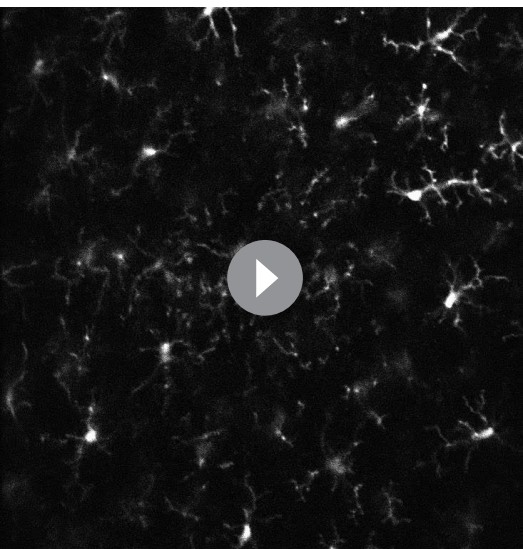

**Video 2.** Cx3cr1-GFP+ myeloid cell response in 35 day recovery mice after laser ablation. Representative video of Cx3cr1-GFP⁺ myeloid cell response to laser ablation (1–5 s long) captured over a 62-min time period in 35 day recovery mice obtained via two-photon imaging. Each frame captures 30 s of elapsed time. https://elifesciences.org/articles/66738#video2

develop a paradigm without notable surviving microglia.

Previous attempts to achieve an empty microglial niche have fallen short, reporting ~95% or less microglial depletion efficiency (*Lund et al., 2018*). Here, we utilize sustained high-dose CSF1Ri administration (specifically 14 days of PLX3397 600 ppm) and show we can obtain 99.98% microglial depletion. In doing so, we identify a CNS myeloid cell subset that repopulates the brain parenchyma from SVZ/WM areas, without contributions from the periphery. We describe this form of repopulation as WM repopulation due to its unprecedented level of depletion efficiency and distinct characteristics from global microglial (GLOBAL) repopulation. Unlike GLOBAL repopulation paradigms, in which surviving microglia proliferate in clusters to give rise to new microglia (*Bruttger et al., 2015*) or more uniformly throughout the brain (*Elmore et al., 2014*; *Zhan et al., 2019*), WM repopulation dynamics involve specific spatiotemporal patterns and a dynamic migratory wave of proliferating cells. In WM repopulation, SVZ/WM microglia give rise to the majority of repopulating cells, including cortical microglia; however, this is not a unique property of WM microglia.

Under these conditions, sufficient depletion of local microglia favors renewal from more distal cells that proliferate and migrate to fill the niche over local repopulation. Here, and in previous studies we have shown that incomplete (≤99%) microglial elimination leads to repopulation (i.e. GLOBAL repopulation) from all surviving microglia, including cortical microglia (*Elmore et al., 2014*), suggesting that there is a specific threshold of surviving microglia necessary for local repopulation. In addition, previous studies report a major contribution of peripheral BM-derived myeloid cells to the repopulating cell population following an empty microglial niche or the persistent loss of microglia (in which microglia cannot repopulate the niche) (*Lund et al., 2018*; *Cronk et al., 2018*). Since these models rely on tamoxifen administration, it could be possible that BM-derived myeloid cell engraftment in the CNS results from an experimental or technical caveat related to toxin administration rather than the presence of an empty microglial niche. In line with this, a recent study has reported that tamoxifen expands macrophage populations and should be reconsidered as a neutral agonist in myeloid cell lineage studies (*Rojo et al., 2018*). Recent scrutiny has also been placed on the use of CSF1R inhibitors, implicating long-term changes in BM-derived macrophages (*Lei et al., 2020*). In this study, we show that high dose and sustained CSF1Ri treatment can result in alterations to monocyte precursor populations; however, these changes do not translate in significant changes to peripheral monocyte populations, which we again show do not contribute to CNS myeloid cell repopulation in the absence of toxin, irradiation or injury.

WM repopulating cells initially appear in specific neuroanatomical niches (first in the SVZ – the site of where surviving CSF1Ri-resistant SVZ/WM myeloid cells reside) and spread throughout the brain in a distinct pattern: via WM tracts to the caudoputamen, optical tract, internal capsule, cerebral peduncle, and finally to cortical areas. The caudoputamen is closely associated with the lateral ventricle, SVZ and WM tracts in which myeloid cells initially appear and migrate, and we believe this spatial association plays a large role in why certain parenchymal areas see more appreciable repopulating cell deposition. Notably, this distribution of initially repopulating cells in ventricular regions and subsequent migration pattern exhibit strong anatomical parallels to microglial colonization and distribution of the embryonic and postnatal brain, in which microglia enter the brain via ventricular routes and remain restricted in WM zones before migrating out to the rest of the brain, with the cortical plate being one of the last areas of colonization (*Tay et al., 2017*; *Ueno et al., 2013*;

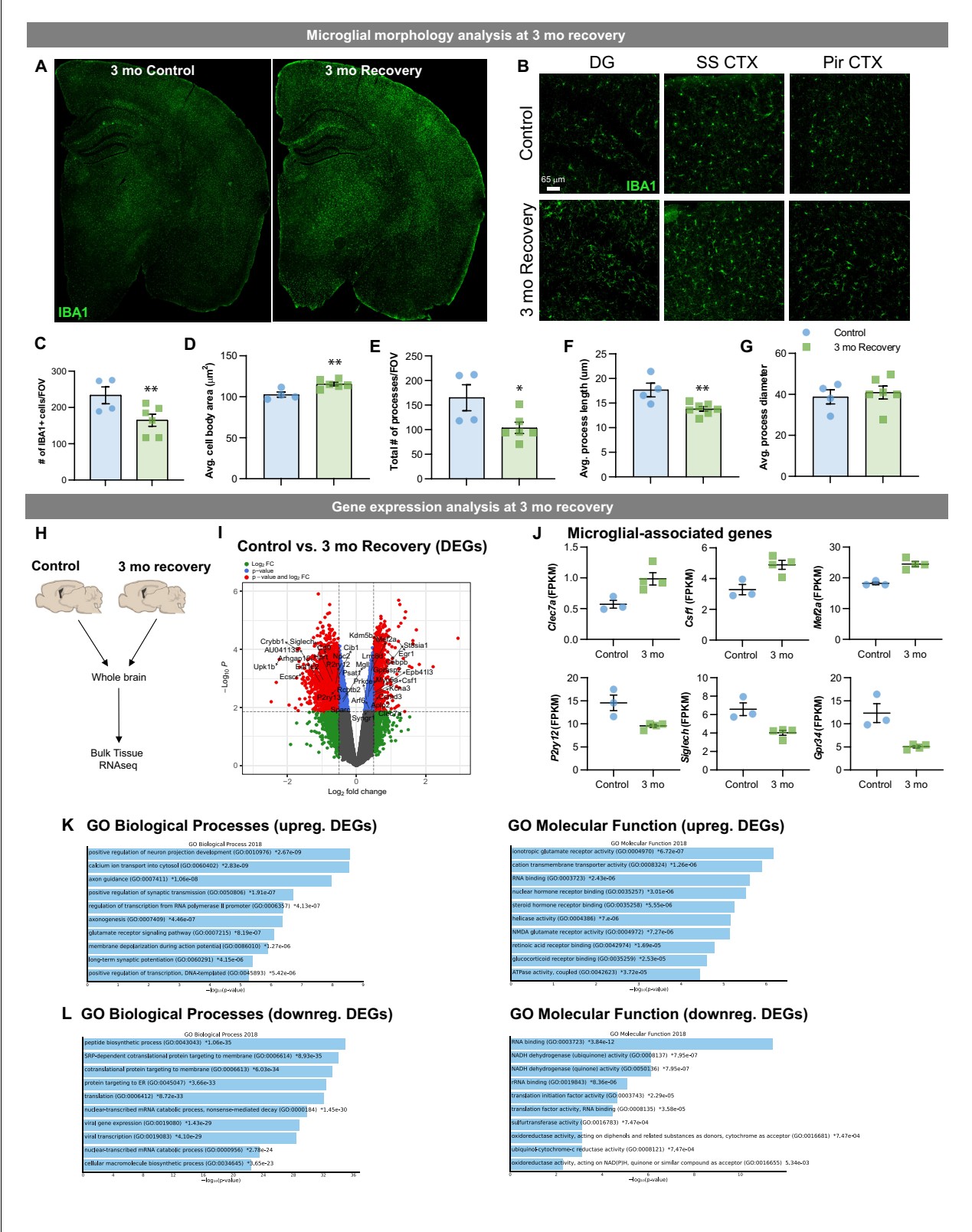

**Figure 10.** Lasting phenotypic and transcriptional profiles of WM repopulating cells after 3 month recovery. Two-month-old WT mice treated with vehicle or PLX3397 (600 ppm in chow) for 14 days. Chow was then withdrawn, and animals were allowed to recover on control diet for three months, assessing the long-term effects of filling the brain with WM repopulated myeloid cells. (A-B) Representative whole brain slice and 20x immunofluorescence confocal images of IBA1+ cells in control and 3-month recovery mice. (C-G) Quantification of IBA1+ cells and morphology: IBA1+

*Figure 10 continued on next page*

*Figure 10 continued*

cells per field of view (C), cell body area (D), total number of IBA1$^+$ processes (E), average process/filament length (F), and average process/filament diameter (G). (H-L) Transcriptional analysis of 3-month recovery WM repopulation brain tissue. (A) Bulk tissue RNA-seq analysis was performed on whole brain hemispheres from control and 3-month recovery brains. (I) Volcano plot displaying the fold change of DEGs (log2 scale) and their significance (y axis, -log10 scale) between control vs. 3 month recovery. (J) Quantification of relative expression (FPKM, fragments per kilobase of transcript per million) for microglial-associated genes in control and 3 month recovery brain tissue. (K-L) Gene ontology analysis of upregulated (K) and downregulated (L) DEGs between control and 3 month recovery mice. Data are represented as mean ± SEM (n=3-4). * p < 0.05, ** p < 0.01. DG, dentate gyrus; SS CTX, somatosensory cortex; Pir CTX, piriform cortex.

The online version of this article includes the following source data for figure 10:

**Source data 1.** Lasting phenotypic and transcriptional profiles of WM repopulating cells after 3-month recovery.

*Verney et al., 2010*; *Ginhoux et al., 2013*; *Xavier et al., 2014*; *Lopez-Atalaya et al., 2018*). Del Rio-Hortega observed the accumulation of microglia along ventricles and white matter areas in the developing brain within the first postnatal week (*Rio-Hortega, 1939*; *Rio-Hortega, 1932*). In 1939, Kershman used the term 'fountain of microglia' referring to 'hot spots' of activated microglia during human embryogenesis. This observation has been confirmed by several studies over the decades, and recent publications have identified a distinct microglia population that appears postnatally in myelinated regions (*Hagemeyer et al., 2017*; *Benmamar-Badel et al., 2020*; *Wlodarczyk et al., 2017*). A recent study reported that the fountain of microglia (amoeboid microglia migrating from the ventricular zone) phagocytoses OPCs in the corpus callosum during development (*Nemes-Baran et al., 2020*). It should be noted that we do not believe WM repopulating cells represent a microglial precursor. Most importantly, our findings highlight important anatomical structures that facilitate microglial/myeloid cell migration in an empty microglial niche, which play an essential role in development, but also appear intact in the adult brain.

Utilizing a microglial depletion and repopulation paradigm that successfully achieves an empty microglial niche, we identified a unique subset of myeloid cells in the SVZ/WM area that appears to serve as the source for WM repopulating cells. Notably, our novel paradigm results in the replacement of microglia with *Cx3cr1*$^+$ myeloid cells originating from the SVZ and associated WM areas, allowing us to study the biology of these cells, and how they adapt to extrinsic environmental cues from grey matter, rather than the WM areas they are normally restricted.

SVZ/WM myeloid cells are initially resistant to CSF1R inhibition, which we believe owes to unique properties of the SVZ environment, as once WM repopulating cells (deriving from SVZ/WM myeloid cells) take up residence in other parenchymal areas, they once again are susceptible to CSF1Ri treatment. Although the signals responsible for shaping macrophage/microglia identity are still being discovered, studies have revealed that the local niche or microenvironment can play an active role in establishing macrophage identity (*Lavin et al., 2014*; *Gosselin et al., 2014*). We postulate that a combination of local factors, such as transcription factors, availability of CSF1, and epigenetic regulation (via tissue-specific enhancers) (*Blériot et al., 2020*), contribute to the uniqueness of SVZ/WM microglia and their ability to escape initial PLX3397 treatment. Studies indicate that WAMs/DAMs upregulate CSF1 and TREM2 (*Keren-Shaul et al., 2017*; *Safaiyan et al., 2021*), signaling factors that promote microglial survival (*Elmore et al., 2014*; *Wang et al., 2015*; *Zheng et al., 2017*; *Easley-Neal et al., 2019*), which may help explain enhanced SVZ/WM microglia survival. It remains unclear whether PLX3397 concentration levels are lower in specific brain regions (i.e. bioavailability of the compound is lower in WM compared to grey matter areas possibly due to differential lipid abundance and compound solubility). However, we believe this is unlikely given the extent of microglial elimination in most WM areas and the close proximity of surviving SVZ/WM microglia to the ventricles (and exposure to higher peripheral doses of the compound).

Previous studies have shown that *Csf1-* and *Csf1r*-deficient mice exhibit severe macrophage population deficits, but that not all macrophages are depleted (*Rojo et al., 2019*; *Dai et al., 2002*; *Cecchini et al., 1994*), possibly through contributions from GM-CSF, IL-3 or other important myeloid cell growth factors (*Pixley and Stanley, 2004*). It was also shown that some macrophages persist in the absence of CSF1R in *Csf1r*-deficient zebrafish (*Kuil et al., 2020*). Since 3.5 months treatment with PLX3397 resulted in the eventual loss of SVZ/WM microglia, we hypothesize that this population of SVZ/WM microglia is partially resistant to CSF1R and requires a different threshold for cell death. These cells may rely on other factors for cell survival. A recent study identified a MAC2$^+$ CSF1Ri-

resistant microglial population following 14 days of PLX5622 (1200 ppm) treatment, postulating that alterations in of TREM2-TYROBP could explain CSF1Ri resistance (*Zhan et al., 2020*). This study highlights the existence of a progenitor-like microglial cells that is resistant to CSF1Ri; however, it is important to note that the authors report 88% depletion of IBA1[+] cells with similar repopulation dynamics to what we term GLOBAL repopulation. Thus, we do not believe the identified MAC2[+] population significantly contributes to WM repopulation.

In addition, several studies have highlighted the existence of microglial cells residing in brain-specific regions with distinct identities and properties (*Masuda et al., 2019*; *Li et al., 2019*; *Böttcher et al., 2019*). WM repopulating cells maintain altered phenotypes and responses to LPS, even after extended periods of time, suggesting that these SVZ/WM-derived cells are highly distinct from other microglia. In line with identifying distinct properties of myeloid cells in the SVZ/WM, previous studies have reported the unique heterogeneity of this microglial subset compared to subsets in other brain regions (*Stratoulias et al., 2019*; *Tan et al., 2020*). Similar to the morphological and molecular findings in WM repopulating cells, SVZ microglia are less branched (or less ramified), express surface markers that are commonly associated with an alternatively active phenotype (i.e. expressing high levels of anti-inflammatory cytokines IL-4 and IL-10), and lower expression levels of P2RY12 (*Ribeiro Xavier et al., 2015*). Interestingly, the existence of *Cx3cr1*[+] IBA1[-] cells in the SVZ and RMS has been reported (*Ribeiro Xavier et al., 2015*). In humans, microglia in the SVZ exhibit a more activated phenotype that was distinct from all other brain regions, and show higher expression of CD45, CD64, CD68, CX3CR1, EMR1, HLA-DR, indicative of microglial activation, and proliferation markers like Ki67 compared to other subsets (*Böttcher et al., 2019*). Moreover, WM microglia have been reported to display unique properties during postnatal stages, including an amoeboid morphology and enriched expression of genes related to microglial priming, phagocytosis, and migration (*Staszewski and Hagemeyer, 2019*). Thus, it appears that WM repopulating myeloid cells fit many of the previously reported characteristics of SVZ/WM microglia.

Once established in the brain, the gene expression profile of the WM repopulating cell shows enhanced enrichment for DAM genes (e.g., *Clec7a*, *Axl*, *Ly9*, *Apoe*, *Itgax*). A recent single-cell RNA-seq analysis has revealed a WM microglia-specific cluster and specifically a population of white matter-associated microglia (WAMs) that are dependent on TREM2 signaling (*Safaiyan et al., 2021*). The similarities between WAMs and WM repopulating cells are striking: WAMs are characterized by the downregulation in homeostatic microglial genes (*P2ry12*, *Hexb*) and upregulation in DAM-associated genes (*Apoe*, *B2m*, *Lyz2*, *and Clec7a*), cathepsins, and major histocompatibility complex (MHC) class II-related genes (*H2-D1*, *H2-K1*), which are also observed in WM repopulating cells. Droplet-based scRNAseq analysis revealed that these cells are located in the corpus callosum, a WM region with close proximity to the SVZ. WAM signature genes also include *Lpl* and *Itgax*, which are also elevated in 28 day WM repopulating cells. Furthermore, WM repopulating cells and WAMs express markers associated phagocytic activity, including CLEC7A and AXL (*Safaiyan et al., 2021*). An early postnatal phagocytic subset of microglia located in the WM (termed PAMs) has been identified, which also shares a gene signature with DAMs (*Li et al., 2019*). It appears that WM repopulating cells closely resemble WAMs, however, PAMs/ATMs may represent the developmental equivalent of this cell population. These data reinforce the discovery and existence of a distinct population of microglia in the WM; however further study is needed to examine whether surviving SVZ/WM microglia exist in the naïve brain as WAMs or whether CSF1Ri treatment induces this WAM-like phenotype.

WM is comprised primarily of myelinated axons that connect neurons in different regions of the brain. While WM abnormalities have been detected in a variety of neurological disorders, several microglial-related diseases (adult-onset leukoencephalopathy with axonal spheroids and pigmented glia; ALSP, Nasu-Hakola disease), caused by mutations in microglial-associated genes (e.g. *Csf1r*, *Tyrobp*, *Trem2*), have implicated microglial dysfunction/activation in extensive WM loss and altered cognitive function (*Sirkis et al., 2021*). Recent evidence shows that WAMs engulf MBP[+] particles during disease and facilitate myelin debris clearance (*Safaiyan et al., 2021*), thus developing ways to manipulate or enhance WM microglial function could prove therapeutically beneficial. Furthermore, since WAM cell numbers increase dramatically with age, identifying their source, properties, and mechanisms of emergence in the brain could prove vital to our understanding of aging and age-related disorders.

In conclusion, this study unveils the presence of a myeloid cell subtype originating from the SVZ and associated WM areas with increased CSF1Ri resistance that yields a dynamic wave of repopulating cells to reconstitute the microglial-depleted brain. These cells exhibit distinct properties compared to homeostatic microglia, sharing similar phenotypic and transcriptional profiles to DAMs and WAMs. Together, these results not only highlight the complexity and diversity of myeloid cells in the adult brain, but establish a model system that provides new insight on myeloid cell homeostasis and dynamics in the brain.

# Materials and methods

## Key resources table

| Reagent type (species) or resource | Designation | Source or reference | Identifiers | Additional information |
|---|---|---|---|---|
| Strain, strain background (*M. musculus*) | B6.129P2(C)-Cx3cr1$^{tm2.1(cre/ERT2)Jung}$ | Jackson Laboratory | IMSR Cat# JAX:020940, RRID:IMSR_JAX:020940 | |
| Strain, strain background (*M. musculus*) | STOCK Ascl1$^{tm1.1(Cre/ERT2)Jejo}$/J | Jackson Laboratory | IMSR Cat# JAX:012882, RRID:IMSR_JAX:012882 | |
| Strain, strain background (*M. musculus*) | C57BL/6-Tg (Nes-cre/ERT2)KEisc/J | Jackson Laboratory | IMSR Cat# JAX:016261, RRID:IMSR_JAX:016261 | |
| Strain, strain background (*M. musculus*) | B6.129X1-Gt(ROSA)26Sor$^{tm1(EYFP)Cos}$/J | Jackson Laboratory | IMSR Cat# JAX:006148, RRID:IMSR_JAX:006148 | |
| Strain, strain background (*M. musculus*) | B6.129(Cg)-Cx3cr1$^{tm1Litt}$Ccr2$^{tm2.1Ifc}$/JernJ | Jackson Laboratory | IMSR Cat# JAX:032127, RRID:IMSR_JAX:032127 | |
| Strain, strain background (*M. musculus*) | B6.129S4-Ccl2$^{tm1Rol}$/J | Jackson Laboratory | IMSR Cat# JAX:004434, RRID:IMSR_JAX:004434 | |
| Strain, strain background (*M. musculus*) | C57BL/6-Tg(CAG-EGFP)131Osb/LeySopJ | Jackson Laboratory | IMSR Cat# JAX:006567, RRID:IMSR_JAX:006567 | |
| Strain, strain background (*M. musculus*) | C57BL/6J | Jackson Laboratory | IMSR Cat# JAX:000664, RRID:IMSR_JAX:000664 | |
| Antibody | Anti-Iba1 (Rabbit polyclonal) | FUJIFILM Wako Shibayagi | Cat# 019–19741, RRID:AB_839504 | IF(1:1000) |
| Antibody | Anti-Iba1 (Goat polyclonal) | Abcam | Cat# ab5076, RRID:AB_2224402 | IF(1:1000) |
| Antibody | Anti-Cd11b (Rat monoclonal) | Bio-rad | Cat# MCA711, RRID:AB_321292 | IF(1:50) |
| Antibody | Anti-P2RY12 (Rabbit polyclonal) | Sigma-Aldrich | Cat# HPA014518, RRID:AB_2669027 | IF(1:200) |
| Antibody | Anti-TMEM119 (Rabbit monoclonal) | Abcam | Cat# ab209064, RRID:AB_2800343 | IF(1:200) |
| Antibody | Anti-AXL (Goat polyclonal) | R and D Systems | Cat# AF854, RRID:AB_355663 | IF(1:100) |
| Antibody | Anti-Dectin-1 (CLEC7A) (Rat monoclonal) | InvivoGen | Cat# mabg-mdect, RRID:AB_2753143 | IF(1:30) |
| Antibody | Anti-Ki67 (Rabbit monoclonal) | Abcam | Cat# ab16667, RRID:AB_302459 | IF(1:200) |
| Antibody | Anti-Ccl12 (Goat polyclonal) | R and D Systems | Cat# AF428, RRID:AB_2070875 | 100 ug/0.5 ml sterile HBSS |

*Continued on next page*

*Continued*

| Reagent type (species) or resource | Designation | Source or reference | Identifiers | Additional information |
|---|---|---|---|---|
| Antibody | Anti-Nestin (Mouse monoclonal) | Abcam | Cat# ab6142, RRID:AB_305313 | IF(1:200) |
| Antibody | Anti-MASH1 (Mouse monoclonal) | BD Biosciences | Cat# 556604, RRID:AB_396479 | IF(1:200) |
| Antibody | Anti-TIE2 (Mouse monoclonal) | Abcam | Cat# ab24859, RRID:AB_2255983 | IF(1:100) |
| Antibody | Anti-GFAP (Chicken polyclonal) | Abcam | Cat# ab4674, RRID:AB_304558 | IF(1:3000) |
| Antibody | Anti-doublecortin (DCX) (Goat polyclonal) | Santa Cruz Biotechnology | Cat# sc-8066, RRID:AB_2088494 | IF(1:200) |
| Antibody | Anti-Olig2 (Rabbit monoclonal) | Abcam | Cat# ab109186, RRID:AB_10861310 | IF(1:200) |
| Antibody | Anti-SOX2 (Goat polyclonal) | R and D Systems | Cat# AF2018, RRID:AB_355110 | IF(1:200) |
| Antibody | Anti-GFP (Rabbit polyclonal) | Abcam | Cat# ab6556, RRID:AB_305564 | IF(1:200) |
| Antibody | Anti-GFP (Chicken polyclonal) | Abcam | Cat# ab13970, RRID:AB_300798 | IF(1:200) |
| Antibody | Anti-PU.1 (Rabbit monoclonal) | Cell Signaling Technology | Cat# 2258, RRID:AB_2186909 | IF(1:200) |
| Antibody | Anti-Collagen IV (Rabbit polyclonal) | Abcam | Cat# ab6586, RRID:AB_305584 | IF(1:200) |
| Antibody | Anti-Parvalbumin (Mouse monoclonal) | Millipore | Cat# MAB1572, RRID:AB_2174013 | IF(1:200) |
| Antibody | Anti-Wisteria floribunda lectin (WFA) (Biotinylated) | Vector Laboratories | Cat# B-1355, RRID:AB_2336874 | IF(1:1000) |
| Antibody | Anti-PDGF Receptor alpha (PDGFRα) (Rabbit polyclonal) | Abcam | Cat# ab124392, RRID:AB_10978090 | IF(1:200) |
| Antibody | Anti-Myelin Basic Protein (MBP) (Rat monoclonal) | Millipore | Cat# MAB386, RRID:AB_94975 | IF(1:200) |
| Antibody | Anti-NeuN (Mouse monoclonal) | Millipore | Cat# MAB377, RRID:AB_2298772 | IF(1:1000) |
| Software, algorithm | Leica Application Suite X (LASX) | Leica | RRID:SCR_013673 | |
| Software, algorithm | Imaris | Bitplane | RRID:SCR_007370 | |
| Software, algorithm | Image J | Image J | RRID:SCR_003070 | |
| Software, algorithm | Fiji | Fiji | RRID:SCR_002285 | |
| Software, algorithm | Weighted Gene Co-expression Network Analysis | Software R package | RRID:SCR_003302 | |
| Software, algorithm | EdgeR | Bioconductor software package | RRID:SCR_012802 | |
| Software, algorithm | DESeq | Bioconductor software package | RRID:SCR_000154 | |

*Continued on next page*

*Continued*

| Reagent type (species) or resource | Designation | Source or reference | Identifiers | Additional information |
|---|---|---|---|---|
| Software, algorithm | LIMMA | Bioconductor software package | RRID:SCR_010943 | |
| Software, algorithm | Glimma | Bioconductor software package | RRID:SCR_017389 | |
| Software, algorithm | ggplot2 | Software R package | RRID:SCR_014601 | |
| Software, algorithm | EnhancedVolcano | Bioconductor software package | RRID:SCR_018931 | |
| Software, algorithm | clusterProfiler | Bioconductor software package | RRID:SCR_016884 | |
| Software, algorithm | maSigPro | Bioconductor software package | RRID:SCR_001349 | |
| Software, algorithm | Enrichr | Enrichr | RRID:SCR_001575 | |
| Software, algorithm | Ethovision XT | Noldus | RRID:SCR_000441 | |
| Software, algorithm | GraphPad Prism | GraphPad | RRID:SCR_002798 | |

## Compounds

PLX3397 was provided by Plexxikon Inc (Berkeley, CA) and formulated in AIN-76A standard chow by Research Diets Inc at the doses indicated in the text. PLX3397 was provided in chow at 600 ppm.

## Mice

All mice were obtained from The Jackson Laboratory (Bar Harbor, ME) unless otherwise indicated. $Cx3cr1^{CreERT2}$ (020940), $Ascl1^{CreERT2}$ (012882), and $Nestin^{CreERT2}$ (016261) mice were bred to R26-YFP (006148) reporter mice. $Cx3cr1^{GFP/GFP}Ccr2^{RFP/RFP}$ (032127) mice were bred to C57BL/6 to obtain $Cx3cr1^{GFP/+}Ccr2^{RFP/+}$ mice. H2K-*BCL*-2 transgenic mice were gifted from Irving Weissman. Ccl2 KO (004434) mice were obtained from The Jackson Laboratory. For transplant studies, bone marrow cells were isolated from CAG-EGFP mice (006567). All other mice were male C57BL/6 (000664) mice. Animals were housed with open access to food and water under 12 hr/12 hr light-dark cycles. All mice were aged to 1.5 months unless otherwise indicated.

## Animal treatments

All rodent experiments were performed in accordance with animal protocols approved (AUP-17–179) by the Institutional Animal Care and Use Committee at the University of California, Irvine (UCI). *Microglial depletion:* Mice were administered ad libitum with PLX3397 at a dosage of 600 ppm (to eliminate microglia) or vehicle (control) for 14 days. To stimulate repopulation, PLX3397 was withdrawn and replaced with vehicle. *LPS treatment:* Lipopolysaccharide (LPS; *Escherichia coli* 0111:B4; L4130, Sigma-Aldrich, St. Louis, MO) was dissolved in phosphate-buffered saline (PBS) and administered intraperitoneally (IP) at a dose of 0.33 mg/kg animal body weight either 6 or 24 hr prior to sacrifice. *BrdU labeling:* Bromodeoxyuridine (BrdU; 000103, Thermo Fisher Scientific, Waltham, MA) was administered at a dose of 1 ml/100 g body weight (per manufacturer's instructions) for four consecutive days. Mice were sacrificed 24 hr following last BrdU injection. *Tamoxifen treatment:* Tamoxifen (10540-29-1, Sigma-Aldrich) was suspended in corn oil for 60 min at 50°C. To obtain efficient conversion of *loxP* alleles a dose of 5 mg tamoxifen/ 25 g animal body weight was delivered orally over five consecutive days. Animals were injected with tamoxifen immediately following PLX3397 inhibition to track the lineage of the repopulating cells (for all lines except $Cx3cr^{CreERT2}$, in which TAM was administered 21 days prior to PLX3397 treatment). In vivo *neutralization of CCL12:* 100 ug of polyclonal goat anti-CCL12/MCP-5 (AF428, R and D Systems, Minneapolis, MN) or goat IgG in 0.5 ml of sterile HBSS was administered per mouse via i.p. injection at day 1, day 3, day 5, and day 6

(25 ug x 4) recovery (i.e., days post PLX3397 withdrawal). *Bone marrow transplant:* C57BL/6 mice were anesthetized with isoflurane and then irradiated with 1000 cGy (head-shielded) and reconstituted via retroorbital injection with 2 x 10$^6$ whole BM cells from CAG-EGFP mice. Blood was measured 4, 8, and 12 weeks post transplantation to track granulocyte chimerism. At 12 weeks post transplantation, the mice were euthanized and BM was harvested and analyzed by flow cytometry for HSC chimerism. This established an average percent chimerism of >40% in HS irradiated mice. *Tissue collection:* Following treatments, adult mice were sacrificed via carbon dioxide inhalation and perfused transcardially with 1X PBS. For mouse pups (below the age of postnatal day P9), animals were fully sedated using ice and then decapitated. Brains were extracted and dissected down the midline, with one half flash-frozen for subsequent RNA and protein analyses, and the other half drop-fixed in 4% paraformaldehyde. Fixed brains were cryopreserved in PBS + 0.05% sodium azide + 30% sucrose, frozen, and sectioned at 40 µm on a Leica SM2000 R sliding microtome for subsequent immunohistochemical analyses. *Subventricular zone microdissection and isolation:* Extracted brains were immersed in ice-cold HBSS (14025092, Thermo Fisher Scientific) and cut in half along the sagittal axis. Following removal of the septum, a thin layer of the rostral and lateral walls of the lateral ventricles were extracted from each hemisphere with a microsurgical stab knife (52–1501, Unique Technologies, Mohnton, PA) and immediately frozen in RNA isolation buffer solution.

## Histology and confocal microscopy

Fluorescent immunolabeling followed a standard indirect technique as described previously (*Elmore et al., 2014*). Brain sections were stained with antibodies against: IBA1 (1:1000; 019–19741, Wako Chemicals, Richmond, VA; and ab5076, Abcam, Cambridge, UK), CD11b (1:50, MCA711, Bio-Rad Laboratories, Hercules, CA), P2RY12 (1:200, HPA014518, Sigma-Aldrich), TMEM119 (1:200, ab209064, Abcam), AXL (1:100, AF854, R and D Systems), Dectin-1 (also known as CLEC7A; 1:30, mabg-mdect, Invivogen, San Diego, CA), Ki67 (1:200, ab16667, Abcam), NESTIN (1:200, ab6142, Abcam), MASH1 (1:200, 556604, BD Biosciences, San Jose, CA), TIE2 (1:100, ab24859, Abcam), GFAP (1:3000, ab4674, Abcam), DCX (1:200, sc-8066, Santa Cruz, Dallas, TX), OLIG2 (1:200, ab109186, Abcam), SOX2 (1:200, AF2018, R and D Systems), YFP/GFP (1:200, ab6556, Abcam), GFP (1:200, ab13970, Abcam), PU.1 (1:200, 2258, Cell Signaling Technology, Danvers, MA), Collagen IV (1:200, ab6586, Abcam), Parvalbumin (1:200, MAB1572, Millipore, Burlington, MA), WFA (1:1000, B-1355, Vector Laboratories, Burlingame, CA), PDGFRα (1:200, ab124392, Abcam), MBP (1:200, MAB386, Millipore), and NeuN (1:1000, MAB377, Millipore). For DAPI staining, mounted brain sections were cover-slipped using Fluoromount-G with DAPI (00-4959-52, Invitrogen, Carlsbad, CA). High resolution fluorescent images were obtained using a Leica TCS SPE-II confocal microscope (Leica Microsystems, Wetzlar, Germany) and LAS-X v 3.3.0 software. Images in the cortex were taken in the somatosensory cortex unless otherwise indicated. For confocal imaging, one field of view (FOV) per brain region was captured per mouse unless otherwise indicated. To capture brightfield images and whole brain stitches, automated slide scanning was performed using a Zeiss AxioScan. Z1 equipped with a Colibri camera (Zeiss, Oberkochen, Germany) and Zen AxioScan 2.3 software. Microglial morphology was determined using the filaments module in Bitplane Imaris 7.5 (Bitplane, Zurich, Switzerland), as described previously (*Elmore et al., 2015*). Cell quantities were determined using the spots module in Imaris. Percent coverage measurements were determined in Image J (NIH, Bethesda, MD).

## Cranial window implantation

Mice were anesthetized with isoflurane (Patterson Veterinary, Greeley, CO) in O$_2$ (2% for induction, 1–1.5% for maintenance). To provide perioperative analgesia, minimize inflammation, and prevent cerebral edema, Carprofen (10 mg/kg, s.c., Zoetis, Parsippany-Troy Hills, NJ) and Dexamethasone (4.8 mg/kg, s.c. Phoenix Pharmaceuticals, St. Joseph, MO) were administered immediately following induction. Ringer's lactate solution (0.2 mL/20 g/hr, s.c, Hospira, Lake Forest, IL) was given throughout the surgery to replace fluid loss. Sterile eye ointment (Rugby Laboratories, Hempstead, NY) was applied to prevent corneal drying. Surgical tools were sterilized using a hot glass bead sterilizer (Germinator 500, CellPoint Scientific, Gaithersburg, MD). Following hair removal, Povidone-iodine (Phoenix) and Lidocaine Hydrochloride Jelly (2%, Akorn, Lake Forest, IL) was used to disinfect and numb the scalp, respectively. The scalp and underlying connective tissue were removed to expose

the parietal and interparietal bone. Lidocaine Hydrochloride injectable (2%, Phoenix) was used for muscle analgesia and the right temporal muscle detached from the superior temporal line. The skull was dried using ethanol (70% in DI water) and a thin layer of Vetbond Tissue Adhesive (3M, Saint Paul, MN) applied to the exposed surface. Custom-printed ABS headplates were attached using Contemporary Ortho-Jet liquid and powder (Lang Dental, Wheeling, IL) at an angle parallel to the skull. A small craniotomy (3 mm diameter) was performed over the right hemisphere 2.5 mm anterior and 3 mm lateral lambda. Hemostatic gelfoam sponges (Pfizer, New York, NY) pre-soaked in sterile saline (CareFusion AirLife Modudose, CareFusion/BD, San Diego, CA) were used to absorb dural bleeding. Surgery was terminated if dural tears or intracerebral bleeding was observed. A 4 mm glass coverslip (World Precision Instruments, Sarasota, FL) was placed over the exposed brain and its edges attached to the skull first with a thin layer of Vetbond and second with dental acrylic. Following surgery, mice recovered in their home cage over a warm heating pad until normal behavior resumed (~15–30 min). Postoperative care consisted of daily Carprofen injections (10 mg/kg, s.c.) for 1 week.

## Two-photon imaging

Fluorescence was gathered with a resonant two-photon microscope (Neurolabware, Los Angeles, CA) with 900 nm excitation light (Mai Tai HP, Spectra-Physics, Santa Clara, CA). A 20x water immersion lens (1.0 NA, Olympus, Tokyo, Japan) was used with magnification 4. Emissions were filtered using a 510/84 nm and 607/70 nm BrightLine bandpass filter (Semrock, Rochester, NY). Image sequences were gathered using Scanbox acquisition software (Scanbox, Los Angeles, CA) at a depth of 200–260 µm below the meninges. An electrically tunable lens was used to image 20 planes (326x325µm, 3 µm z step), each sampled at 0.5 Hz. Laser damage consisted of line scanning at magnification 25 for 1–10 s at 800 nm.

## Quantification of homeostatic motility and response to laser ablation

All image stacks were processed and analyzed using the image processing package FIJI, a distribution of NIH Image J software (*Schindelin et al., 2012*). Stacks were temporally binned by taking the sum for each pixel over 30 frames (1 min.). Homeostatic motility was quantified manually by measuring the difference of visibly moving processes over 2 min. For each mouse, 10 microglia were chosen at random and the first five extension or retraction observed were recorded for a total of 50 observations per mouse. Microglia respond to laser damage by extending their processes toward the site of injury. We took advantage of increasing fluorescence at the site of damage from infiltrating GFP-positive processes to quantify microglia response to laser ablation. The average GFP intensity within a circle (r = 50 µm) centered at site of damage at any timepoint ($t_x$) was normalized to the intensity in that area at 1 min. post ablation ($t_1$). To determine differences in GFP intensity within groups over time and between groups at any timepoint, we used a repeated measures two-way ANOVA corrected for multiple comparison (Geisser-greenhouse correction).

## PK analysis

PLX3397 concentration in plasma and cerebellum were analyzed for pharmacokinetic (PK) data by Integrated Analytical Solutions, (Inc).

## FACS analysis

Myeloid cells were extracted from whole hemispheres, isolated into single-cell suspensions and identified using fluorescence-activated cell sorting (FACS) gating for $CD11b^+CD45^{int}$ as previously described (*Elmore et al., 2014*). Cells were stained with the following surface antibodies purchased from Biolegend (San Deigo, CA) at 1:200 unless otherwise indicated: CD34-eFlour660 (1:50, 50-0341-80, eBioscience, San Diego, CA), Sca-1-AF700 (1:100, 108141), CD16/32-PE (101307), Ter119-PE/Cy5 (116209), ckit/CD117-PE/Cy7 (25-1171-81, eBioscience), CD150/SLAM-PerCP-eFlour710 (46-1502-82, eBioscience), CD11b-APC (101212), CD11b-PE (101208), Gr1-AF700 (108422), CD45-AF700 (103128), CD45-APC/Cy7 (103116), NK1.1-PE (108707), CD3-PE/Cy7 (100220), CD19-PerCyanine5.5 (45-0193-82, eBioscience), CD11c-APC/Cy7 (117323), Ly6C-PE (1:400, 128007), Ly6G-5.5 (127615). For HSCs, CMPs, and GMPs, all cells were gated on live (PI⁻), Ter119⁻ cells and then identified with the following gating strategy: HSCs: $FcyR^-$, $ckit^+$ $Sca^+$ $CD34^-$, $SLAM^+$, CMPs: $FcyR^-$, $ckit^+$,

Sca⁻, CD35⁺, and GMPs: FcyR⁺, ckit⁺, Sca⁻, CD34⁺. Samples were acquired with the BD LSRII or BD Fortessa X20, and sorted with the BD FACS Aria II.

## Nanostring analysis

RNA was extracted and purified from frozen half brains using an RNA Plus Universal Mini Kit (Cat. #73404, Qiagen). For nCounter analysis, total RNA was diluted to 20 ng/µl and probed using a mouse nCounter PanCancer Immune Profiling Panel (Nanostring Technologies, Seattle, WA, USA) profiling ~700 immunology-related mouse genes. Counts for target genes were normalized to the best fitting house-keeping genes as determined by nSolver software. The WGCNA package was used to evaluate the quality of reads, as well, as identify and remove appropriate outliers, based on standard deviation within normalized expression values. PCA plots were generated using plot3D. Negative binomial linear regression analysis was performed using EdgeR, DESeq, and Limma packages to generate FDR and log fold change values. Top significant genes are displayed as a volcano plot constructed using GLimma, ggplot2, and EnhancedVolcano.

## RNA-sequencing and analysis

Total RNAs were extracted by using RNeasy Mini Kit (Qiagen, Hilden, Germany). RNA integrity number (RIN) was measured by Agilent 2100 Bioanalyzer (Agilent Technologies, Santa Clara, CA) and samples with RIN >= 7.0 were kept for cDNA synthesis. cDNA synthesis and amplification were performed followed by Smart-seq2 (*Picelli et al., 2014*) standard protocol. Libraries were constructed by using the Nextera DNA Sample Preparation Kit (Illumina, San Diego, CA). Libraries were base-pair selected based on Agilent 2100 Bioanalyzer profiles and normalized as determined by KAPA Library Quantification Kit (Illumina). The libraries were sequenced using paired-end 43 bp mode on Illumina NextSeq500 platform with around 10 million reads per sample. *Read alignment and expression quantification:* Pair-end RNA-seq reads were aligned using STAR v.2.5.1b with the options (`−outFilterMismatchNmax 10 −outFilterMismatchNoverReadLmax 1 −outFilterMulti-mapNmax 10`) (*Dobin et al., 2013*). Rsubread was used to generate feature counts (*Liao et al., 2019*). Gene expression was measured using Limma, edgeR, and org.Mm.eg.db packages with expression values normalized as RPKM (*Robinson et al., 2010*; *McCarthy et al., 2012*; *Carlson, 2018*; *Ritchie et al., 2015*).

### Differential expression analysis

Libraries with uniquely mapping percentages higher than 80% were considered to be of good quality and kept for downstream analysis. Protein coding and long non-coding RNA genes, with expression RPKM > = one in at least three samples, were collected for subsequent analysis. Differential expression analysis was performed by using Limma, edgeR, and org.Mm.eg.db (*Robinson et al., 2010*; *McCarthy et al., 2012*; *Carlson, 2018*; *Ritchie et al., 2015*). Differentially expressed genes (DEGs) were selected by using false discovery rate (FDR)<0.05. Top significant genes are displayed as a volcano plot constructed using GLimma, ggplot2, and EnhancedVolcano (FDR < 0.05, LogFC >1) (*Blighe, 2019*). PCA plots were generated using plot3D (*Soetaert, 2017*).

### Gene ontology and pathway analysis

DEGs were analyzed for Gene ontology (GO) enrichment by clusterProfiler using a hypergeometric test with corrected p-values < 0.05 (*Yu et al., 2012*). These results were then plotted with GOplot. maSigPro package was used to identify genes that show different gene expression profiles over time (*Conesa et al., 2006*). Heatmaps were generated by mapping RPKM values to genes identified in maSigPro and then constructed using gplots. Normalized (min max normalization for each individual gene) log2-transformed expression values are displayed as a heatmap with hierarchical clustering utilizing gplots. maSigPro-selected gene clusters were identified and enriched for GO using cluster-Profiler. GOplot was used to correlate genes and important pathways. DEGs of 3 mo recovery mice were analyzed for Gene ontology (GO) enrichment by enrichR with corrected p-values < 0.05.

## Behavioral and cognitive analysis

Mouse behavior, motor function, and cognition was evaluated using the following tasks: elevated plus maze, open field, novel place/novel object, sociability test, and spontaneous alternation Y-maze

in the order listed, and as previously described unless otherwise indicated (*Elmore et al., 2014*; *Elmore et al., 2015*; *Spangenberg et al., 2019*; *Spangenberg et al., 2016*). Testing was conducted at 28 days recovery (i.e. after CSF1Ri removal and microglial repopulation).

## Sociability test

The Crawley's or Three-Chamber Sociability test assesses general sociability, or time spent with another rodent. In brief, animals were placed in a Three-chamber Sociability Test box (19 cm x 45 cm) with two dividing walls made of clear Plexiglas allowing free access to each chamber. During habituation, the subject mouse is placed in the middle chamber for 5 min for adaption. During testing (24 hr after habituation), a stranger mouse (inside a wire containment cup) is placed in one of the side chambers, and the subject mouse was placed in the center chamber and allowed to access and explore all three chambers for 10 min. The placement of the stranger mouse in the left and right chambers is systemically altered between trials. The duration of time spent in each chamber, velocity, and distance traveled was measured. *Spontaneous Alternation Y-Maze:* For this task, mice were placed in a Y-maze (35.2 cm arm length x 5 cm width x 20 cm sidewall height). Each animal was allowed to freely explore the arena for 8 min. Distinct intra-maze visual cues were positioned at the end of each arm for spatial orientation. Spontaneous alternation, which measures the willingness of an animal to explore new environments, was measured by the number of triads, or entry of all three arms in a consecutive sequence (i.e. ABC and not BAB). Unless otherwise indicated, behavioral readouts for all tasks were calculated from video using the EthoVision XT 14 tracking system (Noldus, Leesburg, VA).

## Data analysis and statistics

Statistical analysis was performed with Prism Graph Pad (v.8.1.1, GraphPad Software, San Diego, CA). To compare two groups, the unpaired Student's t-test was used. To compare multiple groups, a one-way ANOVA with Tukey's posthoc test was performed. For all analyses, statistical significance was accepted at $p < 0.05$. All bar graphs are represented as means +/- SEM and significance expressed as follows: *$p < 0.05$, **$p < 0.01$, ***$p < 0.001$. n is given as the number of mice within each group.

## Acknowledgements

We thank Edna Hingco and Ayer Darling Jue for their excellent technical assistance, and Brian L West and Andrey Rymar at Plexxikon, Inc for providing and formulating CSF1Ri chow and pharmacokinetics analysis. We are grateful to Claudia I Czimczik for proofreading the manuscript. This work was supported by the National Institutes of Health (NIH) under awards: R01NS083801 (NINDS), R01AG056768 (NIA), and P50AG016573 (NIA) to KNG; F31NS108611 (NINDS) to JDC and T32NS082174 (NINDS) to YG. LAH was supported by the Alzheimer's Association Research Fellowship (AARF-16–442762).

## Additional information

### Funding

| Funder | Grant reference number | Author |
|---|---|---|
| National Institute of Neurological Disorders and Stroke | R01NS083801 | Kim N Green |
| National Institute on Aging | RF1AG056768 | Kim N Green |
| National Institute on Aging | P50AG016573 | Kim N Green |
| National Institute of Neurological Disorders and Stroke | F31NS108611 | Joshua Crapser |
| National Institute of Neurological Disorders and Stroke | T32NS082174 | Yasamine Ghorbanian |
| Alzheimer's Association | AARF-16-442762 | Lindsay A Hohsfield |

The funders had no role in study design, data collection and interpretation, or the decision to submit the work for publication.

### Author contributions

Lindsay A Hohsfield, Conceptualization, Data curation, Formal analysis, Funding acquisition, Validation, Investigation, Visualization, Methodology, Writing - original draft, Writing - review and editing; Allison R Najafi, Yasamine Ghorbanian, Data curation, Formal analysis, Investigation, Methodology, Writing - review and editing; Neelakshi Soni, Formal analysis, Investigation, Methodology; Joshua Crapser, Conceptualization, Methodology, Writing - review and editing; Dario X Figueroa Velez, Data curation, Investigation, Visualization, Methodology; Shan Jiang, Formal analysis, Validation, Visualization, Methodology; Sarah E Royer, Investigation, Methodology; Sung Jin Kim, Data curation, Investigation, Methodology; Caden M Henningfield, Data curation; Aileen Anderson, Supervision, Writing - review and editing; Sunil P Gandhi, Resources, Supervision; Ali Mortazavi, Resources, Software, Supervision; Matthew A Inlay, Resources, Supervision, Writing - review and editing; Kim N Green, Conceptualization, Resources, Software, Formal analysis, Supervision, Funding acquisition, Project administration, Writing - review and editing

### Author ORCIDs

Lindsay A Hohsfield https://orcid.org/0000-0002-6018-656X
Aileen Anderson http://orcid.org/0000-0002-8203-8891
Kim N Green https://orcid.org/0000-0002-6049-6744

### Ethics

Animal experimentation: All rodent experiments were performed in accordance with animal protocols approved (AUP-17-179) by the Institutional Animal Care and Use Committee at the University of California, Irvine (UCI).

### Decision letter and Author response

Decision letter https://doi.org/10.7554/eLife.66738.sa1
Author response https://doi.org/10.7554/eLife.66738.sa2

## Additional files

### Supplementary files

• Transparent reporting form

### Data availability

Sequencing data have been deposited in GEO under accession code GSE166092, and can be explored in an interactive fashion at http://rnaseq.mind.uci.edu/green/. All other data generated or analysed during this study are included in the manuscript and support files.

The following dataset was generated:

| Author(s) | Year | Dataset title | Dataset URL | Database and Identifier |
| --- | --- | --- | --- | --- |
| Green KN, Hohsfield LA, Soni N | 2021 | Subventricular zone/white matter microglia reconstitute the empty adult microglial niche in a dynamic wave | http://www.ncbi.nlm.nih.gov/geo/query/acc.cgi?acc=GSE166092 | NCBI Gene Expression Omnibus, GSE166092 |

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
