## [Decision Letter]

**Acceptance summary:**

This is a comprehensive cellular and molecular investigation of the repopulation dynamics of microglia following extensive elimination through administration of the CSF1r antagonist PLX3397. The investigators describe an intriguing capacity for a small subpopulation of white matter and subventricular zone surviving microglia to repopulate the entire brain. They provide extensive characterization of the "wave" pattern of migration and proliferation during repopulation and provide important information about the transcriptional profile of these cells. This study has important implications for our understanding of microglia development and repopulaiton dynamics as well as implications for therapeutics with CSF1r antagonists.

**Decision letter after peer review:**

Thank you for submitting your article "Subventricular zone/white matter microglia reconstitute the empty adult microglial niche in a dynamic wave" for consideration by *eLife*. Your article has been reviewed by 3 peer reviewers, including Jaime Grutzendler as the Reviewing Editor and Reviewer #1, and the evaluation has been overseen by Carla Rothlin as the Senior Editor.

Essential revisions:

These are the main issues to address but please look at the individual reviews for further details that should also be considered:

1) Since this paper is entirely dependent on the idea that there is a subpopulation of microglia cells originating from SVZ, that repopulates the cortex after PLX3397 treatment, it will be critical to show better quality high- resolution imaging through confocal tiling spanning large areas of the cortex and white matter. Stain and show several markers of microglia to more convincingly demonstrate that there are no surviving microglia in the cortex that would explain local repopulation. Related to this, In lines 164 for Figure 2C, the authors claim that microglial markers are lost rather than downregulated. But the images have faint (green) signals that seem to be more indicative of downregulation rather than loss. Please explain this.

2) On the basis of the appearance of microglia at different locations after 14 days of PLX3397 treatment, the authors claim that the repopulating cells elicit a dynamic wave migrating radially and tangentially from the SVZ/ ventricular zone, white matter tract, and caudoputamen. To establish this observation more convincingly and to rule out the possibility of sparse local repopulation specifically in the cortex, as some cells can be seen in figure 3 B (stage III) in the cortex, the authors need to present a more detailed temporal and spatial resolution of the "Wavefront". More detailed imaging is required using high-resolution confocal tiling capturing different time points between stage III and IV to show the gradual emergence of the wavefront from the white matter tract and invading the cortex. A similar extensive analysis of the wavefront in the lineage tracing using cx3cr1-creER should also be performed.

3) The authors make several quantitative pronouncements without adequate quantitative data that is needed. Eg., while co-expression of certain proteins and myeloid cells (or the lack thereof) are mentioned (lines 247-249), quantitative assessments are required i.e. what percent of IBA1+ cells are NESTIN+ or MASH^+^ etc? Similarly, the authors make claims of ~25% or ~100% labeling in (lines 260-261) but precise numbers should be provided. It is clear to this reviewer that at least for Figure 4E, there are red (IBA1+ cells) that are not CX3CR1+. Moreover, for the various "stages" in Figure 3B-D, there is no indication of time (days) following PLX withdrawal which would help substantiate temporal claims. Data in Figure 6A-B are just images and would be helped by some quantification of the number of cells. Similarly, data in Figure 6G-K can be presented as percent of GFP cells amongst IBA1 cells as discussed in lines 361-367 and 371. In lines 394-395 while the number of DEG genes is specified for END repopulated cells compared to control microglia (65), the number is not stated for PND repopulated cells. The authors should provide the exact number rather than stating it as a "few" genes.

4) Do the authors have any sense of what is unique about the surviving cells in the SVZ, why do they escape PLX3397 treatment? If they are simply cx3cr1 positive myeloid cells, the transcriptome data provided only answers how these cells differ once they start repopulation the brain, however it does not answer how they were unique prior to PLX3397 treatment that made them survive. Additional data characterizing this would be important to further determine the relevance of this finding. This issue was also raised by Reviwer#3 (1a).

5) The issue of pharmacokinetic analysis of PLX3397 is important. One possibility is that the cells in the white matter rather than being more resistant to PLX3397 receive a relatively lower concentration of the drug than in the cortex due to either reduced perfusion of these areas or alternatively since PLX3397 is lipophilic, it could be incorporated in the myelin sheets thereby reducing its effective concentration. Ideally, the methods used to measure PLX3397 brain concentration should be applied separately to cortical and white matter tissues. What happens if the concentration of PLX3397 used is higher? Is the white matter wavefront still formed?

Perhaps complete elimination of myeloid cells would be achieved with a longer PLLX treatment paradigm. The authors should consider an up to 2-3 month treatment protocol. In addition, the mention of "sustained" in line 196 should be replaced with the specific duration i.e. 14 days to remove any potential confusion that the authors could mean longer that 14 days.

6) After 28 days of recovery period the authors observed that the repopulating cells after PLX3397 treatment were transcriptionally different from the control population. It would be important to know whether these transcriptional differences in the empty niche repopulating microglia are temporary or are sustained even after a longer recovery time. It is possible that these cells show a *delayed* transition to more control microglial-like features which may be detected at longer than 28 days. Looking at up to 3 months after reconstitution would help to clarify whether the differences between reconstituted cells and control microglia are indeed not due to a slower restoration in the microglial phenotype.

7) In the AD mouse model, the authors administered PLX3397 and analyzed the effect on amyloid plaque deposition after 3 months of recovery (Figure 9). On the basis of this experiment, the authors claim that the repopulating cells could reduce the amyloid load. However, the significance of this experiment is unclear and does not fit with the central theme of the paper. There is a complete lack of controls addressing amyloid synthesis, processing, secretion, and clearance before and after PLX 3397 treatment. Therefore, the data provided is superficial and not sufficient to demonstrate such a causal relationship. This part should be completely revamped or eliminated from the study.

Also, it is difficult to draw appropriate conclusions from the 5XFAD studies provided in Figure 9E-P because they lack the 14d PLX treatment results. This is needed to determine whether some of the features are unchanged between control and repopulated 5XFAD results or else the 14d depletion is could have itself changed things and then the repopulation would restore the change. Therefore, the 14d PLX results is critically needed data.

8) The relative resistance of the microglia in the SVZ and adjacent WM track could very well be due to the local cues/environment, which alter the signaling and lead to relative resistance. While interesting from the experimental perspective, it might not be physiologically relevant for how endogenous microglia repopulate the brain.

9) The comparison of control microglia vs. 28-day repopulated microglia in response to LPS is not particularly informative because of the artificial nature of the timing and treatment. It could also be due to the cross talk with astrocytes and other cell types, which are also altered under the 2-week 3397 treatment.

(10) The H2k-BCL2 experiment (Figure S1B-C ) is potentially interesting. Why put it in supplementary? It would be best to move to main figures and add proper quantification. The H2K-BCL2 studies are a bit confusing. First, the statement that these mice provide "overexpression of BCL2 in myeloid lineages affords some resistant to CSF1Ri-induced cell death" is not substantiated by published (or new) data. In addition, no evidence is provided that microglia indeed show over expression of BCL2. Third, the mechanism of BCL2 protection from cell death is not mentioned (presumably through reduction in apoptosis?) Fourth, in the initial study by this group (Elmore et al., 2014), they suggested that CSF1Ri-induced cell death using PLX3397 occurs by apoptosis stating "To confirm that microglia undergo cell death with CSF1R inhibition, we found that many microglia stained for active caspase-3 in the same animals, a classic marker of apoptosing cells (Figure S2J- S2L)" on page 381. Since blocking apoptosis with BCL2 overexpression only increases survival from 0.2-5-10%, do the authors want to expand on whether cell death primarily occurs by apoptosis as they suggested in a previous study?

*Reviewer #1:*

In this article, Hohsfield et al. developed a method to achieve complete elimination of microglia in the adult brain using a sustained high dose of PLX3397 for 14 days. On the basis of immunofluorescence staining of microglia and different transgenic lines, the authors show that the repopulating myeloid cells are cx3cr1 positive and originate from the subventricular zone and white matter tract. The extensive CSF1R inhibition unveils the presence of inhibitor-resistant myeloid cells in the subventricular zone/white matter areas. The authors show that the repopulating cells were CCR2-RFP negative, eliminating the possibility of peripheral immune cell infiltration. Furthermore, the repopulating microglia are transcriptionally distinct from the control microglia and were enriched with disease-associated microglia genes, like AXL, ApoE, Clec7a, and cst7. To understand the functional impact of these empty niche repopulating microglia on mice, the authors performed various behavioral tests. Also, in AD-like mice, the authors show that administration of PLX3397 for 14days could reduce the amyloid load in the cortex and amygdala.

Overall, this study is comprehensive and depicts a potentially interesting microglial repopulation dynamic under extensive microglial depletion. The potential relevance of this information is that these method for microglia ablation is widely used and that the repopulation mechanism could have relevance for understanding mechanisms of microglia development. If PLX3397 or similar drugs are ever approved , it is important to know how they affect the brain. However, the major claims of this paper, the dynamic wave of microglia repopulating the whole brain and complete elimination after a sustained high dose of PLX3397, need more extensive and rigorous analysis. Following are the major concerns we have with this paper.

1. Since this paper is entirely dependent on the idea that there is a subpopulation of microglia cells originating from SVZ, that repopulates the cortex after PLX3397 treatment, it will be critical to show better quality high- resolution imaging through confocal tiling spanning large areas of the cortex and white matter. Stain and show several markers of microglia to more convincingly demonstrate that there are no surviving microglia in the cortex that would explain local repopulation.

2. On the basis of the appearance of microglia at different locations after 14 days of PLX3397 treatment, the authors claim that the repopulating cells elicit a dynamic wave migrating radially and tangentially from the SVZ/ ventricular zone, white matter tract, and caudoputamen. To more convincingly establish this observation and to rule out the possibility of sparse local repopulation specifically in the cortex, as some cells can be seen in figure 3 B (stage III) in the cortex, the authors need to present a more detailed temporal and spatial resolution of the "Wavefront". More detailed imaging is required using high-resolution confocal tiling capturing different time points between stage III and IV to show the gradual emergence of the wavefront from the white matter tract and invading the cortex. A similar extensive analysis of the wavefront in the lineage tracing using cx3cr1-creER should also be performed.

3. The issue of pharmacokinetic analysis of PLX3397 is important. One possibility is that the cells in the white matter rather than being more resistant to PLX3397 receive a relatively lower concentration of the drug than in the cortex due to either reduced perfusion of these areas or alternatively since PLX3397 is lipophilic, it could be incorporated in the myelin sheets thereby reducing its effective concentration. Ideally, the methods used to measure PLX3397 brain concentration should be applied separately to cortical and white matter tissues. What happens if the concentration of PLX3397 used is higher? Is the white matter wavefront still formed?

4. Do the authors have any sense of what is unique about the surviving cells in the SVZ, why do they escape PLX3397 treatment? If they are simply cx3cr1 positive myeloid cells, the transcriptome data provided only answers how these cells differ once they start repopulation the brain, however it does not answer how they were unique prior to PLX3397 treatment that made them survive. Additional data characterizing this would be important to further determine the relevance of this finding.

5. After 28 days of recovery period the authors observed that the repopulating cells after PLX3397 treatment were transcriptionally different from the control population. It would be interesting to know whether these transcriptional differences in the empty niche repopulating microglia are temporary or are sustained even after a longer recovery time.

6. In the AD mouse model, the authors administered PLX3397 and analyzed the effect on amyloid plaque deposition after 3 months of recovery (Figure 9). On the basis of this experiment, the authors claim that the repopulating cells could reduce the amyloid load. However, the significance of this experiment is unclear and does not fit with the central theme of the paper. There is a complete lack of controls addressing amyloid synthesis, processing, secretion, and clearance before and after PLX 3397 treatment. Therefore, the data provided is superficial and not sufficient to demonstrate such a causal relationship.

7. The H2k-BCL2 experiment (Figure S1B-C ) is potentially interesting. Why put it in supplementary? It would be best to move to main figures and add proper quantification.

*Reviewer #2:*

The manuscript by Hohsfield et al. used high dose of CSF1R/FLT/Kit inhibitor to diminish brain microglia to less than 0.1%, which mostly located in the SVZ and adjacent white matter, called END. They then described their molecular characteristics and the repopulation routes. There are interesting observations with broad description of the characteristics and the resistant population and the population routes. The study also confirmed some of the previous findings, such as the brain-origin of the repopulation. They also showed even after 28 days, the repopulated microglia from END exhibit differential responses to LPS.

The major concern of the study is that the findings failed to lead to deeper understanding of microglial homeostasis regulation in health and in disease in its current form.

1) The relative resistance of the microglia in the SVZ and adjacent WM track could very well be due to the local cues/environment, which alter the signaling and lead to relative resistance. While interesting from the experimental perspective, it might not be physiologically relevant for how endogenous microglia repopulate the brain.

2) Some of the conclusions are confirmation of previous studies, such as the origin of END population.

3) The comparison of control microglia vs. 28-day repopulated microglia in response to LPS is not particularly informative because of the artificial nature of the timing and treatment. It could also be due to the cross talk with astrocytes and other cell types, which are also altered under the 2-week 3397 treatment.

*Reviewer #3:*

In this study by Hohsfield et al., the authors set out to develop a complete elimination of microglia in the brain followed by its reconstitution. Using a high dose (600ppm) PLX3397 14 day treatment, they show the most effective depletion strategy for IBA1+-brain cells at 99.98%. This is indeed the most extensive microglial depletion strategy to date. They refer to this as a microglial "empty niche" environment and then proceed to determine the reconstitution mechanisms. Interestingly, unlike previous paradigms (referred to as a partially depleted niche), where microglia are either reconstituted by proliferation of residual microglia or infiltration of peripheral myeloid cells, in this empty niche scenery, reconstitution is slower and only partial by a month. Furthermore, reconstitution follows defined routes similar to that previously reported for developmental colonization of myeloid cells that become microglia. However, even after a month of populating the brain, these cells remain phenotypic and functionally distinct from endogenous microglia. remarkable, this paradigm when applied to AD its shown to reduce amyloid plaques burden. These are interesting findings, many of which are novel. However, some of the conclusions are derived from insufficiently substantiated data that lack some appropriate controls and enough quantification and so can be improved by addressing these concerns for an otherwise interesting subject matter.

This is manuscript is an attempt to show a novel CSF1Ri-resistent myeloid cell-type that originates in the subventricular zone/white matter and reconstitutes an empty microglial niche following near total CSF1R-dependent microglial elimination. The authors claim that reconstituting myeloid cells populate the empty microglia niche in the brain through migratory routes akin to that employed by endogenous microglia during development. These myeloid cells show a distinct profile from homeostatic microglia and maintain a distinct profile following extended residence in the brain. Finally, in a model of AD, reconstituting the brain with this approach and these cells resulted in a lower AD load that suggests this could be a promising approach to plaque burden in AD. Despite these extremely exciting findings, the manuscript in its current form suffers from significant concerns that need to be addressed before it can be suitable for publication at *eLife*.

1. Overstated or insufficiently substantiated claims: several details in the manuscript, while interesting overstate the evidence provided by the data or are not substantiated by the current data.

a. Figure 3 seeks to provide spatio-temporal evidence for the "dynamic wave" of repopulation. however, while these findings are interesting, they seem to be based of mainly spatial (without temporal) data taken "between 7 and 14" days of PLX withdrawal. While the impression is given that the authors assess the myeloid cell migration at different timepoints between these time periods, the authors seem to make these conclusions and claims solely from static images at (rather than between) 7 and 14 days. To substantiate 'temporal" claims, the authors need to provide analysis at different time points between 7 and 14 days which they don't seem to have done.

b. The authors make several quantitative pronouncements without adequate quantitative data that is needed. Eg., while co-expression of certain proteins and myeloid cells (or the lack thereof) are mentioned (lines 247-249), quantitative assessments are required i.e. what percent of IBA1+ cells are NESTIN+ or MASH^+^ etc? Similarly, the authors make claims of ~25% or ~100% labeling in (lines 260-261) but precise numbers should be provided. It is clear to this reviewer that at least for Figure 4E, there are red (IBA1+ cells) that are not CX3CR1+. Moreover, for the various "stages" in Figure 3B-D, there is no indication of time (days) following PLX withdrawal which would help substantiate temporal claims. Data in Figure 6A-B are just images and would be helped by some quantification of the number of cells. Similarly, data in Figure 6G-K can be presented as percent of GFP cells amongst IBA1 cells as discussed in lines 361-367 and 371. In lines 394-395 while the number of DEG genes is specified for END repopulated cells compared to control microglia (65), the number is not stated for PND repopulated cells. The authors should provide the exact number rather than stating it as a "few" genes.

c. A major claim that permeates the manuscript is that the residual myeloid cells are CSF1Ri-resistant. However, this is based upon a 14-day treatment regimen. Perhaps complete elimination of myeloid cells would be achieved with a longer PLLX treatment paradigm. The authors should consider an up to 2-3 month treatment protocol. In addition, the mention of "sustained" in line 196 should be replaced with the specific duration i.e. 14 days to remove any potential confusion that the authors could mean longer that 14 days.

d. Similarly, the density phenotypes (Figure 1F) and the morphology phenotypes (Figure 1 J-L) were only assessed up to 28 days following PLX withdrawal. It is possible that these cells show a *delayed* transition to more control microglial-like features which may be detected at longer than 28 days. This reviewer recommends looking at up to 3 months after reconstitution (which the authors did with the 5XFAD data in Figure 9). This would help to clarify whether the differences between reconstituted cells and control microglia are indeed not due to a slower restoration in the microglial phenotype.

e. In lines 164 for Figure 2C, the authors claim that microglial markers are lost rather than downregulated. But the images have faint (green) signals that seem to be more indicative of downregulation rather than loss. Please explain this.

f. Lines 150-151 suggest that repopulating cells display larger cell bodies with no indication of a recovery to normal levels despite the fact that the corresponding Figure shows this in Figure 1J.

g. For data in Figure 3E, the cells are referred to as "proliferating END repopulating myeloid cells" (line 233) but there is no clear evidence of colocalization of Ki67 and IBA1.

h. The authors state that the myeloid cells they identify are CSF1Ri-resistant after 14 days of PLX3397 (600ppm) treatment. But to ensure that they are indeed truly resistant, a longer (and "sustained"-line 1960) exposure of perhaps up to 2 months would be needed.

i. In lines 308 – 311, the authors claim that since CCL12 antibodies reduced populating cell numbers but not the total distance of spreading. However, it is not clear what "the total distance of cell spreading" is. In the images, the territory occupied by the green cells (myeloid cells) seems to be smaller in antibody treated tissues. Moreover, although the authors claim that this combined observation leads to the conclusion that CCL12 may play a role in the proliferation of repopulating cells, this is not the only possibility. For example, the antibody could affect the survival of cells rather than the proliferation. The authors could detect proliferation using markers in control and CCL12 antibody treatment.

j. In line 340-341, the authors claim the CMPs are expanded by flow but they do not provide the gating strategy here. This will help to compare to the cells assessed in the other study.

k. Finally, it is difficult to draw appropriate conclusions from the 5XFAD studies provided in Figure 9E-P because they lack the 14d PLX treatment results. This is needed to determine whether some of the features are unchanged between control and repopulated 5XFAD results or else the 14d depletion is could have itself changed things and then the repopulation would restore the change. Therefore, the 14d PLX results is critically needed data.

2. Internally inconsistent findings: Some aspects of the manuscript provide some inconsistent results that need to be addressed:

a. In Figure 1E and 1F, the regions are the same (ssCTX) but the values for thee control conditions are wildly different (~80 cells per FOV in 1E v. ~190 cells per FOV in 1F). This inconsistency is glaring and needs to be explained and/or rectified.

b. In Figure 1F-G, 14d recovery from PLX treatment leads to an INCREASE of ~50% (from ~190 – ~90 cells) in the ssCTX or ~20% (from ~320 – ~250 cells). However, in Figure 6F, 14d recovery from PLX leads to a 100% INCREASE in cell numbers (from ~3000 – ~6000 cells). These results are inconsistent and should be explained or reconciled.

3. Poor data presentation and Figures:

a. In Figure 1, the flow of data is supposed to be a comparison between PND and END repopulation. However, for most of the results, only END data is presented e.g. 1G; 1I-L.

b. To be considerate of color-blind individuals, the manuscript will be helped by color coding images in green and magenta rather than green and red.

c. Many of the Figures provide full brain images that do not allow sufficient assessment of the claims given based on the images provided. This reviewer thinks the points will be clearer if large brain images are saved for supplemental figures and more detailed images are left in the manuscript proper. E.g. the 14d PLX condition in Figure 1D seems to have some green puncta that may be considered cells but is better determined by more focused images as in D1-4. These kinds of concerns are present in many of the Figures including Figure 2A-C (rather than the whole brain images, individual P2YR12/ IBA1 single frames in addition to the merged frames would be helpful), Figure 6G-J and Figure 9C (see 9D where the cells and amyloid can be more clearly seen).

d. For some figure panels, there isn't always consistency in what is being shown.

e. It would be helpful if the protein stains are consistent. E.g. IBA is red in Figure 4C-E but green in Figure 4F-H. Similarly, Ki67 is magenta in Figure 4F but red in Figure 4G-H. It would be helpful to maintain the same color scheme (especially for a marker like IBA that is used throughout the manuscript) as best as possible.

f. It is not clear why the supplemental Figures are presented as e.g. for Figure S1 "Figure 1-Supplement 1". This supplemental Figure is related to Main Figure 2 not main Figure 1. This Figure is also labeled as Tmem119 in panel A but the legend say "P2RY12". The pattern in the figure is also inconsistent because the control, 14 and 28 days are in the cortex but the 7d is in the WM. This is given without explanation. If necessary, both regions can be shown but the inconsistency is confusing.

g. For Figure 2H, it seems like the expression of IBA1 is very low in the original image that is brightened in the insert. Why is this?

4. Outstanding questions:

a. Although the study highlights the existence of the CSF1Ri-resident population, it is not clear whether this population exists in the normal brain (which would imply a "progenitor" pool) or is induced to take on this phenotype following PLX treatment. The study does not seem interested in this important question. This could potentially be determined by staining for some of the upregulated DEG genes (proteins) in the SVZ/WM of the untreated brain. If the number following 14d PLX (~15 in line 170) is much more reduced than that in the naïve brain, then it becomes difficult to argue for a "resistant" population since the population would have been reduced from whatever the number in the naive brain to 15.

b. The H2K-BCL2 studies are interesting but a bit confusing. First, the statement that these mice provide "overexpression of BCL2 in myeloid lineages affords some resistant to CSF1Ri-induced cell death" is not substantiated by published (or new) data. In addition, no evidence is provided that microglia indeed show over expression of BCL2. Third, the mechanism of BCL2 protection from cell death is not mentioned (presumably through reduction in apoptosis?) Fourth, in the initial study by this group (Elmore et al., 2014), they suggested that CSF1Ri-induced cell death using PLX3397 occurs by apoptosis stating "To confirm that microglia undergo cell death with CSF1R inhibition, we found that many microglia stained for active caspase-3 in the same animals, a classic marker of apoptosing cells (Figure S2J- S2L)" on page 381. Since blocking apoptosis with BCL2 overexpression only increases survival from 0.2-5-10%, do the authors want to expand on whether cell death primarily occurs by apoptosis as they suggested in a previous study?

c. The language used in the discussion on lines 547-549 is strong. The authors seem to be confusing different models. Their findings do not "refute" those other findings since they used different approaches and unlike the current findings, it is likely that an empty microglial niches did not occur in those studies.

d. Finally, it is not clear that the identity of the remaining IBA1+ cells in this study is different from.

[Editors' note: further revisions were suggested prior to acceptance, as described below.]

Thank you for resubmitting your work entitled "Subventricular zone/white matter microglia reconstitute the empty adult microglial niche in a dynamic wave" for further consideration by *eLife*. Your revised article has been evaluated by Carla Rothlin (Senior Editor) and a Reviewing Editor.

The manuscript has been improved but there are some remaining issues that need to be addressed, as outlined below:

1) The paper has significantly improved. However, there are still concerns about the scope of the claims about the surviving microglia being the so-called WAMs. The fact that this small number of cells in WM survives could be due to many factors including lower drug bioavailability in WM. Please tone down this claim and explain possible alternative reasons for the survival of these cells post-drug treatment.

2) While in this particular case it appears that WM microglia are responsible for most of the cortical repopulation, this is a unique situation and may not reflect a specific property of WM microglia. Several papers including recent ones in *eLife* have clearly shown repopulation from surviving cortical microglia. Please include these in the discussion and tone down the claims of the uniqueness of WM microglia in their ability to repopulate the brain ( PMID: 33054973, PMID: 34250902)

3) Please go through the critiques from the 3 reviewers and address them as much as possible. No need for further experiments, just data clarifications, and additions to the Discussion section.

*Reviewer #1:*

In the revised manuscript Hohsfield et al. have answered many of the concerns raised. Specifically:

1. In the earlier version of this manuscript the images showing the "wavefront" of repopulating microglia in the cortex was not very convincing. In the revised manuscript, as per our suggestion, the authors have added high-resolution confocal tiling of large areas showing the wavefront at different stages of the recovery. This data does show more convincingly that dividing microglia emerge from areas newa corpus callosum and appear to advance towards cortex. However, this data also seems to show patchy cortical areas with proliferating microglia that to not appear to emerge from the Wavefront. This suggest that other sources of regenerating microglia exist in the courtext (likely small number of surviving microglia)

2. To address our concerns about the uniqueness of microglia in the white matter tract and their resistance to PLX3397 treatment, the authors postulate based on recent literature that these microglia are resistant due to enhanced TREM2 and CSF1 expression. Additionally, the authors added bulk tissue transcriptome data of 14 days PLX-treated and after 3-5 days recovery time. The authors show that the initial repopulating microglia have genes upregulated in the pathways related to myeloid cell activation/priming, pathogen sensing, and monocyte-macrophage signaling.

3. We had concerns that whether after longer recovery time post PLX treatment, the repopulated microglia would still be transcriptionally different or not. To address this the authors have added transcriptome data after 3 months of recovery and suggest that they are transcriptionally different from the control.

4. We asked whether a longer duration of PLX3397 treatment will eliminate microglia from SVZ and white matter tract or not. The authors in the revised manuscript report that a longer duration of PLX3397 (3 months) eliminates microglia from the white matter region. This suggests that the small population of surviving microglia in WM has a different threshold for cell death, or the bioavailability of the compound is somehow lower in WM compared to cortex. Ideally the authors should discuss this in greater detail in their final version as it is difficult to say with the current data the precise reason for the relative sparing of some WM microglia

5. The authors took our suggestion to completely remove the Alzheimer's disease model data as the data was not conclusive and to add the H2K-BCL2 with more characterization in the main figures.

Overall, as it is the paper is comprehensive and contains interesting data related to the repopulation dynamics of microglia post extensive elimination using PLX3397 for 14 days. The repopulating microglia are transcriptionally different than the control homeostatic microglia and they maintain this difference up to 3 months of recovery. Since the origin of this most (although not likely all) repopulating microglia is from white matter, the authors claim to uncover new insights into white matter microglia function and dynamics. However, we are not very confident about the authors claim of the uniqueness of the white matter microglia. As the authors showed that a longer duration of PLX3397 treatment (3 months) could completely eliminate microglia from white matter, suggesting that they are susceptible to PLX3397 too. Also, the perfusion of this drug in different regions of the brain is not understood. The transcriptional differences observed in the repopulating microglia could be due to the enhanced load of phagocytosis due to extensive PLX3397 treatment. A recent article published in *eLife* by Mendes et al. (*eLife* 2021;10:e61173) using in vivo 2 photon imaging showed that the microglia post depletion in cortex repopulate rapidly and locally rather than from migrating microglia and attain morphological maturity within days. Thus, we suggest authors tone down their conclusion about uncovering the uniqueness of white matter microglia. I would suggest changes in the title and abstract which at the moment to us appear to strongly suggest that WM are very unique and the pathway for migration towards cortex could have implications during development, etc. These are interesting hypotheses but are not necessarily well supported by the data

*Reviewer #2:*

The manuscript addressed many of the concerns, and is much improved in the quality of the presentation. A comparison with the WAM from the recent study is interesting. A. highly relevant *eLife* paper published last year reported single cell RNA-seq analyses of a relative resistant population ( *eLife*. 2020 Oct 15;9:e51796. doi: 10.7554/*eLife*.51796). How do these two resistant populations compare in terms of transcriptomic signatures?

*Reviewer #3:*

This seems to be a much-improved manuscript but this reviewer cannot fully assess the revisions because it lacks appropriate notification of the changes in the text to allow visualization / assessments of the amendments. For example, for Reviewer 3's comment 1f, the authors state that they corrected the reference to the recovery to normal levels of the cell soma. But this reviewer does not see this mention in the text. The same is true for the response to comment 1g and 1i. Moreover, page numbers referenced in the Essential Revisions do not match up with the text. Also, in response to Reviewer 3 question 1d, Figure 9 is mentioned rather than Figure 10. Also, for 2b, "Figure 6F" is no longer that and the new figure should be mentioned in the response. In response to question 4b, the authors reference Essential Revisions 9 but the response is Essential Revisions 10. This reviewer considers this a set of very unfortunate failures that excessively prolonged this review that has made reviewing the revision extremely difficult.

At a minimum for the next revision, the authors should (i) easily identify the parts of the text that have been revised in a different-colored font and (ii) accurately cite the line numbers in the text where the said revisions are located in the response letter. This should be done for ALL Reviewer 3's comments (both the individualized and the Essential revisions) especially Essential comment 10, points 2-4.

For this reviewer, the following questions have been addressed and DO NOT need to be revisited in a revised resubmission: Questions 1a, b, d, j, k, 2a, 3a, d, f, g, 4c.

For Figure 1c, answers are provided in Essential Revisions 5. But these are not quite satisfactory. The question was whether "residual myeloid cells are CSF1Ri-resistant" with a 14-day treatment regimen. The authors state that "[w]e would like to point out that in many cases we do achieve 100% microglial elimination in whole brain sections following 14d PLX3397 treatment". This actually implies that these cells are not "resistant". In addition, the additional data provided for 3.5 months indeed shows that all microglia are eliminated but the important question of whether the wavefront still forms in this context is not answered as the authors do not show a recovery data after 3.5 months.

For 3b, please work with the journal to generate color blind-sensitive figures.

For 3c, while this reviewer does not think this answer is satisfactory, I will defer to the Editor and other reviewers. If they agree with this response, then I will agree to it too.

For 3e, it is a simple enough request to change the color scheme to facilitate ease of understanding by readers. Please, do this.

For question 4a, the location of the amended tests should be identified. Moreover, the authors do not really answer the question. The question was whether the "resistant" pool (~15 cells) of cells exist in the naïve brain. From the response, it seems like the authors think these cells are equivalent too recently published WAMs. This WAMs are more numerous that these resistant cells in the naïve brain. Therefore, can the authors differentiate between naïve cells becoming WAMs with PLX treatment and naïve WAMs?

---

## [Author Response]

Essential revisions:These are the main issues to address but please look at the individual reviews for further details that should also be considered:1) Since this paper is entirely dependent on the idea that there is a subpopulation of microglia cells originating from SVZ, that repopulates the cortex after PLX3397 treatment, it will be critical to show better quality high- resolution imaging through confocal tiling spanning large areas of the cortex and white matter. Stain and show several markers of microglia to more convincingly demonstrate that there are no surviving microglia in the cortex that would explain local repopulation. Related to this, In lines 164 for Figure 2C, the authors claim that microglial markers are lost rather than downregulated. But the images have faint (green) signals that seem to be more indicative of downregulation rather than loss. Please explain this.

We thank the reviewer for this comment. To address this concern, we have put together a new figure (Figure 3) that focuses on this issue, providing high-resolution confocal images of large regions of the cortex and white matter stained for several microglial markers (see Figure 3B-G, Figure 5—figure supplement 4F). We have also re-generated all our figures in Adobe Illustrator, which provide much improved quality images to explore. For Figure 2C, the faint green signals (Cx3cr1-CreERT2) are unspecific or vascular staining that arises when no staining is present in the brain. This is more apparent when the green channel is boosted to show that the brain is indeed in focus, however, we have adjusted the green channel back to its original levels. We apologize for the confusion it caused. Furthermore, if the cells were still present in these brains (with downregulated staining) then we would also observe faint red (Iba1+ staining), which we do not, and repopulation would occur in clusters radiating out from surviving cells, which we do not observe. Additionally, downregulation of signal would not explain the disappearance of cells from our Cx3cr1-CreERT2 linage tracing experiments, or the vast literature that has confirmed that microglia are killed with CSF1Ri (or genetic ablation of CS1/IL34, or CSF1R).

2) On the basis of the appearance of microglia at different locations after 14 days of PLX3397 treatment, the authors claim that the repopulating cells elicit a dynamic wave migrating radially and tangentially from the SVZ/ ventricular zone, white matter tract, and caudoputamen. To establish this observation more convincingly and to rule out the possibility of sparse local repopulation specifically in the cortex, as some cells can be seen in figure 3 B (stage III) in the cortex, the authors need to present a more detailed temporal and spatial resolution of the "Wavefront". More detailed imaging is required using high-resolution confocal tiling capturing different time points between stage III and IV to show the gradual emergence of the wavefront from the white matter tract and invading the cortex. A similar extensive analysis of the wavefront in the lineage tracing using cx3cr1-creER should also be performed.

To address this concern, we have added a new supplemental figure (Figure 2—figure supplement 1) that provides high-resolution whole brain (Figure 2—figure supplement 1A) and confocal images (Figure 2—figure supplement 1B) of various timepoints of the “wavefront.” Figure 5—figure supplement 4 provides two whole brain images of the wavefront in Cx3cr1-CreERT2 mice, with a high-resolution image of the “wavefront.”

3) The authors make several quantitative pronouncements without adequate quantitative data that is needed. Eg., while co-expression of certain proteins and myeloid cells (or the lack thereof) are mentioned (lines 247-249), quantitative assessments are required i.e. what percent of IBA1+ cells are NESTIN+ or MASH^+^ etc? Similarly, the authors make claims of ~25% or ~100% labeling in (lines 260-261) but precise numbers should be provided. It is clear to this reviewer that at least for Figure 4E, there are red (IBA1+ cells) that are not CX3CR1+. Moreover, for the various "stages" in Figure 3B-D, there is no indication of time (days) following PLX withdrawal which would help substantiate temporal claims. Data in Figure 6A-B are just images and would be helped by some quantification of the number of cells. Similarly, data in Figure 6G-K can be presented as percent of GFP cells amongst IBA1 cells as discussed in lines 361-367 and 371. In lines 394-395 while the number of DEG genes is specified for END repopulated cells compared to control microglia (65), the number is not stated for PND repopulated cells. The authors should provide the exact number rather than stating it as a "few" genes.

The reviewer makes a valid point and we have added quantifications to support our claims accordingly. For lines 247-249 (now lines 280-287), we have added quantifications of the percentage of IBA1+ cells positive for precursor and other cell lineage markers (see Figure 5—figure supplement 3F-H). For lines 260-261 (now lines 297-306), we have quantified the percentage of IBA1+ cells that express YFP (see Figure 5—figure supplement 4B). For Figure 3B-D (now Figure 2), we have added approximate time/days following PLX withdrawal, but it should be noted that this timing is dependent on extent of depletion and can vary among animals. For Figure 6A-B (now Figure 7), we have added a quantification of the number of IBA1+ cells within the choroid plexus and outside of it (in the parenchymal areas) (see Figure 7B). For Figure 6G-K (now Figure 7), we have added quantification of the percentage of GFP+ IBA1+ cells above the graph (see Figure 7K). For lines 394-395, we have added the number of DEGs for GLOBAL repopulated cells (now line 445).

4) Do the authors have any sense of what is unique about the surviving cells in the SVZ, why do they escape PLX3397 treatment? If they are simply cx3cr1 positive myeloid cells, the transcriptome data provided only answers how these cells differ once they start repopulation the brain, however it does not answer how they were unique prior to PLX3397 treatment that made them survive. Additional data characterizing this would be important to further determine the relevance of this finding. This issue was also raised by Reviwer#3 (1a).

We postulate local cues from the white matter environment and ventricular zone allow these myeloid cells to escape PLX3397 treatment. This is demonstrated by the fact that once they leave these areas, they become susceptible to PLX3397 treatment, as demonstrated in Figure S2I-J, similar to non-SVZ/WM microglia. Although the signals responsible for shaping macrophage/microglia identity are still being discovered, studies have revealed that the local niche or microenvironment can play an active role in establishing macrophage identity (Gosselin et al., 2014 *Cell*; Lavin et al., 2014 *Cell*). We postulate that a combination of transcription factors, availability of CSF1, and epigenetic regulation (via tissue-specific enhancers) contribute to the uniqueness of WM/SVZ microglia and their ability to escape PLX3397 treatment. Several studies have shown that the parenchymal microglial population is heterogeneous highlighting the existence of cells residing in brain-specific regions with distinct identities (Bottcher et al., 2019 *Nat Neurosci*, Li et al., 2019 *Neuron*, Masuda et al., 2019 *Nature*). In fact, a recent single-cell RNAseq analysis has revealed a white matter microglia-specific cluster and specifically a population of white matter-associated microglia (WAMs) that are dependent on TREM2 signaling. The similarities between these WAMs and WM repopulating cells are striking, including a shared disease-associated microglia (DAM) gene signature, along with an upregulation in genes and markers associated phagocytic activity, including CLEC7A and AXL (Safaiyan et al., 2020 *Neuron*). These recent data indicate that our WM repopulating cells are likely WAMs. Notably, DAMs are also resistant to CSF1Ri (including PLX3397; Spangenberg et al., 2016 *Brain*). We postulate that surviving SVZ/WM microglia could exhibit enhanced survival due to an upregulation of TREM2 and CSF1 (upregulated in DAMs/WAMs), which can also act as survival signals (Zheng et al., 2018 *Front Aging Neurosci*). We have added text regarding this issue to the discussion (see lines 634-646). To address the reviewer’s concern about further characterization, we have performed transcriptional characterization of the tissue following 14d PLX3397 treatment (prior to repopulation) and following 5d/7d recovery (see Figure 3J-L and Figure 6A-D), and characterization of WM repopulating cells at 28d recovery (see Figure 8B-D). We have also added a new figure (see Figure 3) that characterizes the expression of multiple microglial markers in surviving WM/SVZ microglia. Unfortunately, due to the extremely low amount of cells that are present at 14d, it is not feasible at this time to perform transcriptional analyses of these cells. It should be noted that we do not believe these cells change their identity during repopulation.

5) The issue of pharmacokinetic analysis of PLX3397 is important. One possibility is that the cells in the white matter rather than being more resistant to PLX3397 receive a relatively lower concentration of the drug than in the cortex due to either reduced perfusion of these areas or alternatively since PLX3397 is lipophilic, it could be incorporated in the myelin sheets thereby reducing its effective concentration. Ideally, the methods used to measure PLX3397 brain concentration should be applied separately to cortical and white matter tissues. What happens if the concentration of PLX3397 used is higher? Is the white matter wavefront still formed?Perhaps complete elimination of myeloid cells would be achieved with a longer PLLX treatment paradigm. The authors should consider an up to 2-3 month treatment protocol. In addition, the mention of "sustained" in line 196 should be replaced with the specific duration i.e. 14 days to remove any potential confusion that the authors could mean longer that 14 days.

We thank the reviewers for this comment. We would like to point out that most white matter microglia are eliminated by 14d PLX3397. This is apparent throughout the corpus callosum. It is only specifically in areas near white matter/ventricular zone (near the ventricles) that CSF1R-resistant surviving cells are found. Thus, it doesn’t appear that the cells in the white matter receive a different concentration, in fact with their close proximity to the ventricles, one would assume the opposite was true. We have added a discussion of this issue in the discussion (see lines 641-644). We would also like to point out that in many cases we do achieve 100% microglial elimination in whole brain sections following 14d PLX3397 treatment. In these cases, the wavefront still forms. As requested, we have also added data in which animals were treated with PLX3397 (600 ppm) for 3.5 mo (see Figure 3—figure supplement 2F-H). We have replaced “sustained” in line 196 with 14d PLX3397 (see line 253-4).

6) After 28 days of recovery period the authors observed that the repopulating cells after PLX3397 treatment were transcriptionally different from the control population. It would be important to know whether these transcriptional differences in the empty niche repopulating microglia are temporary or are sustained even after a longer recovery time. It is possible that these cells show a *delayed* transition to more control microglial-like features which may be detected at longer than 28 days. Looking at up to 3 months after reconstitution would help to clarify whether the differences between reconstituted cells and control microglia are indeed not due to a slower restoration in the microglial phenotype.

We agree with the author that it would be important to analyze repopulating microglia at a later timepoint and have isolated brain tissue at 90d recovery and performed RNA sequencing analysis. The new data from 90d (~3 mo) recovery mice can be found in Figure 10. These data provide evidence that WM repopulating cells remain phenotypically/morphologically and transcriptionally distinct from homeostatic and GLOBAL repopulating cells rather than regain a homeostatic microglial phenotype.

7) In the AD mouse model, the authors administered PLX3397 and analyzed the effect on amyloid plaque deposition after 3 months of recovery (Figure 9). On the basis of this experiment, the authors claim that the repopulating cells could reduce the amyloid load. However, the significance of this experiment is unclear and does not fit with the central theme of the paper. There is a complete lack of controls addressing amyloid synthesis, processing, secretion, and clearance before and after PLX 3397 treatment. Therefore, the data provided is superficial and not sufficient to demonstrate such a causal relationship. This part should be completely revamped or eliminated from the study.Also, it is difficult to draw appropriate conclusions from the 5XFAD studies provided in Figure 9E-P because they lack the 14d PLX treatment results. This is needed to determine whether some of the features are unchanged between control and repopulated 5XFAD results or else the 14d depletion is could have itself changed things and then the repopulation would restore the change. Therefore, the 14d PLX results is critically needed data.

We agree with the reviewer. Given the recent findings by Safaiyan et al. (2020), which highlight the existence of WAMs that represent a potentially protective response to aging and disease (including increased phagocytic potential), and the striking similarities between WAMs and our WM repopulating cells, the significance of this experiment and our study has changed. We have chosen to remove these data and will provide a more thorough analysis of these data and findings in a subsequent manuscript.

8) The relative resistance of the microglia in the SVZ and adjacent WM track could very well be due to the local cues/environment, which alter the signaling and lead to relative resistance. While interesting from the experimental perspective, it might not be physiologically relevant for how endogenous microglia repopulate the brain.

It should be noted that the appearance of repopulating cells is similar in phenotypical profiles and anatomical locations to microglia in the developing brain (Ueno et al., 2013 *Nat Neurosci*). We also have follow-up data highlighting how infiltrating myeloid cells use similar routes to spread throughout the adult brain. In addition, a recent study has highlighted a population of amoeboid microglia that migrate from the ventricular zone into the corpus callosum and engulf OPCs during early postnatal development (Nemes-Baran et al., 2020 *Cell Rep*). Given the recent findings by Safaiyan et al. (2020), which highlight the existence of WAMs, and the striking similarities between WAMs and our WM repopulating cells, the significance of this experiment has changed. Notably, WAMs are present at low numbers in the adult brain but increase dramatically with age, indicating their relevance to brain aging and age-related diseases. Understanding the sources, properties, and emergence of these cells in the brain is therefore of great importance, which our study sheds light on. Thus, we argue that these findings may have high physiological relevance and novel future application for understanding disease pathogenesis. We would also like to point out that the study of any microglial depletion model (e.g. Elmore et al. 2014, *Neuron*; Buttger et al. 2016, *Immunity*) is non-physiological, however, depletion methods provide us with a tool to better understand the biological role and function of myeloid cells as well as their population dynamics in the CNS. All microglial depletion models have caveats, despite this, the field still has gained much insight on microglial biology, and this is especially important for a cell that has been implicated in many neurological disorders. Given the disparate findings regarding microglial heterogeneity, in what appears to be dependent on the method of isolation and RNAseq technique utilized, microglial depletion models could serve an important role in helping decipher these differences.

9) The comparison of control microglia vs. 28-day repopulated microglia in response to LPS is not particularly informative because of the artificial nature of the timing and treatment. It could also be due to the cross talk with astrocytes and other cell types, which are also altered under the 2-week 3397 treatment.

We agree with the reviewer that it is important to consider cross talk with astrocytes and other cell types. Unfortunately, this study did not focus on reporting changes in astrocytes in this paradigm. It is important to note though that during LPS administration, astrocytes are also present, along with repopulating cells. We have added bulk tissue RNAseq analysis of 28d recovery so readers can observe other cellular contributions (see Figure 3J-L). Several studies using microglial depletion and repopulation models have relied on LPS challenge to evaluate the functional immune response differences between endogenous and repopulating cells (O’Neil et al. 2018 *Acta Neuropathol. Commun*; Lund et al. 2018 *Nat Comm;* Elmore et al. 2018 *Aging Cell*). Previously, we have shown that GLOBAL repopulating (at 21d recovery or repopulation) microglia respond in a similar fashion to LPS challenge as control microglia (Elmore et al. 2015 *PLoS One*), whereas in this study we show that WM repopulating cells exhibit functionally alterations compared to control microglia. To address this comment, we have tempered our conclusions from this study and added text about the possible caveats of this experiment to the discussion, including timing/treatment and astrocytes/other cell type contributions (see lines 486-87).

(10) The H2k-BCL2 experiment (Figure S1B-C ) is potentially interesting. Why put it in supplementary? It would be best to move to main figures and add proper quantification. The H2K-BCL2 studies are a bit confusing. First, the statement that these mice provide "overexpression of BCL2 in myeloid lineages affords some resistant to CSF1Ri-induced cell death" is not substantiated by published (or new) data. In addition, no evidence is provided that microglia indeed show over expression of BCL2. Third, the mechanism of BCL2 protection from cell death is not mentioned (presumably through reduction in apoptosis?) Fourth, in the initial study by this group (Elmore et al., 2014), they suggested that CSF1Ri-induced cell death using PLX3397 occurs by apoptosis stating "To confirm that microglia undergo cell death with CSF1R inhibition, we found that many microglia stained for active caspase-3 in the same animals, a classic marker of apoptosing cells (Figure S2J- S2L)" on page 381. Since blocking apoptosis with BCL2 overexpression only increases survival from 0.2-5-10%, do the authors want to expand on whether cell death primarily occurs by apoptosis as they suggested in a previous study?

We agree with the reviewer and have now added the H2K-BCL2 experiment to the main figures as well as proper quantification (see Figure 4A-D). To address confusion regarding these studies, we have also done the following: (1) added microglial quantification of control (see Figure 4A) and 14d PLX3397 treated H2K-BCL2 mice (see Figure 4B-D) – which addresses the reviewer concern about the enhanced survival of microglia in these mice and the resistance of these mice to CSF1Ri, (2) updated citations and statements about BCL expression in transgenic mice, (3) added text to postulate the mechanism of BCL2 protection from cell death, and (4) added text to expand on how CSF1Ri results in cell death (see Lines 259-269).

Reviewer #1:

All comments are addressed in Essential Revisions above.*Reviewer #2:*

The manuscript by Hohsfield et al. used high dose of CSF1R/FLT/Kit inhibitor to diminish brain microglia to less than 0.1%, which mostly located in the SVZ and adjacent white matter, called END. They then described their molecular characteristics and the repopulation routes. There are interesting observations with broad description of the characteristics and the resistant population and the population routes. The study also confirmed some of the previous findings, such as the brain-origin of the repopulation. They also showed even after 28 days, the repopulated microglia from END exhibit differential responses to LPS.The major concern of the study is that the findings failed to lead to deeper understanding of microglial homeostasis regulation in health and in disease in its current form.(1) The relative resistance of the microglia in the SVZ and adjacent WM track could very well be due to the local cues/environment, which alter the signaling and lead to relative resistance. While interesting from the experimental perspective, it might not be physiologically relevant for how endogenous microglia repopulate the brain.

See Essential Revisions #8.

(2) Some of the conclusions are confirmation of previous studies, such as the origin of END population.

To date, no microglial depletion study has reported the existence of a CSF1Ri resistant microglial cell located in the SVZ/WM tracts. Previous microglial depletion models have shown that repopulation of the microglial niche arises from surviving microglia (microglia we term GLOBAL), which we demonstrate in this manuscript are transcriptionally and functionally distinct despite arising from a Cx3cr1+ cell source. Given the recent discovery of WAMs and the striking similarities to WM microglia, we believe the origin of WM microglia does not share the same origin as surviving microglia in gray matter areas (which we will follow up on in a future publication).

(3) The comparison of control microglia vs. 28-day repopulated microglia in response to LPS is not particularly informative because of the artificial nature of the timing and treatment. It could also be due to the cross talk with astrocytes and other cell types, which are also altered under the 2-week 3397 treatment.

See Essential Revisions #9.

Reviewer #3:This is manuscript is an attempt to show a novel CSF1Ri-resistent myeloid cell-type that originates in the subventricular zone/white matter and reconstitutes an empty microglial niche following near total CSF1R-dependent microglial elimination. The authors claim that reconstituting myeloid cells populate the empty microglia niche in the brain through migratory routes akin to that employed by endogenous microglia during development. These myeloid cells show a distinct profile from homeostatic microglia and maintain a distinct profile following extended residence in the brain. Finally, in a model of AD, reconstituting the brain with this approach and these cells resulted in a lower AD load that suggests this could be a promising approach to plaque burden in AD. Despite these extremely exciting findings, the manuscript in its current form suffers from significant concerns that need to be addressed before it can be suitable for publication at eLife.1. Overstated or insufficiently substantiated claims: several details in the manuscript, while interesting overstate the evidence provided by the data or are not substantiated by the current data.a. Figure 3 seeks to provide spatio-temporal evidence for the "dynamic wave" of repopulation. however, while these findings are interesting, they seem to be based of mainly spatial (without temporal) data taken "between 7 and 14" days of PLX withdrawal. While the impression is given that the authors assess the myeloid cell migration at different timepoints between these time periods, the authors seem to make these conclusions and claims solely from static images at (rather than between) 7 and 14 days. To substantiate 'temporal" claims, the authors need to provide analysis at different time points between 7 and 14 days which they don't seem to have done.

See Essential Revisions #2.

b. The authors make several quantitative pronouncements without adequate quantitative data that is needed. Eg., while co-expression of certain proteins and myeloid cells (or the lack thereof) are mentioned (lines 247-249), quantitative assessments are required i.e. what percent of IBA1+ cells are NESTIN+ or MASH^+^ etc? Similarly, the authors make claims of ~25% or ~100% labeling in (lines 260-261) but precise numbers should be provided. It is clear to this reviewer that at least for Figure 4E, there are red (IBA1+ cells) that are not CX3CR1+. Moreover, for the various "stages" in Figure 3B-D, there is no indication of time (days) following PLX withdrawal which would help substantiate temporal claims. Data in Figure 6A-B are just images and would be helped by some quantification of the number of cells. Similarly, data in Figure 6G-K can be presented as percent of GFP cells amongst IBA1 cells as discussed in lines 361-367 and 371. In lines 394-395 while the number of DEG genes is specified for END repopulated cells compared to control microglia (65), the number is not stated for PND repopulated cells. The authors should provide the exact number rather than stating it as a "few" genes.

See Essential Revisions #3.

c. A major claim that permeates the manuscript is that the residual myeloid cells are CSF1Ri-resistant. However, this is based upon a 14-day treatment regimen. Perhaps complete elimination of myeloid cells would be achieved with a longer PLLX treatment paradigm. The authors should consider an up to 2-3 month treatment protocol. In addition, the mention of "sustained" in line 196 should be replaced with the specific duration i.e. 14 days to remove any potential confusion that the authors could mean longer that 14 days.

See Essential Revisions #5.

d. Similarly, the density phenotypes (Figure 1F) and the morphology phenotypes (Figure 1 J-L) were only assessed up to 28 days following PLX withdrawal. It is possible that these cells show a *delayed* transition to more control microglial-like features which may be detected at longer than 28 days. This reviewer recommends looking at up to 3 months after reconstitution (which the authors did with the 5XFAD data in Figure 9). This would help to clarify whether the differences between reconstituted cells and control microglia are indeed not due to a slower restoration in the microglial phenotype.

We would like to point out to the reviewer that we did evaluate microglial density and morphology phenotypes in Figure 9 and did observe differences between reconstituted cells and control microglia in wildtype mice at 3 mo recovery. However, to make these observations more apparent, we have removed the 5xFAD data and instead focused our data on the cellular (and transcriptional) alterations in wildtype animals after 3 months of reconstitution (see Figure 10).

e. In lines 164 for Figure 2C, the authors claim that microglial markers are lost rather than downregulated. But the images have faint (green) signals that seem to be more indicative of downregulation rather than loss. Please explain this.

See Essential Revisions #1.

f. Lines 150-151 suggest that repopulating cells display larger cell bodies with no indication of a recovery to normal levels despite the fact that the corresponding Figure shows this in Figure 1J.

We thank the reviewer for pointing this out and have corrected the text accordingly.

g. For data in Figure 3E, the cells are referred to as "proliferating END repopulating myeloid cells" (line 233) but there is no clear evidence of colocalization of Ki67 and IBA1.

We thank the reviewer for pointing this out and have added a reference to this colocalization in the text.

h. The authors state that the myeloid cells they identify are CSF1Ri-resistant after 14 days of PLX3397 (600ppm) treatment. But to ensure that they are indeed truly resistant, a longer (and "sustained"-line 1960) exposure of perhaps up to 2 months would be needed.

See Essential Revisions #5.

i. In lines 308 – 311, the authors claim that since CCL12 antibodies reduced populating cell numbers but not the total distance of spreading. However, it is not clear what "the total distance of cell spreading" is. In the images, the territory occupied by the green cells (myeloid cells) seems to be smaller in antibody treated tissues. Moreover, although the authors claim that this combined observation leads to the conclusion that CCL12 may play a role in the proliferation of repopulating cells, this is not the only possibility. For example, the antibody could affect the survival of cells rather than the proliferation. The authors could detect proliferation using markers in control and CCL12 antibody treatment.

We have clarified “total distance” in the Figure Legend and added this additional interpretation to the manuscript.

j. In line 340-341, the authors claim the CMPs are expanded by flow but they do not provide the gating strategy here. This will help to compare to the cells assessed in the other study.

We have included our written gating strategy in the Methods section for HSCs, CMPs and GMPs. However, we would be happy to provide a visual of our gating strategy to the Reviewer.

k. Finally, it is difficult to draw appropriate conclusions from the 5XFAD studies provided in Figure 9E-P because they lack the 14d PLX treatment results. This is needed to determine whether some of the features are unchanged between control and repopulated 5XFAD results or else the 14d depletion is could have itself changed things and then the repopulation would restore the change. Therefore, the 14d PLX results is critically needed data.

We have removed this data.

2. Internally inconsistent findings: Some aspects of the manuscript provide some inconsistent results that need to be addresseda. In Figure 1E and 1F, the regions are the same (ssCTX) but the values for thee control conditions are wildly different (~80 cells per FOV in 1E v. ~190 cells per FOV in 1F). This inconsistency is glaring and needs to be explained and/or rectified.

We apologize for the inconsistency and have re-quantified these samples.

b. In Figure 1F-G, 14d recovery from PLX treatment leads to an INCREASE of ~50% (from ~190 – ~90 cells) in the ssCTX or ~20% (from ~320 – ~250 cells). However, in Figure 6F, 14d recovery from PLX leads to a 100% INCREASE in cell numbers (from ~3000 – ~6000 cells). These results are inconsistent and should be explained or reconciled.

We apologize for the reviewer if this presents a conflicting inconsistency. However, we would like to point out that Figure 1F-G only focuses on a small area of the ssCTX at 14d recovery, whereas Figure 6F involves the quantification of the entire brain. We observe in many cases, especially during a stage in which the wave is still making its way across the brain (or the cortex during that 14d recovery), some variability; some parts of the cortex will be fully repopulated and some will be packed with cells in a “wave.” We hope this helps explaining why there would be such a dramatic increase in the number of cells when looking at the entire brain vs. one small brain region.

3. Poor data presentation and Figures:a. In Figure 1, the flow of data is supposed to be a comparison between PND and END repopulation. However, for most of the results, only END data is presented e.g. 1G; 1I-L.

We have performed extensive analysis of GLOBAL repopulation over the years (Elmore et al. 2014; Elmore et al. 2015; Rice et al. 2017; Elmore et al. 2018; Najafi et al. 2018), and use this manuscript instead to characterize and focus on WM repopulation, which is why GLOBAL characterization is limited in the manuscript.

b. To be considerate of color-blind individuals, the manuscript will be helped by color coding images in green and magenta rather than green and red.

We agree with the reviewer about the importance of inclusion and consideration for color-blind individuals, unfortunately changing our color scheme at this time will be difficult, but we will utilize green and magenta for future manuscripts. We are also willing to work with the journal after acceptance to address this issue and generate more color-blind friendly images.

c. Many of the Figures provide full brain images that do not allow sufficient assessment of the claims given based on the images provided. This reviewer thinks the points will be clearer if large brain images are saved for supplemental figures and more detailed images are left in the manuscript proper. E.g. the 14d PLX condition in Figure 1D seems to have some green puncta that may be considered cells but is better determined by more focused images as in D1-4. These kinds of concerns are present in many of the Figures including Figure 2A-C (rather than the whole brain images, individual P2YR12/ IBA1 single frames in addition to the merged frames would be helpful), Figure 6G-J and Figure 9C (see 9D where the cells and amyloid can be more clearly seen).

We thank the reviewer for this suggestion. We agree that more detailed images for the reader to explore would be important, but also argue that given the nature and motility of repopulation, we believe whole brain images do provide a greater lens for exploration. To address the issue of better assessment of claims, we have imported our Figures into Adobe Illustrator, which now allow for better resolution of whole brain images and individual cells.

d. For some figure panels, there isn't always consistency in what is being shown.

If the reviewer provided a specific example, we would be happy to correct the figure panels.

e. It would be helpful if the protein stains are consistent. E.g. IBA is red in Figure 4C-E but green in Figure 4F-H. Similarly, Ki67 is magenta in Figure 4F but red in Figure 4G-H. It would be helpful to maintain the same color scheme (especially for a marker like IBA that is used throughout the manuscript) as best as possible.

We apologize for the inconsistency in our color schemes and did make every effort to try to keep IBA1 in the same channel. Unfortunately, changing the color schemes at this time for all Figures would not be feasible, but again we will be aware of this issue for future manuscripts.

f. It is not clear why the supplemental Figures are presented as e.g. for Figure S1 "Figure 1-Supplement 1". This supplemental Figure is related to Main Figure 2 not main Figure 1. This Figure is also labeled as Tmem119 in panel A but the legend says "P2RY12". The pattern in the figure is also inconsistent because the control, 14 and 28 days are in the cortex but the 7d is in the WM. This is given without explanation. If necessary, both regions can be shown but the inconsistency is confusing.

We have corrected these labeling inconsistencies. As for using WM for 7d recovery, this is because microglia are only found near the SVZ and WM areas at this timepoint. This pattern was also presented in Figure 1.

g. For Figure 2H, it seems like the expression of IBA1 is very low in the original image that is brightened in the insert. Why is this?

The Insert for Figure 2H is a 63X high resolution image of that microglial cell. The image was not brightened.

4. Outstanding questions:a. Although the study highlights the existence of the CSF1Ri-resident population, it is not clear whether this population exists in the normal brain (which would imply a "progenitor" pool) or is induced to take on this phenotype following PLX treatment. The study does not seem interested in this important question. This could potentially be determined by staining for some of the upregulated DEG genes (proteins) in the SVZ/WM of the untreated brain. If the number following 14d PLX (~15 in line 170) is much more reduced than that in the naïve brain, then it becomes difficult to argue for a "resistant" population since the population would have been reduced from whatever the number in the naive brain to 15.

We are interested in determining whether this population exists in the normal brain. We indeed explored whether our findings indicated a progenitor pool and went about staining for several precursor cell markers and lineage traced for those in Figure 5. A recent single-cell RNAseq analysis has revealed a white matter microglia-specific cluster and specifically a population of white matter-associated microglia (WAMs) that are dependent on TREM2 signaling. The similarities between these WAMs and WM repopulating cells are striking, including a shared disease-associated microglia (DAM) gene signature, along with an upregulation in genes and markers associated phagocytic activity, including CLEC7A and AXL (Safaiyan et al., 2020 *Neuron*). These recent data indicate that our WM repopulating cells are WAMs, reinforcing the existence of a distinct population of microglia in the WM that does indeed exist in the normal brain. We have updated our manuscript accordingly.

b. The H2K-BCL2 studies are interesting but a bit confusing. First, the statement that these mice provide "overexpression of BCL2 in myeloid lineages affords some resistant to CSF1Ri-induced cell death" is not substantiated by published (or new) data. In addition, no evidence is provided that microglia indeed show over expression of BCL2. Third, the mechanism of BCL2 protection from cell death is not mentioned (presumably through reduction in apoptosis?) Fourth, in the initial study by this group (Elmore et al., 2014), they suggested that CSF1Ri-induced cell death using PLX3397 occurs by apoptosis stating "To confirm that microglia undergo cell death with CSF1R inhibition, we found that many microglia stained for active caspase-3 in the same animals, a classic marker of apoptosing cells (Figure S2J- S2L)" on page 381. Since blocking apoptosis with BCL2 overexpression only increases survival from 0.2-5-10%, do the authors want to expand on whether cell death primarily occurs by apoptosis as they suggested in a previous study?

See Essential Revisions #9.

c. The language used in the discussion on lines 547-549 is strong. The authors seem to be confusing different models. Their findings do not "refute" those other findings since they used different approaches and unlike the current findings, it is likely that an empty microglial niches did not occur in those studies.

We have tempered our language in the discussion accordingly.

[Editors' note: further revisions were suggested prior to acceptance, as described below.]

(1) The paper has significantly improved. However, there are still concerns about the scope of the claims about the surviving microglia being the so-called WAMs. The fact that this small number of cells in WM survives could be due to many factors including lower drug bioavailability in WM. Please tone down this claim and explain possible alternative reasons for the survival of these cells post-drug treatment.

We thank the editors for their comment. We have toned down the claim that the surviving microglia are WAMs (see lines 47, 49, 110, 486 (deletion), 488-9, 709-11, 720, 723 (deletion), 727-8). We have also added to our section in the discussion explaining the alternative reasons for microglial survival in WM areas post-drug treatment (see previously added lines 645-57, newly added lines 654-5, 657-8, 659-72) as well as to the Results section (see lines 226-9).

(2) While in this particular case it appears that WM microglia are responsible for most of the cortical repopulation, this is a unique situation and may not reflect a specific property of WM microglia. Several papers including recent ones in eLife have clearly shown repopulation from surviving cortical microglia. Please include these in the discussion and tone down the claims of the uniqueness of WM microglia in their ability to repopulate the brain ( PMID: 33054973, PMID: 34250902)

We agree with the editors that repopulating cortical areas is not a specific or unique property of WM microglia. In 2014, we identified the unique ability of microglia (including cortical microglia) to repopulate the brain following CSF1R inhibition (Elmore et al., 2014 *Neuron*). Thus, it was never our intention to make this claim. Under repopulation conditions in which an empty microglial niche is not achieved (< 98% IBA1+ cell loss), so in the papers that are cited above in which 88% (Zhan et al. 2020) and 75% (Mendes et al. 2021) microglial loss were reported, cortical repopulation can be derived from cortical microglia. Under conditions in which < 98% microglial loss is achieved, we term this form of repopulation “GLOBAL repopulation,” and would like to clarify that repopulating cells derive from the nearest surviving microglia in the brain region where the repopulating cells appear. For this reason, we state on lines 122-3, “repopulation is dependent on the local proliferation and clonal expansion of surviving microglia.” However, under WM repopulation (> 98% IBA1+ cell loss), there are no surviving cells in a particular brain region to give rise to cells, thus cortical repopulation (even hippocampal or thalamic repopulation) does not occur due to surviving microglia in that area. We apologize for the confusion and have included a clarification of this in the discussion (see lines 590-97). We have also toned down our claims about the uniqueness of WM microglia to repopulate the brain (see lines 43 (deletion), 44 (deletion), 45 (deletion), 725-6 (deletion)). We have also added these suggested citations (see lines 123, 668).

(3) Please go through the critiques from the 3 reviewers and address them as much as possible. No need for further experiments, just data clarifications, and additions to the Discussion section.

We have gone through the critiques from the 3 reviewers and have addressed them as much as possible (see below).

Reviewer #1:In the revised manuscript Hohsfield et al. have answered many of the concerns raised. Specifically:1. In the earlier version of this manuscript the images showing the "wavefront" of repopulating microglia in the cortex was not very convincing. In the revised manuscript, as per our suggestion, the authors have added high-resolution confocal tiling of large areas showing the wavefront at different stages of the recovery. This data does show more convincingly that dividing microglia emerge from areas newa corpus callosum and appear to advance towards cortex. However, this data also seems to show patchy cortical areas with proliferating microglia that to not appear to emerge from the Wavefront. This suggest that other sources of regenerating microglia exist in the courtext (likely small number of surviving microglia)

We have noted this observation in the Results section (see lines 190-2).

2. To address our concerns about the uniqueness of microglia in the white matter tract and their resistance to PLX3397 treatment, the authors postulate based on recent literature that these microglia are resistant due to enhanced TREM2 and CSF1 expression. Additionally, the authors added bulk tissue transcriptome data of 14 days PLX-treated and after 3-5 days recovery time. The authors show that the initial repopulating microglia have genes upregulated in the pathways related to myeloid cell activation/priming, pathogen sensing, and monocyte-macrophage signaling.

We are glad to have addressed the reviewer’s concerns.

3. We had concerns that whether after longer recovery time post PLX treatment, the repopulated microglia would still be transcriptionally different or not. To address this the authors have added transcriptome data after 3 months of recovery and suggest that they are transcriptionally different from the control.

We are glad to have addressed the reviewer’s concerns.

4. We asked whether a longer duration of PLX3397 treatment will eliminate microglia from SVZ and white matter tract or not. The authors in the revised manuscript report that a longer duration of PLX3397 (3 months) eliminates microglia from the white matter region. This suggests that the small population of surviving microglia in WM has a different threshold for cell death, or the bioavailability of the compound is somehow lower in WM compared to cortex. Ideally the authors should;d discuss this in greater detail in their final version as it is difficult to say with the current data the precise reason for the relative sparing of some WM microglia

We have added to our section in the discussion explaining the alternative reasons for microglial survival in WM areas post-drug treatment (see previously added lines 645-57, newly added lines 658-72) as well as to the Results section (see lines 226-9).

5. The authors took our suggestion to completely remove the Alzheimer's disease model data as the data was not conclusive and to add the H2K-BCL2 with more characterization in the main figures.

We are glad to have addressed the reviewer’s concerns.

Overall, as it is the paper is comprehensive and contains interesting data related to the repopulation dynamics of microglia post extensive elimination using PLX3397 for 14 days. The repopulating microglia are transcriptionally different than the control homeostatic microglia and they maintain this difference up to 3 months of recovery. Since the origin of this most (although not likely all) repopulating microglia is from white matter, the authors claim to uncover new insights into white matter microglia function and dynamics. However, we are not very confident about the authors claim of the uniqueness of the white matter microglia. As the authors showed that a longer duration of PLX3397 treatment (3 months) could completely eliminate microglia from white matter, suggesting that they are susceptible to PLX3397 too. Also, the perfusion of this drug in different regions of the brain is not understood. The transcriptional differences observed in the repopulating microglia could be due to the enhanced load of phagocytosis due to extensive PLX3397 treatment. A recent article published in eLife by Mendes et al. (eLife 2021;10:e61173) using in vivo 2 photon imaging showed that the microglia post depletion in cortex repopulate rapidly and locally rather than from migrating microglia and attain morphological maturity within days. Thus, we suggest authors tone down their conclusion about uncovering the uniqueness of white matter microglia. I would suggest changes in the title and abstract which at the moment to us appear to strongly suggest that WM are very unique and the pathway for migration towards cortex could have implications during development, etc. These are interesting hypotheses but are not necessarily well supported by the data

We thank the reviewer for their comment. We have toned down our claims about the uniqueness of WM microglia to repopulate the brain (see lines 43 (deletion), 44 (deletion), 45 (deletion), 725-6 (deletion)). However, we would like to highlight that we provide several examples of the uniqueness of these cells to homeostatic microglia in our manuscript, aside from their initial resistance to 14d PLX3397 (600 ppm) treatment, including their migratory patterns during repopulation, morphology, expression of cell surface markers, and transcriptional profile. We also provide experiments that describe their ability to fill the microglial niche without repercussions on other cell types or behavior, indicating the nuanced complexity of distinct microglial populations and their functions. We would also like to point out that we are not alone in our claim that white matter microglia are unique, other studies have demonstrated this as well (Safaiyan et al., 2021, van der Poel, 2019, Hagemeyer et al., 2017). It should also be noted that the report by Mendes et al. (2021) uses 7d PLX5622 treatment, which results in partial depletion – 75% of microglia are depleted. Under this paradigm, GLOBAL repopulation would occur, which we state in the manuscript derives from surviving microglia and exhibits distinct properties (such as the appearance of clusters of surviving microglia that proliferate to give rise to repopulating cells) compared to WM repopulation (with the majority of cells deriving from a wave of proliferating cells). We postulate that there is a certain threshold of microglial depletion (what we define as an empty microglial niche of > 98% depletion) that must be reached in order for WM repopulation to occur. We have made changes to the abstract, but would argue that the title succinctly and appropriately describes our findings in this manuscript. We would like to know if there is room for discussion with the editors on other appropriate titles.

Reviewer #2:The manuscript addressed many of the concerns, and is much improved in the quality of the presentation. A comparison with the WAM from the recent study is interesting. A. highly relevant eLife paper published last year reported single cell RNA-seq analyses of a a relative resistant population ( ELife. 2020 Oct 15;9:e51796. doi: 10.7554/eLife.51796). How do these two resistant populations compare in terms of transcriptomic signatures?

The reviewer highlights an interesting paper that we also found extremely relevant for our study. We have added this paper to our Discussion section (see lines 666-72). The authors identify a MAC2+ progenitor-like microglial cell that is resistant to CSF1Ri. It is important to note that the authors do not achieve an empty microglial niche, reporting a loss in 88% of IBA1+ cells, meaning this type of repopulation (GLOBAL repopulation) is different than WM repopulation we describe in the current manuscript. Although we have not done comprehensive transcriptional analysis between these cells, the authors report an upregulation in *Lyz2* and MHC-associated genes, and a downregulation in *Tmem119, Mafb, Cx3cr1*, and *Csf1r*. Our repopulating cells also upregulate *Lyz2* and MHC-associated genes, and downregulate many canonical microglial signature genes, however, we would like to point out that DAMs and WAMs also display a similar transcriptional profile. Thus, we cannot draw many conclusions from this data. We have also stained our WM repopulating cells for MAC2 and although some of the surviving cells in the WM are MAC2+, the repopulating cells that give rise to the microglial population are MAC2-. We are in the midst of preparing a follow-up manuscript that will detail and clarify these findings.

Reviewer #3:This seems to be a much-improved manuscript but this reviewer cannot fully assess the revisions because it lacks appropriate notification of the changes in the text to allow visualization / assessments of the amendments. For example, for Reviewer 3's comment 1f, the authors state that they corrected the reference to the recovery to normal levels of the cell soma. But this reviewer does not see this mention in the text. The same is true for the response to comment 1g and 1i. Moreover, page numbers referenced in the Essential Revisions do not match up with the text. Also, in response to Reviewer 3 question 1d, Figure 9 is mentioned rather than Figure 10. Also, for 2b, "Figure 6F" is no longer that and the new figure should be mentioned in the response. In response to question 4b, the authors reference Essential Revisions 9 but the response is Essential Revisions 10. This reviewer considers this a set of very unfortunate failures that excessively prolonged this review that has made reviewing the revision extremely difficult.

We apologize to the reviewer for their frustrating reviewing experience. It was not our intention to make this revision difficult. We provided a manuscript that identified revised text changes in red in our last revision. We are unclear as to why the reviewer did not receive this version. We have also noticed that the PDF version of the article file does not have accurate line numbers. We are unsure as to why this occurs, but would recommend that the reviewer use the word file for accurate line numbers. We have added line numbers to this response so the reviewer can better find our changes. Please note that previous text changes made in the last round of revisions are colored in orange and the recently made text changes for this round of revision are in blue (both chosen based on colorblind friendly recommendations).